# A Framework for Automated Supraglacial Lake Detection and Depth Retrieval in ICESat-2 Photon Data Across the Greenland and Antarctic Ice Sheets

Philipp Sebastian Arndt[1] and Helen Amanda Fricker[1]

[1]Scripps Polar Center, University of California San Diego, 8885 Biological Grade, La Jolla, CA 92037, USA

**Correspondence:** Philipp Sebastian Arndt (parndt@ucsd.edu)

**Abstract.** Water depths of supraglacial lakes on the ice sheets are difficult to monitor continuously, due the lakes' ephemeral nature and inaccessible locations. Supraglacial lakes have been linked to ice shelf collapse in Antarctica and accelerated flow of grounded ice in Greenland. However, the impact of supraglacial lakes on ice dynamics has not been quantified accurately enough to predict their contribution to future mass loss and sea level rise. This is largely because ice-sheet-wide assessments of meltwater volumes rely on models that are poorly constrained due to a lack of accurate depth measurements. Various recent case studies have demonstrated that accurate supraglacial lake depths can be obtained from ICESat-2's ATL03 photon-level data product. ATL03 comprises hundreds of terabytes of unstructured point cloud data, which has made it challenging to use this bathymetric capability at scale. Here, we present two new algorithms – Flat Lake and Underlying Ice Detection (FLUID) and Surface Removal and Robust Fit (SuRRF) – which together provide a fully automated and scalable method for lake detection and along-track depth determination from ATL03 data, and establish a framework for its large-scale implementation using distributed high-throughput computing. We report FLUID/SuRRF algorithm performance over two regions known to have significant surface melt – Central West Greenland and Amery Ice Shelf catchment in East Antarctica – during two melt seasons. FLUID/SuRRF reveals a total of $1249$ ICESat-2 lake segments up to $25\,\mathrm{m}$ deep, with more water during higher melt years. In absence of ground truth data, manual annotation of test data suggests that our method reliably detects melt lakes along ICESat-2's ground tracks whenever the lakebed is visible or partially visible, and estimates water depths with a mean absolute error $< 0.27\,\mathrm{m}$. These results imply that our proposed framework has the potential to generate a comprehensive data product of accurate meltwater depths across both ice sheets.

## 1 Introduction

Earth is warming and both of its ice sheets (Greenland and Antarctica) are losing mass to the ocean at increasing rates (Rignot et al., 2011; Smith et al., 2020), leading to sea level rise. There is growing evidence that some of this retreat is irreversible, thus committing coastal communities to embrace costly sea level rise adaptation strategies for decades and centuries to come (DeConto et al., 2021; Garbe et al., 2020; Gregory et al., 2020; Nordhaus, 2019). To address the resulting societal challenges, policy makers and coastal planners require accurate and actionable sea level rise projections (Moon et al., 2020). However, the projected contribution from the ice sheets is highly uncertain, to the point that the Sixth Assessment Report of the United

Nations Intergovernmental Panel on Climate Change designated it as *"deep uncertainty"* (IPCC AR6; Fox-Kemper, 2021). Building confidence in projections of the ice sheets' contribution to future sea level rise requires better mechanistic understanding of relevant mass balance processes for inclusion in ice sheet models (Golledge, 2020; Aschwanden et al., 2021). However, ice sheet wide details of many of these processes are poorly known because they have been under observed in both space and time.

Supraglacial melting on the ice sheets is one example of a process which has a potentially important contribution to future sea level rise projections, yet has been under observed so is poorly understood. In a warming climate, supraglacial lakes have the potential to trigger positive feedback loops and catastrophic collapse (Gilbert and Kittel, 2021), yet the underlying mechanisms and associated likelihoods are vaguely defined due to a lack of high-quality observations (Arthur et al., 2020a). In particular, models that attempt to capture the influence of supraglacial hydrology on ice sheet behavior require accurate estimates of

volumes of pooled surface meltwater as input (Zwally et al., 2002; Parizek and Alley, 2004; Krawczynski et al., 2009; Robel and Banwell, 2019). However, there are few direct *in situ* observations of supraglacial lake depths (none in Antarctica, and ten lakes up to $11.5\,\text{m}$ deep in Greenland), which leads to errors in total water volume estimates. This introduces biases into model inputs for meltwater flow, impacting projections of future ice sheet evolution (Melling et al., 2024). To ensure that coupled hydrological-dynamical models accurately represent the underlying physics, it is important to find a method to acquire lake

depths that are accurate, and also spatially and temporally and continuous.

Launched in 2018, NASA's Ice, Cloud and land Elevation Satellite (ICESat-2) laser altimeter became the first (and thus far only) satellite capable of making direct, accurate water depth measurements from space, due to its green light being able to penetrate water, which allows its sensor to register the elevation of photons that were reflected from both the lake surface and the lakebed (Fig. 1; e.g., Fair et al., 2020; Fricker et al., 2021; Xiao et al., 2023). This allows ICESat-2 to measure water

depths up to $41\,\text{m}$ under ideal conditions (very clear water and high bottom reflectivity), with typical accuracies of about $0.5\,\text{m}$ (Dietrich et al., 2024). While ICESat-2 has the unique capability to make direct and accurate measurements of water depth from space, its fundamental limitation is spatial coverage. ICESat-2 data are limited to discrete, one-dimensional ground tracks that are coarsely spaced on the Earth's surface ($\sim 9.9\,\text{km}$ between neighboring reference tracks and $\sim 3.3\,\text{km}$ between all neighboring beam pair tracks at $70\,^\circ\,\text{N/S}$) with a relatively long revisit time of three months. This means that no supraglacial

lake depth data product derived from ICESat-2 *alone* is able to provide samples of all (or even nearly all) supraglacial lakes on the ice sheets: ICESat-2's track spacing means that the majority of lakes form in locations that ICESat-2 ground tracks never sample, and the three-month return period means that for a significant number of ground tracks ICESat-2 never passes over at the time at which melt lakes are visible. ICESat-2 is also unable to penetrate optically thick clouds, thus further limiting the amount of data available for water depth measurements.

While ICESat-2 data alone cannot be used to continuously monitor melt lake volumes, several case studies have shown that ICESat-2 depth measurements can be used to constrain parameters in models that estimate lake volumes from satellite imagery (Datta and Wouters, 2021; Leeuwen, 2023; Lutz et al., 2024). For instance, Datta and Wouters demonstrated that it is possible to accurately extrapolate depths along ICESat-2's ground track segments to the full lake basins that these segments intersect. To be able to use ICESat-2 to improve depth estimates of supraglacial lakes in locations where (and at times when)

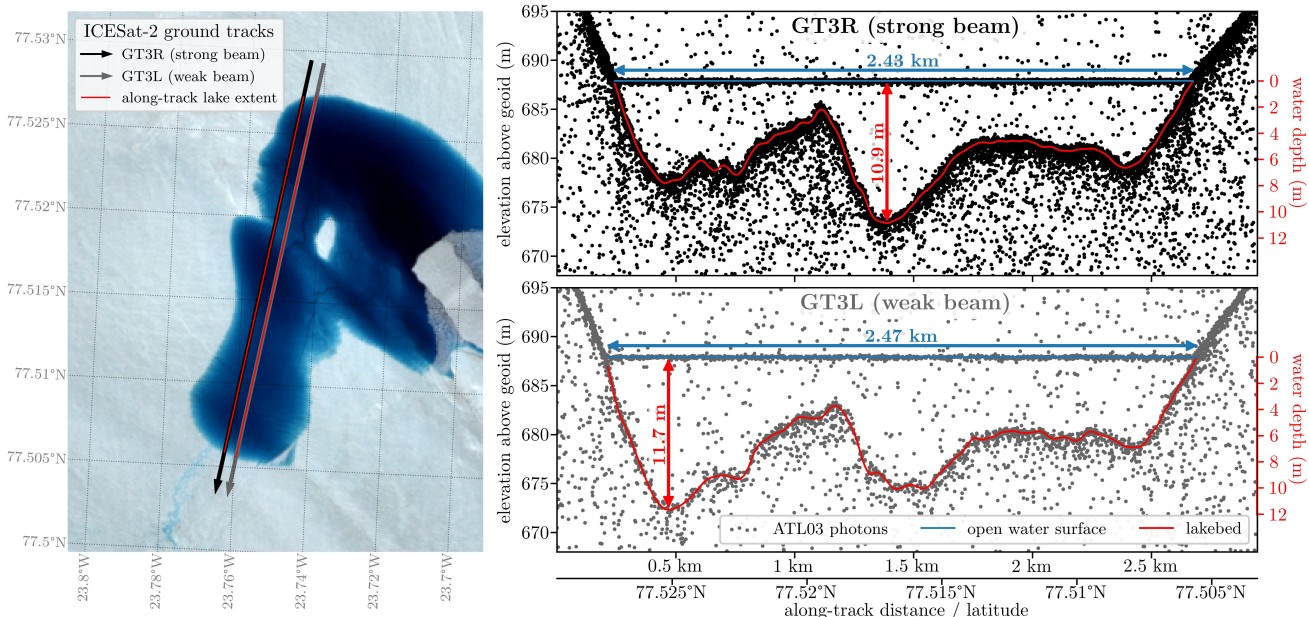

**Figure 1.** An ICESat-2 ATL03 data segment over a supraglacial lake, showing a particularly strong bathymetric return from the lakebed. (Data from ICESat-2 Track 406 on 20 July 2021; granule: ATL03_20210720053125_04061205_006_01.h5. The imagery in the left panel is a Sentinel-2 scene from the same day: S2B_MSIL2A_20210720T151809_N0301_R068_T27XVG_20210720T175839.)

ICESat-2 measurements are not directly available, it will be necessary to rely on statistical methods that can generalize the relationship between water depth and reflectance for a particular passive optical sensor under a wide variety of conditions and independently of the availability training data that are close-by in space and time (Hastie et al., 2009). For this to work effectively, the data that is used to train statistical learning models capable of multiple non-linear regression for representing a complex depth-reflectance relationship need to adequately cover the parameter space defined by the combination of predictors that are included (Markham and Rakes, 1998; Wang et al., 2022). Since ICESat-2 observations of melt lakes are relatively sparse, it is therefore crucial to to obtain as many ICESat-2 depth estimates as possible from different locations and times (and thus under a wide variety of environmental conditions) to be able to effectively use ICESat-2 to improve monitoring of meltwater volumes across the ice sheets. This suggests that large-scale extraction of accurate supraglacial lake depths from a wide range of ICESat-2 photon-level data in combination with concurrent optical satellite imagery can provide a labeled training data set enabling the application of machine learning methods (e.g., Leeuwen, 2023) capable of generating a well-constrained data-driven model for ice-sheet-wide lake volume estimation (Melling et al., 2024).

While automated and scalable algorithms for lake detection and depth retrieval in ATL03 photon data have been proposed (e.g., Datta and Wouters, 2021; Xiao et al., 2023), in practice no previous ICESat-2 studies have applied supraglacial lake depth estimation methods to more than a handful of manually picked lake segments or data granules, or presented a straightforward pathway to large-scale computational implementation across the ATL03 data catalog, which comprises hundreds of terabytes

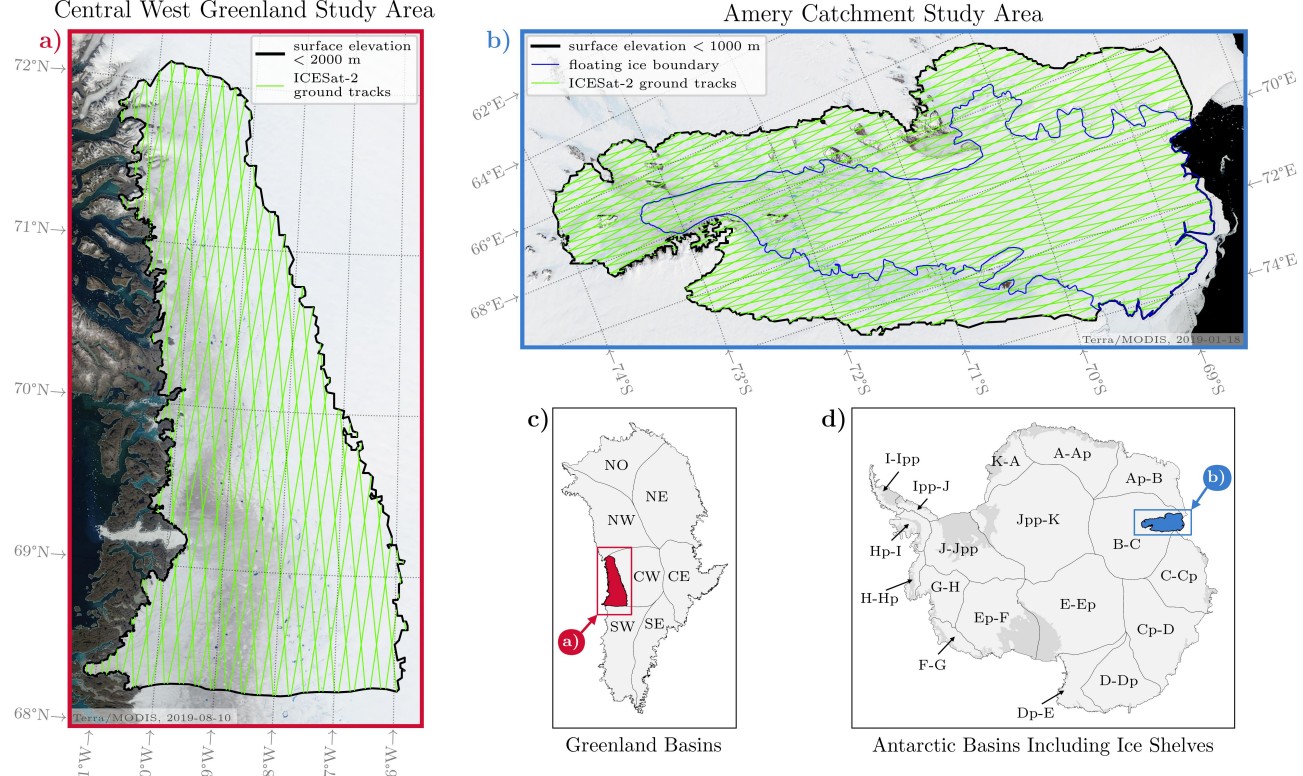

**Figure 2.** Study regions for testing the FLUID/SuRRF framework. a) and b) Maps of the two regions chosen for this study: Central West Greenland (CW drainage basin) and the Amery catchment (B-C drainage basin). The black outlines show the boundaries of the regions which were obtained by thresholding the corresponding ice sheet drainage basins (c and d) by the elevations shown in the legends. The green lines show ICESat-2 reference ground track coverage. c) and d): Maps of the Ice sheet Mass Balance Inter-comparison Exercise (IMBIE) drainage basins for Greenland (Mouginot and Rignot, 2019) and Antarctica (Mouginot et al., 2017). Insets show the locations of the two study areas.

of unstructured point cloud data (Neumann et al., 2023b). To address this challenge, we present a framework for ice-sheet-wide implementation of our own fully automated and scalable algorithm for along-track lake segment detection and depth determination from ICESat-2 data. Here, we present this algorithm, apply it to two entire drainage basins in Greenland and Antarctica (Sect. 3.5, Fig. 2) using distributed high-throughput computing, and demonstrate its performance for two full melt seasons.

## 2 Background

### 2.1 How supraglacial lakes affect ice sheet mass loss

Supraglacial water has different roles in Greenland and Antarctic ice sheet mass loss processes, largely because of its different spatial extent on each ice sheet. Across most of the Greenland Ice Sheet's ablation zone, meltwater pools in supraglacial lakes that extend from the ice margins up to about $2000\,\mathrm{m}$ on the plateau, and are forming further inland as temperatures increase (Leeson et al., 2015; Tedstone and Machguth, 2022). On the Antarctic Ice Sheet, pooling of surface meltwater in lakes is not as pervasive and is mostly observed on the floating ice shelves and at low elevations near their grounding zones (Stokes et al., 2019; Corr et al., 2022), with large regional and interannual variability (Arthur et al., 2022). Pooling and storage of meltwater in supraglacial lakes can affect ice sheet mass loss directly or indirectly in four ways:

1. *Surface runoff:* Supraglacial lake drainage and transport of water off the ice sheet through surface streams or englacial pathways contributes to mass loss directly as surface runoff. This is already a significant component of the Greenland Ice Sheet surface mass balance (The IMBIE Team, 2020), but has also been observed on the Antarctic Ice Sheet (Bell et al., 2017; Warner et al., 2021; Trusel et al., 2022) and could become more significant in a warming future (Kingslake et al., 2017; Bell et al., 2017). Such runoff also results in surface elevation lowering, which further increases meltwater production by exposing the ice sheet surface to the higher temperatures found at lower elevations (Levermann and Winkelmann, 2016; Bell et al., 2018).

2. *Surface albedo lowering:* Supraglacial lakes lower the surface albedo, which can further accelerate melting and result in a temperature increase in the adjacent ice column (Tedesco et al., 2012; Ryan et al., 2017; Stokes et al., 2019).

3. *Bedrock lubrication:* On grounded ice, rapid drainage of surface lakes by hydrofracture delivers pulses of meltwater to the base of the ice sheet, which has the potential to lubricate the bedrock and cause acceleration of ice flow due to enhanced basal sliding. This is a well-studied phenomenon in Greenland (e.g., Das et al., 2008; Bartholomew et al., 2010; Tedesco et al., 2013; Davison et al., 2019; Maier et al., 2023) but recent observations suggest that this mechanism is also driving ice flow speed-ups on the Antarctic Ice Sheet (Tuckett et al., 2019), where it could become an increasingly important mechanism as future warming will cause its hydrology to become more similar to Greenland's current ablation zone (Bell et al., 2018).

4. *Ice shelf collapse:* In Antarctica, the ponding and draining of supraglacial lakes can weaken and fracture the floating ice shelves (Munneke et al., 2014; Banwell and Macayeal, 2015; Banwell et al., 2019; Lai et al., 2020), which, in extreme cases, has been linked to their collapse by hydrofracture (MacAyeal et al., 2003; Scambos et al., 2004; Banwell et al., 2013). The resulting loss of buttressing back-stresses leads to accelerated discharge of upstream grounded ice into the ocean, which causes sea level rise (De Angelis and Skvarca, 2003; Scambos et al., 2004; Rignot et al., 2004; Rott et al., 2018). It has been hypothesized that these melt-driven hydrofracture processes could expose marine ice cliffs that are sufficiently tall and weakened to be prone to mechanical failure, which would trigger buoyancy-driven calving and could

therefore lead to sustained, rapid ice sheet collapse, referred to as the Marine Ice Cliff Instability (MICI; Bassis and Walker, 2012; Pollard et al., 2015; DeConto and Pollard, 2016; Bassis et al., 2021, 2024).

Incorporating those processes through which surface meltwater ponding affects ice dynamics into ice sheet models can drastically increase projected future sea level rise (Martin et al., 2019; Edwards et al., 2021), yet they currently rely on poorly constrained parametrizations, making projections highly uncertain (Robel et al., 2019; Pattyn and Morlighem, 2020). This means that there is an urgent need to improve our understanding of the key underlying physical processes based on accurate observations (Hanna et al., 2024).

## 2.2   Observations of supraglacial lake depths

*In situ* observations of melt lake depths are scarce (e.g., Tedesco and Steiner, 2011) due to the challenging logistics and planning required to collect such data. Supraglacial hydrological systems on ice sheets form seasonally in some of Earth's most remote and inaccessible locations, and they can rapidly evolve in complex and unpredictable patterns (Dirscherl et al., 2020; Gantayat et al., 2023), making survey planning difficult. Therefore, to obtain ice-sheet-wide observations of meltwater lake depths for
each melt season, it is necessary to rely on satellite remote sensing techniques (Moussavi et al., 2016; Melling et al., 2024). Besides ICESat-2's novel capability to directly observe water depths from photon refraction, various methods have been used to indirectly estimate lake depths from satellite data, which all have different advantages and disadvantages.

One such method is to apply a **radiative transfer equation** (RTE, Philpot, 1987, 1989) to estimate lake depth from optical imagery (Sneed and Hamilton, 2007; Moussavi et al., 2020; Leeson et al., 2020). This approach has been widely used since
optical imagery provides continuous spatial coverage at short temporal intervals and because it is assumed that its physics-based principles hold everywhere, which makes it possible to apply it at scale. However, the RTE approach relies on poorly constrained choices of water attenuation coefficients and lakebed albedo, and makes simplifying assumptions such as: no suspended particulate matter; a homogeneous lakebed albedo; no surface disturbances caused by wind; and the water column has vertically homogeneous optical properties (Brodskỳ et al., 2022). As a result, it has been shown that the RTE approach can
significantly over- or underestimate lake depths in different environments: in Fricker et al. (2021) the RTE method underestimated depths by 30 to 70 % on Amery Ice Shelf in East Antarctica, whereas in Melling et al. (2024) it overestimated depths by up to 153 % in Southwest Greenland.

Another approach to estimating lake depths is using **empirical models** derived from regression of *in situ* depth measurements with optical imagery (e.g., Tedesco and Steiner, 2011; Legleiter et al., 2014; Pope et al., 2016). However, *in situ* measurements
of supraglacial lake depths are very sparse, with (to the best of our knowledge and at the time of writing) no such data available for Antarctica, and data available for only ten lakes up to $11.5\,\mathrm{m}$ deep on the Greenland Ice Sheet between 2005 and 2024: Box and Ski (2007) sampled two lakes on Jakobshavn Isbræ and Sermeq Avannarleq in 2005, Sneed and Hamilton (2007) sampled one lake on Helheim Glacier in 2008, Tedesco and Steiner (2011) sampled one lake in Central West Greenland in Legleiter et al. (2014) sampled three supraglacial water bodies on Isunnguata Sermia, Russell Glacier in 2012 and Lutz et al.
(2024) sampled three lakes on Zachariæ Isstrøm in 2022. This makes the observations provided by Lutz et al. the only *in situ*

depth data of supraglacial lakes that overlap with the Landsat 8 and Sentinel-2 missions. Further, it has been shown that the relationship between water depth and reflectance values in optical imagery can vary significantly by geographical region (Lutz et al., 2024). Thus, the regression coefficients of these empirical models are limited to the spatial area of the original *in situ* measurements, making them impractical for application on a larger, ice-sheet-wide scale.

A third approach is to use **digital elevation models** (DEMs) of a lake's bed topography that were acquired before it filled or after it drained, and then to determine its fill level from imagery (Moussavi et al., 2016; Yang et al., 2019b). While this has the advantage of being independent of the optical properties of the water column, currently available DEM acquisitions are sporadic and the method cannot account for changes in the lakebed topography between acquisitions, due to, for example, bottom ablation (Tedesco et al., 2012). Because this approach requires acquisitions from before a lake fills or after it drains,

it is not suitable for perennial lakes that freeze over and are buried in winter without draining (Koenig et al., 2015; Schröder et al., 2020; Leppäranta et al., 2013), and it cannot be directly applied to lakes on floating ice shelves, where any filling and draining events result in a hydrostatic adjustment that bends the ice surface (Scambos et al., 2009; Warner et al., 2021).

## 3    Data and methods

We use individual photon data from the ICESat-2 instrument, provided in a data product known as Global Geolocated Photons

(ATL03; Sect. 3.1). Our method to extract water depths consists of two algorithms that are run consecutively on ATL03 data: (1) Flat Lake and Underlying Ice Detection (FLUID, Sect. 3.2) automatically detects the locations of supraglacial lakes visible in the point cloud data; (2) Surface Removal and Robust Fit (SuRRF, Sect. 3.3) determines the along-track depth for each detected lake segment. To automatically compute results for large numbers of ATL03 data files over extensive geographical regions, we use the Open Science Grid's (OSG) Open Science Pool for distributed High-Throughput Computing (Sect. 3.4;

OSG, 2006; Pordes et al., 2007). Figure 3 summarizes the various steps of this method in a flowchart, and we describe each of them in more detail below.

### 3.1   ICESat-2

ICESat-2 was launched in September 2018 and carries the Advanced Topographic Laser Altimeter System (ATLAS) instrument, a photon-counting green light ($532\,\mathrm{nm}$) laser altimeter operating at a frequency of $10\,\mathrm{kHz}$, which results in a $0.7\,\mathrm{m}$

along-track resolution (Markus et al., 2017). ATLAS divides the laser pulse it emits into six beams, forming three beam pairs, each of which consist of a weak and a strong beam. The footprint of each beam is about $11\,\mathrm{m}$ in diameter on the ground (Magruder et al., 2021a). The six resulting ground tracks (GTNXs) are referred to as GT1L, GT1R, GT2L, GT2R, GT3L and GT3R, where the number N refers to the beam pair from left to right in direction of flight, and the X refers to the left ("L") or right ("R") track within each pair (Neumann et al., 2022). The three track pairs are separated $3.3\,\mathrm{km}$ on the ground, and the two

tracks within a pair are separated by $90\,\mathrm{m}$ each. The ICESat-2 spacecraft can fly in either forward or backward orientation, and flips between the two approximately twice a year (Neumann et al., 2019; Smith et al., 2019). In forward orientation, ATLAS's strong beams – numbered 1, 3 and 5 – are on the right side of each beam pair in the direction of flight and point to GTs 3R,

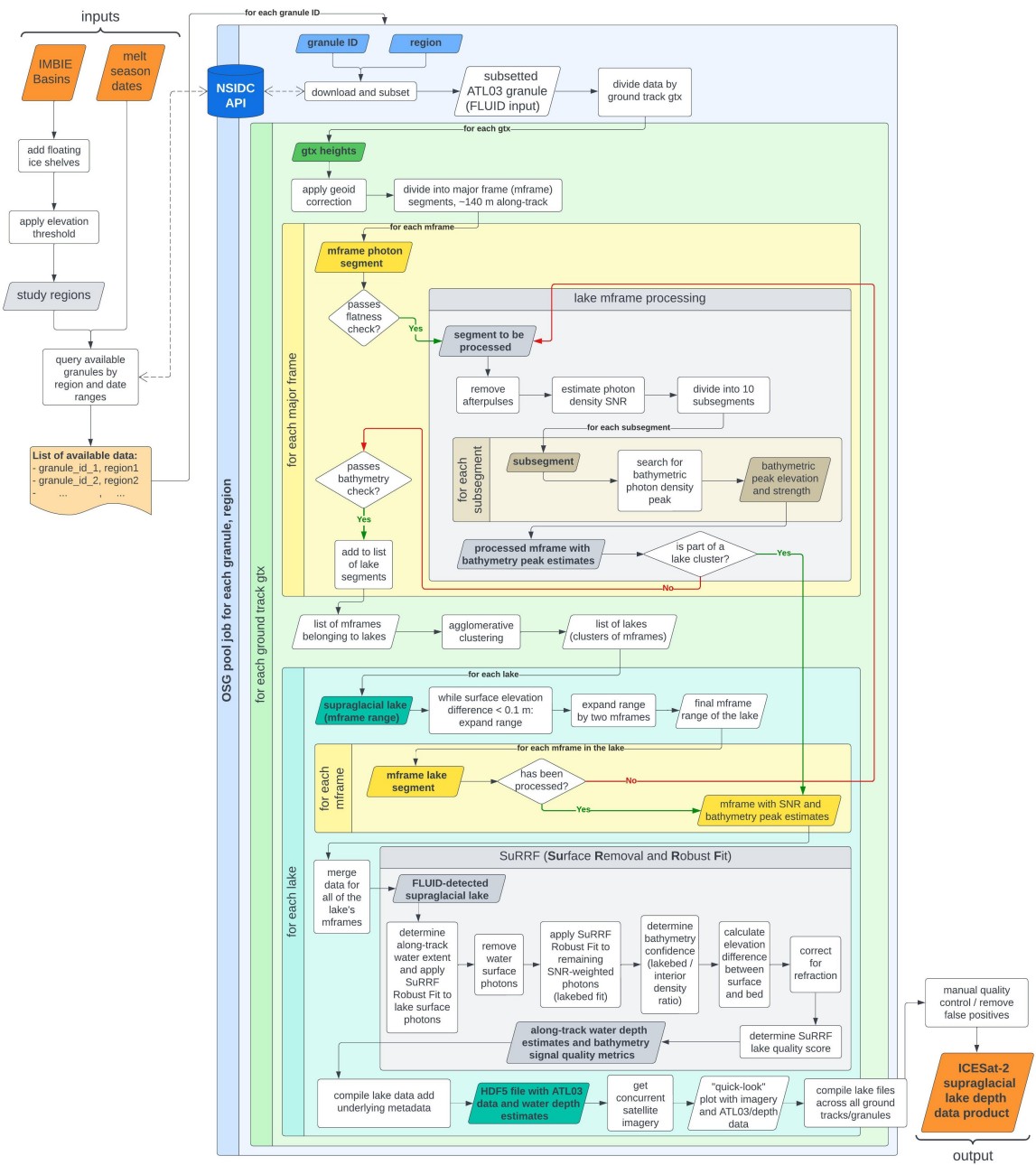

**Figure 3.** Flowchart of the FLUID/SuRRF framework for detecting and determining the depths of supraglacial melt lakes in ICESat-2 data for any melt season over any drainage basin of the Antarctic or Greenland ice sheets. All modules in the blue box can be parallelized for large amounts of input data granules, as a batch of compute jobs on a computer cluster or a platform for distributed High-Throughput Computing, such as the OSG Open Science Pool.

2R and 1R, respectively. Similarly, after a yaw flip to the backward orientation, the strong beams are on the left side of each beam pair and point to GTs 1L, 2L and 3L. This means that the data acquired along a particular GTNX can be associated with a strong or a weak beam, depending on the spacecraft orientation at the time of data acquisition. The ATLAS receiver uses photomultiplier tubes (PMTs) designed to detect individual reflected photons, with 16 independent timing channels for each strong beam and four for each weak beam (Yang et al., 2019a). The strong beams have four times more energy than the weak beams, resulting in a correspondingly higher count of laser photons returned per shot.

ICESat-2 repeats its orbit every 91 days, after completing $1,387$ distinct reference ground tracks (RGTs). Over land ice, ATLAS routinely points to these RGTs near its nadir to acquire repeat measurements (Magruder et al., 2021b). The satellite began targeting the planned RGTs in late March 2019, once the on-orbit pointing calibrations were finalized and updated in the onboard pointing control systems (Martino et al., 2019). Consequently, any observations during the 2018-19 Antarctic melt season do not align with the planned repeat tracks. ICESat-2 was in "safe-hold" from 26 June through 9 July 2019, which means that no data was collected during this $14$-day period coinciding with the 2019 Greenland melt season.

Over shallow ($< 41\,\mathrm{m}$) and non-turbid water bodies, ATLAS's green light is able to pass through the water column, which means that signal photons can be reflected from both the flat open-water surface and the lakebed (Fair et al., 2020; Fricker et al., 2021). Most land ice applications use the ATL06 data product designed for glacier and ice sheet surfaces (Smith et al., 2019). However, the ATL06 algorithm provides only one surface and thus cannot be used to extract meltwater depths. Also, over melt lakes ATL06 segments inconsistently track either the water surface or the lakebed, so the results are ambiguous (Fricker et al., 2021). To overcome this limitation and track both surfaces, our technique relies on the elevations of individual photons, which are distributed in the Global Geolocated Photons (ATL03) data product (Neumann et al., 2023b). To keep the size of each individual ATL03 data file (or "granule") manageable, each RGT orbit of ICESat-2 data is divided into 14 granule regions (Neumann et al., 2023a). This means that each granule is limited to approximately $30\,\mathrm{min}$ of along-track data and rarely exceeds $10\,\mathrm{GB}$ in size. ATL03 reports geolocated photon attributes such as longitude, latitude, along-track distance and height for each individual photon detection event, thus providing an along-track point cloud of photon locations. Geophysical corrections (such as geoid height) are reported at a $20\,\mathrm{m}$ along-track segment rate, and parameters related to on-board data processing (such as telemetry window ranges) are reported at the $50\,\mathrm{Hz}$ ($\approx 140\,\mathrm{m}$ along-track) "major frame" rate (Martino et al., 2022b). In an ATL03 point cloud, signal photons being reflected both from a lake's water surface and its lakebed results in characteristic double returns, which are used by FLUID to detect along-track data segments containing supraglacial lakes and by SuRRF to generate depth estimates for those lake segments. While the strong beam data have a higher signal to noise ratio, we have designed our FLUID/SuRRF method to work well with both strong and weak beams whenever a bathymetric return from the lakebed is discernible, i.e. for lakes with a visible or partly visible lakebed.

### 3.2 Supraglacial lake detection in ATL03: the FLUID algorithm

The Flat Lake and Underlying Ice Detection (FLUID) algorithm takes an ATL03 granule as input, searches for locations that contain potential supraglacial lakes with a bathymetric signal, and then returns along-track segments of the data for all detections. FLUID exploits two unique characteristics of supraglacial lake segments in ATL03 data:

1. photons which are reflected back from an open water surface cluster around a flat line (Sect. 3.2.1, Fig. 4) and

2. a bathymetric return signal must present as a secondary peak in photon density below such a flat surface (Sect. 3.2.4, Fig. 7).

To search for supraglacial lake segments in ATL03, FLUID divides the photon data into $140\,\mathrm{m}$ along-track segments aligning with ICESat-2's "major frames" and selects those that satisfy both of the above requirements. Then, adjacent major frames are iteratively clustered into larger along-track data segments that likely represent all available ATL03 data for an entire supraglacial lake (Sect. 3.2.5).

### 3.2.1    FLUID step 1: identification of flat water surfaces

This step uses the fact that the surface slope of a stationary body of open water is close to zero in geopotential coordinates (i.e. using orthometric photon heights), in contrast to the surrounding ice sheet or ice shelf which mostly have slopes greater than $0.01°$ (Shen et al., 2022; Fan et al., 2022). This simple property enables a computationally inexpensive calculation (a "flatness check") to be applied to geoid-corrected ATL03 height data to check for possible candidate lake segments.

To perform this flatness check, we apply the ATL03-provided geoid correction to photon heights, and divide the data of each
of ICESat-2's six ground tracks into approximately $140\,\mathrm{m}$ along-track segments aligning with ICESat-2's major frames. For each major frame, we bin photon elevations in $0.01\,\mathrm{m}$ intervals and smooth the resulting histogram using a gaussian filter with a standard deviation of $0.05\,\mathrm{m}$, then normalize it by dividing by its largest value. If the smoothed histogram has a single peak, we record the elevation of this peak $h_{\mathrm{peak}}$ as the surface elevation at which a flat surface reflector would be located. In the case of a melt lake segment with a bathymetric return signal, it is possible that the return from the lakebed is stronger than the
surface return. Therefore, if the smoothed histogram has multiple peaks with prominence $> 0.1$, we choose $h_{\mathrm{peak}}$ from the two most prominent peaks, and set it to the elevation of the one located at a higher elevation.

The flatness check is based on ratios between photon densities $d_i$ that we calculate for various elevation bands around $h_{\mathrm{peak}}$ (Appendix A). As illustrated in the lower panels of Fig. 4, $d_0$ is the photon density within an elevation band of $\pm w_{\mathrm{peak}} = 0.1\,\mathrm{m}$ around the photon density peak. If a major frame contains the flat surface of a lake, then most of the surface signal photons
should be contained in this "lake surface elevation band", making $d_0$ significantly larger than the photon density in surrounding elevation bands (see Fig. 4, panel b). $d_1$ and $d_2$ are the photon densities within elevation bands of width $w_{\mathrm{buffer}} = 0.35\,\mathrm{m}$ just below and above the lake surface elevation band, respectively. Due to multiple scattering in the water column of a lake, we expect that over supraglacial lake segments the photon density just below the surface ($d_1$) can take on larger values than the photon density just above the surface ($d_2$). $d_3$ is the photon density within the entire telemetry window except for the lake
surface elevation band, and $d_4$ is the photon density between the top of the lake surface elevation band and the top of the telemetry window. Over a lake segment, most of the telemetry window outside the lake surface elevation band contains only background noise photons, so we expect that the photon densities $d_3$ and $d_4$ need to be significantly smaller than the surface photon density ($d_0$) if the major frame contains a flat lake surface. Since $d_3$ can still contain photons below the lake surface due to multiple scattering and a bathymetric signal, we expect that over supraglacial lake segments $d_3$ can take on larger values than

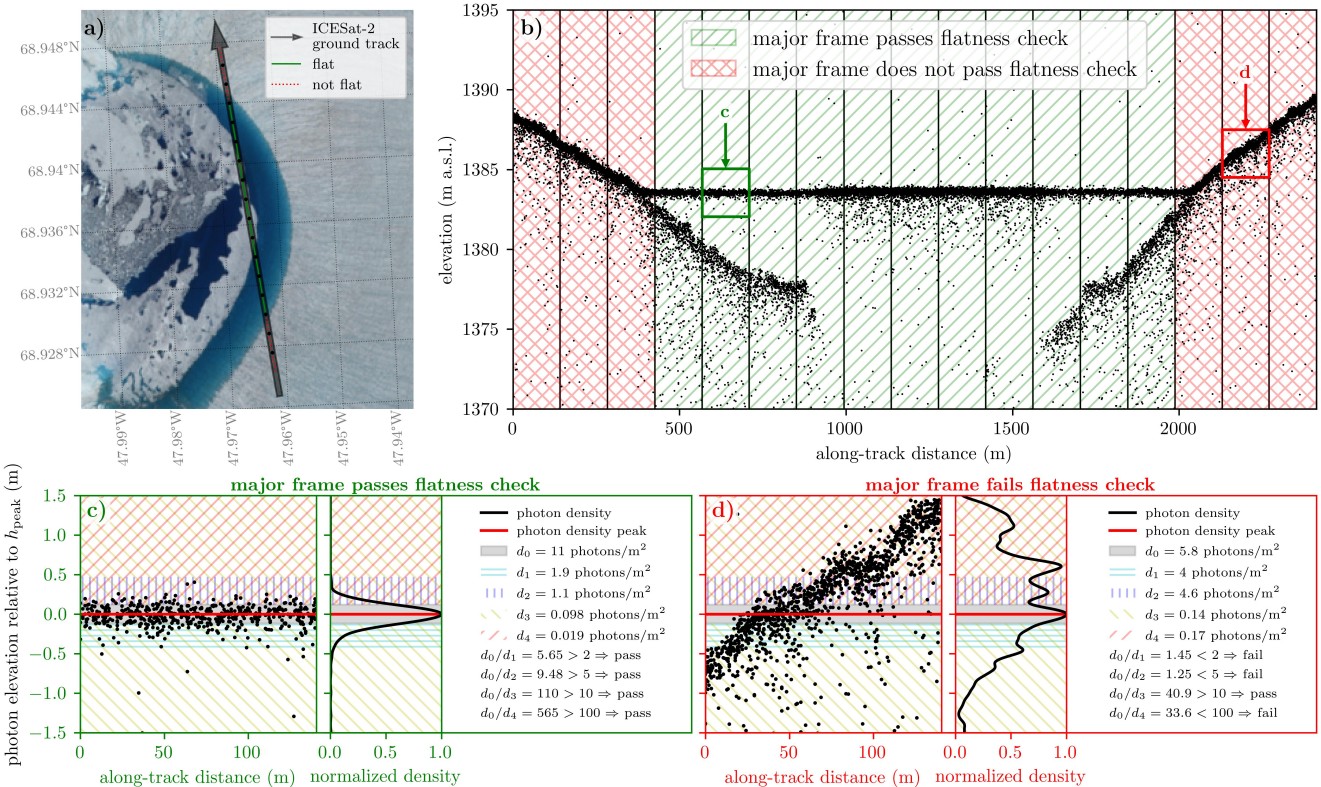

**Figure 4.** FLUID "flatness check" applied to every ATL03 major frame for identifying potential supraglacial lake segments. a) Ground track of an along-track segment of ATL03 data over the Greenland Ice Sheet, crossing a partially ice-covered supraglacial lake; b) Corresponding along-track photon elevations, with major frame boundaries marked by vertical black lines and flatness check outcomes shown in hatching; c) and d) Photon density ratios for a passing and a failing segment, respectively. (Data from ICESat-2 Track 216 GT1L on 12 July 2019 and centered at 68.9370° N, 47.9657° W; granule: ATL03_20190712052659_02160403_006_02.h5, imagery: Sentinel-2 on 13 July 2019)

$d_4$. Based on these assumptions, and using a trial-and-error approach, we defined the following thresholds on the density ratios that need to hold for a major frame to pass the flatness check: $d_0/d_1 \geq 2$, $d_0/d_2 \geq 5$, $d_0/d_3 \geq 10$ and $d_0/d_4 \geq 100$. As part of this trial-and-error approach, we manually assessed the effects of tweaking the above thresholds on a number of hand-picked granules, which we judged to be likely representative of various possible environments, to ensure adequate performance (i.e. granules without surface melt vs. pervasive surface melt, granules with smooth vs. rough background topography, granules containing ice-covered and partially ice-covered lakes, granules containing slush areas, granules containing exposed bedrock, partially cloudy granules, weak vs. strong beam data, night- vs. daytime acquisitions, etc.). Panels a and b of Fig. 4 illustrate the outcome of the flatness check for all the major frames within a short along-track segment of ATL03 data that crosses a partially ice-covered supraglacial lake. Panels b and c show examples of major frames that pass the flatness check and fail the flatness check, respectively, and illustrate the elevation bands that were used to calculate photon density ratios.

Since FLUID assesses the flatness of the surface of full major frames that cover an along-track distance of $\sim 140\,\text{m}$, lake segments with shorter open-water surfaces are not guaranteed to be detected by FLUID. However, lake segments that are significantly shorter than $140\,\text{m}$ are regularly detected by FLUID in practice. This is because the return signal from flat water surfaces is typically much stronger than the return signal from the surrounding ice surfaces, which makes even very short, flat water surfaces dominate the overall distribution of photon elevations within a major frame. The presence of a flat surface within a major frame is a necessary condition for detecting supraglacial bathymetry data, but it is not sufficient. There are many types of surfaces that would pass the flatness test but are not supraglacial lake segments with a bathymetric signal. This includes areas of slush, frozen-over supraglacial lakes covered in ice and snow, any areas of sea ice, ocean water or ice-marginal lakes erroneously included in the ice mask used for subsetting data, and other short along-track sections over firn or glacial ice that happen to be extremely flat by chance. For example, a lake may have partial ice cover, which prevents ICESat-2 from obtaining a bathymetric return (Fig. 4). However, since the ice cover here appears to be thin and flat, the corresponding major frames still pass the flatness check despite the absence of any useful bathymetry data in those segments. This means that the flatness check presented in this chapter serves as a preliminary screening method, helping to efficiently narrow down the number of along-track segments that could potentially contain useful supraglacial bathymetry data. This process makes it computationally feasible to determine whether a bathymetric signal is actually present by performing more complex operations on only the data that remain after checking for a sufficiently flat surface. The following sections describe these methods, which are at first only applied to those major frames that passed the initial flatness check.

### 3.2.2 FLUID step 2: removal of afterpulses

The second step removes artefacts in the ATL03 photon data known as "afterpulses", which appear as additional lines below and parallel to the primary surface return, due to the specifics of the ATLAS sensor (Luthke, 2023; Lu et al., 2021; Martino et al., 2022a). Afterpulses only become noticeable when the sensor is nearly or fully saturated, which means they often appear in ATL03 data over supraglacial lakes because smooth open water surfaces (i.e. the surface of stationary water bodies that are not affected by wind) can result in specular reflection. This suggests that the presence of wind ripples increases the likelihood of detecting a lake segment with a clear bathymetric signal in ATL03 data by preventing sensor saturation and afterpulsing (Lu et al., 2019; Tilling et al., 2020) and also explains why we observe afterpulsing more frequently near the (more wind-shielded) margins of melt lakes than over their (more wind-exposed) interior. Figure 5 (panels d-h) shows an example of an ATL03 data segment over a supraglacial lake in which these afterpulses are clearly visible below the flat water surface. There are three different mechanisms that can cause afterpulses:

1. *Dead-time* afterpulses appear in saturated pulses due to the ATLAS receiver channels only being able to register one photon event roughly every $3\,\text{ns}$. If the return signal is strong enough that all receiver channels register photon events during a time span shorter than this "dead-time", ATLAS cannot register any photons until the receiver channels have recovered. This means that for saturated pulses, afterpulses can appear in intervals of about $3\,\text{ns}$ of photon flight time, equivalent to $\sim 0.45\,\text{m}$ of elevation (Lu et al., 2021).

2. *Internal reflection* afterpulses are found in ATL03 data around $2.36\,\mathrm{m}$, $4.27\,\mathrm{m}$ and $6.59\,\mathrm{m}$ below the surface return (Martino et al., 2022a). These are due to optical reflections internal to the ATLAS receiver.

3. *PMT ionization* afterpulses appear as a broad peak $\sim 12$–$40\,\mathrm{m}$ below the surface when pulses are strongly saturated and cause ionization of the Photomultiplier Tubes (PMTs), which triggers false photon detection events.

Since all of these afterpulses present as secondary peaks in photon density below the primary surface return, they can be mistaken for or obscure any real bathymetric signal returns. Therefore, they need to be removed before determining whether a bathymetric signal is present in the data. ATL03 provides the parameter `quality_ph` that is designed to allow users to filter
out afterpulses. However, this parameter does not remove most dead-time afterpulses and naively removes all data more than $2\,\mathrm{m}$ below the surface for saturated returns (Neumann et al., 2022). This means that using the ATL03-provided `quality_ph` flag is not appropriate when searching for sub-surface return signals in saturated pulses, as it would fail to remove dead-time afterpulses that could be misidentified as bathymetric signals and could remove actual bathymetric signals at depths greater than $2\,\mathrm{m}$ (Fig. 5 panel e). Therefore, we developed an improved afterpulse removal routine that is tailored to bathymetric
applications.

**Afterpulse removal:** We first estimate the saturation level of each pulse, based on the sensor dead-time $t_{\mathrm{dead}}$ which is provided for each beam in the ATL03 data product. Let $n_{\mathrm{ch}}$ be the number of receiver channels, i.e. $n_{\mathrm{ch}} = 4$ for weak beams and $n_{\mathrm{ch}} = 16$ for strong beams. If the total number of photons in a pulse $n_{\mathrm{ph}} \geq n_{\mathrm{ch}}$ we can calculate the minimum vertical distance spanned by any $n_{\mathrm{ch}}$ photons, and denote it by $\Delta h$. We then estimate the sensor saturation ratio as $r_{\mathrm{sat}} = t_{\mathrm{dead}} c / (2 \Delta h)$
if $n_{\mathrm{ph}} \geq n_{\mathrm{ch}}$ and zero otherwise, where $c$ is the speed of light in a vacuum. This means that for saturated pulses ($r_{\mathrm{sat}} \geq 1$), all receiver channels registered a photon within a time frame of $t_{\mathrm{dead}}/r_{\mathrm{sat}}$. For all saturated pulses, we calculate the elevation of the saturated return, $h_{\mathrm{sat}}$ as the mean elevation of the $n_{\mathrm{ch}}$ photons that span $\Delta h$.

To determine the typical locations of afterpulses relative to $h_{\mathrm{sat}}$ in saturated pulses, we compiled a data set of saturated pulses from melt lake segments using an earlier version of FLUID (Arndt and Fricker, 2022), which did not include afterpulse removal.
For each saturated pulse, we subtracted $h_{\mathrm{sat}}$ from the photon elevations and created a histogram of photon counts weighted by $r_{\mathrm{sat}}$ (Fig. 5, panels a-c). The strong peaks in this histogram correspond to the elevations at which afterpulses occur relative to the surface. We found that dead-time afterpulses occurred at four depths: $AP_1^{(\mathrm{dead})} = 0.55\,\mathrm{m}$, $AP_2^{(\mathrm{dead})} = 0.92\,\mathrm{m}$, $AP_3^{(\mathrm{dead})} = 1.50\,\mathrm{m}$ and $AP_4^{(\mathrm{dead})} = 1.85\,\mathrm{m}$. Only the first two of the internal reflection afterpulses were strong enough to significantly contaminate bathymetric data in saturated or near-saturated pulses, at $AP_1^{(\mathrm{ir})} = 2.46\,\mathrm{m}$ and $AP_2^{(\mathrm{ir})} = 4.25\,\mathrm{m}$ below the surface. While the
third internal reflection afterpulse is also visible at $AP_3^{(\mathrm{ir})} = 6.52\,\mathrm{m}$, it appears that this afterpulse is not typically strong enough to be confused for a bathymetric return signal. The broad peak associated with PMT ionization around $AP^{(\mathrm{ion})} \approx 29 \pm 15\,\mathrm{m}$ only became noticeable at the typical length scales of ICESat-2 melt lake segments when $r_{\mathrm{sat}} > 3.5$. For such strongly saturated pulses, we simply discarded any photons $> 12\,\mathrm{m}$ below the surface.

With the locations of the main afterpulses known, we can use a simple empirical method to remove afterpulses from ATL03
data for all major frames that passed the flatness check and for which at least 100 photons are attributed to saturated pulses. For each major frame, we follow the above weighted histogramming procedure and examine the heights of the seven most

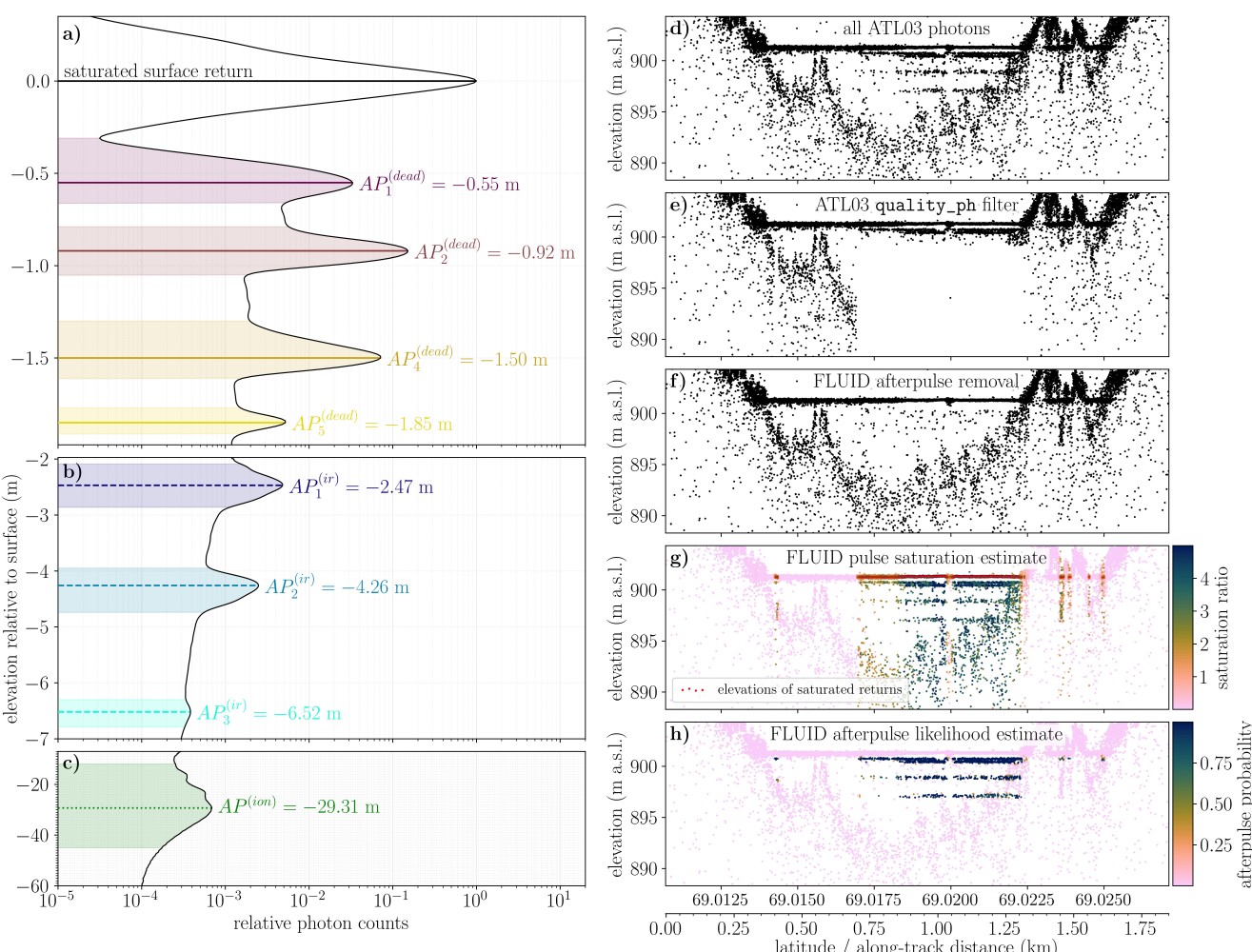

**Figure 5.** FLUID afterpulse removal. (a) to (c): Histogram of photon elevations in saturated pulses from ICESat-2 melt lake segments, relative to the elevation at which saturation occurred. The secondary peaks that appear below the saturated surface return are afterpulses that are caused by (a) dead-time of the ATLAS sensor; (b) internal reflections in the instrument; and (c) PMT ionization. Note that (a), (b) and (c) have different vertical scales. d) to h): Implementation of FLUID's afterpulse removal for a short along-track segment of ATL03 data that crosses a supraglacial lake, with sections of highly saturated (specular) pulses. Known locations of ATLAS afterpulses (a) to (c) are used to remove likely afterpulse photons. (Data from ICESat-2 Track 1222 GT2L on 17 June 2019 and centered at 69.0189° N, 49.0444° W; granule: ATL03_20190617064249_12220303_006_02.h5)

prominent peaks; if any of these peaks align with the relative elevations of the known afterpulses, we consider it evidence for likely afterpulsing and remove any photons that belong to saturated pulses in that elevation band (Fig. 5, panel h). Since this procedure removes photons in saturated pulses only, true bathymetric signals that overlap with the elevation of a known

afterpulse are still retained as long as they appear in any unsaturated pulses. However, if all pulses within an along-track section

of the data are saturated, any true bathymetric signals from a flat lakebed at the elevation of a known afterpulse will be removed from the data because they are practically indistinguishable from the afterpulses that we expect to see in the point cloud under such highly saturated conditions.

While more sophisticated approaches for afterpulse removal are certainly possible, we found that in practice our purely empirical approach strikes a good balance between effectively removing enough afterpulse photons to prevent bathymetric surface fitting methods from considering afterpulses a signal, while also retaining enough photons to prevent removal of actual signal returns whenever it is possible to discern the two.

### 3.2.3   FLUID step 3: photon signal confidence estimation

Once flat "candidate" segments have been identified and afterpulses have been removed, the next step is to assign a signal confidence score to the remaining photons. ATL03 contains many noise photons from various sources, such as solar background noise, atmospheric backscatter, and multiple scattering in translucent media (Neumann et al., 2019; Yang et al., 2023). Release 006 of the ATL03 data product provides two measures that can help discriminate between signal and noise photons. Over the ice sheets, the `signal_conf_ph` parameter gives an estimate of how likely it is that a photon is part of the land ice surface signal, based on slant histogramming (Neumann et al., 2019). This parameter, however, does not consider the possibility of two distinct reflective surfaces that are both signal, and therefore often labels lakebed return photons over supraglacial lakes as noise (Fig. 6, panel a). Since release 006, ATL03 also includes the `weight_ph` parameter, which provides a local metric for relative photon density based on the Yet Another Photon Classifier method (YAPC; Neumann et al., 2022; Sutterley and Gibbons., 2021). For each target photon, the YAPC weight calculation is based on a rectangular window $\pm 3\,\mathrm{m}$ in elevation around the photon location. This can result in sharp photon weight discontinuities $3\,\mathrm{m}$ above and below highly reflective flat surfaces, which are inconsistent with relative local photon density (Fig. 6, panel b). Due to these drawbacks of the ATL03-provided parameters, we developed a new density-based method for photon signal confidence estimation that is more accurate for ICESat-2 melt lake segments (Fig. 6, panel c). This method is based on the inverse euclidean distances between a photon and its $k$-nearest neighbors within a search radius that depends on the background noise rate (Appendix B).

In FLUID, we implement this photon signal probability estimation using a KD-tree approach for querying nearest neighbors of photons, applied to individual major frames. To calculate photon densities within a major frame, we consider additional photons within a sufficiently wide buffer in along-track distance to avoid penalizing photons that are near the major frame margins by not taking into account all their nearest neighbors. Panel c) of Fig. 6 shows the resulting density-based photon signal probabilities for a supraglacial melt lake segment on Amery Ice Shelf.

### 3.2.4   FLUID step 4: secondary bathymetry peak detection

To determine which major frames amongst the ones that pass the flatness check are likely to provide useful bathymetry data, FLUID checks for secondary peaks in photon density below the flat surface return. To do so, we divide each major frame into 10 along-track sub-segments of equal length (i.e. about $14\,\mathrm{m}$ per sub-segment). For each sub-segment we use the FLUID photon-level signal probabilities to calculate photon signal confidence as an empirical smooth function of elevation. To determine

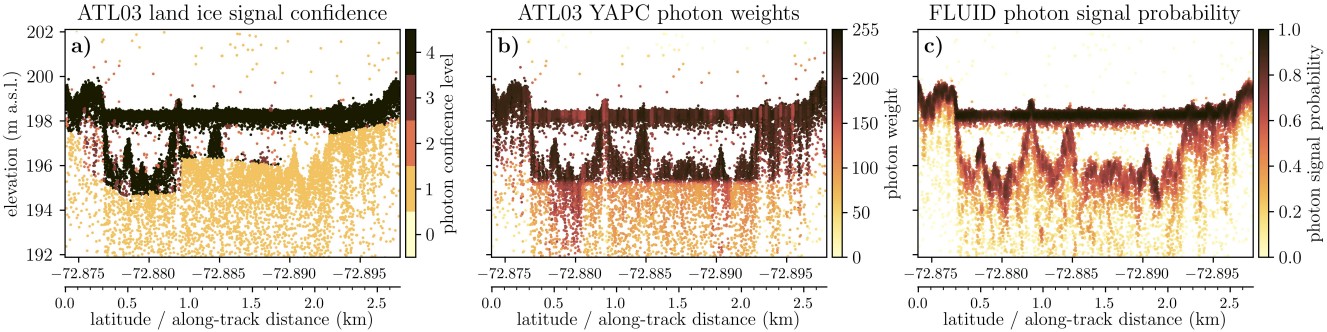

**Figure 6.** A comparison between existing ATL03 photon signal confidence estimates and our method for melt lake segments in FLUID. (Data from ICESat-2 Track 81 GT2L on 2 January 2019 and centered at 72.8859° S, 67.3082° E; granule: ATL03_20190102184312_00810210_006_02.h5)

whether a potential bathymetric signal is present below the lake surface, we determine the elevation of the most prominent below-surface peak in this function for every sub-segment. Based on the along-track locations, elevations and prominences of all peaks that were identified in a given major frame, we define four quality heuristics $q_i$ for different components that we found to affect the overall quality of the bathymetric return (Appendix C). The $q_i$ all take on values between zero and one, with higher values implying a "better" bathymetric signal. We designed the expressions for the quality heuristics such that $q_1$ penalizes major frames with smaller numbers of detected subsurface peaks, $q_2$ penalizes major frames with less prominent peaks, $q_3$ penalizes major frames with a very large overall spread of peak elevations, and $q_4$ penalizes major frames with peak elevations that do not align along a smooth surface. We then define the overall bathymetric quality summary $q_s$ of a major frame as the product of the four $q_i$. We consider the secondary bathymetric peak in photon density strong and coherent enough to pass the bathymetric signal check for any major frames with $q_s \geq 0.1$.

We illustrate this procedure for an along-track segment over Central West Greenland that crosses a supraglacial lake with a bathymetric return signal that varies in strength along the ground track (Fig. 7). In this example, most of the major frames that cover the supraglacial lake's interior pass the bathymetric signal check, with bathymetric photon density peaks smoothly tracing the apparent lakebed. However, two of the major frames within the lake's interior do not have a strong-enough signal of photons reflected from the lakebed to be passing the bathymetric signal check during this step, in which FLUID initially considers each major frame in isolation. These two major frames visibly overlap with the location of a thin partial ice cover near the lake's northern shore (Fig. 7, panel a), which explains why some of the lakebed is occluded. While such areas, where part of the lakebed is occluded, may not pass the bathymetry check, they are later included in the data that make up a full ICESat-2 lake segment, as explained in the next section.

### 3.2.5 FLUID step 5: along-track aggregation of lake segments

Given the collection of major frames that individually pass the bathymetric signal check along a ground track, FLUID aggregates major frames into clusters, each of which likely represents a transect of an entire supraglacial lake. To achieve this,

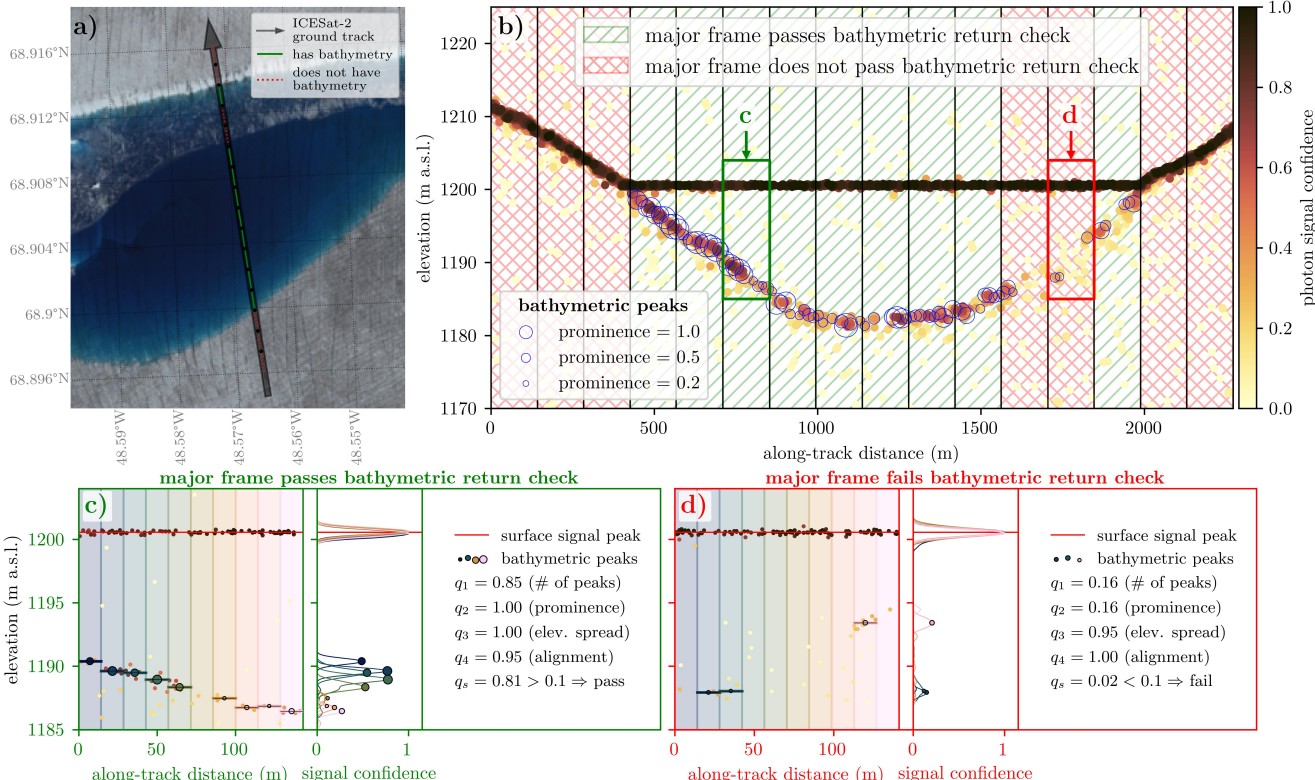

**Figure 7.** FLUID's bathymetric signal check run on every ATL03 major frame that has passed the flatness check. a) Ground track for an along-track segment of ATL03 data over the GrIS, which crosses a supraglacial lake that has a thin partial ice cover near its northern shore. b) Corresponding along-track photon elevations, and locations of detected bathymetric photon density peaks. The vertical black lines are the major frame boundaries, and hatching indicates whether major frames passed the bathymetric signal check or not. c) and d) Bathymetric peak-finding procedure from photon density and associated values of the bathymetric return quality heuristics described in the text and defined in Appendix C, for a passing and a failing major frame, respectively. (Data from ICESat-2 Track 277 GT3R on 16 July 2019 and centered at 68.9062° N, 48.5689° W; granule: ATL03_20190716051841_02770403_006_02.h5, imagery: Sentinel-2 on 16 July 2019)

we use an agglomerative clustering scheme based on two simple assumptions: (i) the water surface elevation within a single ICESat-2 lake segment should be nearly constant along the ground track, and (ii) a ground track rarely crosses the same lake in two distinct locations that are separated by more than about $1.5\,\mathrm{km}$. At the start of the clustering process, each major frame that passed the initial bathymetric return check is considered a singleton cluster with a water surface elevation $h_{\mathrm{surf}}$ equal to the single major frame's surface photon density peak $h_{\mathrm{peak}}$, and major frame start and end IDs $m_{\mathrm{start}} = m_{\mathrm{end}}$ that are both equal to the single major frame's ID. This means that a cluster can be expressed as

$$\mathcal{C}^{(i)} = \left\{ h_{\mathrm{surf}}^{(i)}, m_{\mathrm{start}}^{(i)}, m_{\mathrm{end}}^{(i)} \right\} \tag{1}$$

where the index $i \in 1, 2, \ldots, n_{\mathrm{clusters}}$ is assigned to the $i^{\mathrm{th}}$ cluster when sorting all $n_{\mathrm{clusters}}$ clusters by their respective values of $m_{\mathrm{start}}$. Since major frame IDs are numbers that strictly increase with along-track distance, this means that $m_{\mathrm{end}}^{(i)} < m_{\mathrm{start}}^{(i+1)}$ for all clusters. Now, clusters that are adjacent to each other in along-track coordinates are compared in a pairwise fashion. For all uneven numbers $i < n_{\mathrm{clusters}}$, if

$$\left| h_{\mathrm{surf}}^{(i)} - h_{\mathrm{surf}}^{(i+1)} \right| \leq \Delta h_{\mathrm{max}} = 0.1\,\mathrm{m} \tag{2}$$

and

$$m_{\mathrm{start}}^{(i+1)} - m_{\mathrm{end}}^{(i)} \leq \Delta m_{\mathrm{max}} = 10, \tag{3}$$

clusters $\mathcal{C}^{(i)}$ and $\mathcal{C}^{(i+1)}$ are merged into a new cluster

$$\mathcal{C}^{(i')} = \left\{ \left( h_{\mathrm{surf}}^{(i)} + h_{\mathrm{surf}}^{(i+1)} \right) / 2,\ m_{\mathrm{start}}^{(i)},\ m_{\mathrm{end}}^{(i+1)} \right\}. \tag{4}$$

Equation 2 states that neighboring clusters are only merged if their respective lake surface elevations are within $0.1\,\mathrm{m}$ of each other, and Eq. 3 states that neighboring clusters are further only merged if they are separated by ten major frames that did not pass the bathymetry check, or less (about $1.5\,\mathrm{km}$). This means that if FLUID encounters the unlikely but possible scenario in which a ground track crosses two arms of the same lake, which are separated in along-track distance by more than 10 major frames, then these two crossings are considered to be separate lake segments and returned as two separate files in the output data rather than being merged together into one lake segment. If these two conditions do not result in any two clusters being merged, then the same pairwise comparison is carried out for all even numbers $i < n_{\mathrm{clusters}}$. After an iteration of merging clusters, the indices of the remaining $n'_{\mathrm{clusters}}$ clusters are re-set to $1, 2, \ldots, n'_{\mathrm{clusters}}$, and the same procedure is repeated until no more clusters can be merged based on the conditions above.

The resulting final clustering is now considered the set of ICESat-2 supraglacial lake segments that have been found on each ground track. Note that for simplicity we here use the term *"ICESat-2 lake segment"* (or simply *"lake segment"*) to refer to any single-ground-track segment of ATL03 data with visible bathymetry from one supraglacial lake. If multiple ICESat-2 ground tracks contain data from the same supraglacial lake, the distinct ground track segments are still considered different "ICESat-2 lake segments" for the purpose of this algorithm. For example, the two ATL03 profiles acquired by the two neighboring ground tracks of the center beam pair shown in Fig 1 would be considered two distinct "lake segments" despite ICESat-2 having acquired their underlying data during the same overpass and from the same supraglacial lake. Since multiple ICESat-2 lake segments can be associated with the same supraglacial lake, the total number of unique supraglacial lakes sampled by ICESat-2 is smaller than the total number of supraglacial lake segments reported by FLUID-SuRRF (Sect. 4.1, Table 1).

Since every lake segment that was detected this way is characterized by an along-track range of major frame IDs $[m_{\mathrm{start}},\ m_{\mathrm{end}}]$, FLUID extends these ranges outwards to make sure that no bathymetry data were missed near the edges of any lake segment. To do so, each lake segment's range is extended to include any major frames for which $|h_{\mathrm{peak}} - h_{\mathrm{surf}}| < 0.2\,\mathrm{m}$ as long as such major frames exist within three major frames of the lake segment's range. At the end of this process we add another four major frames as a buffer, two to each side of the lake segment. Since this expansion of the along-track ranges of lake segments can

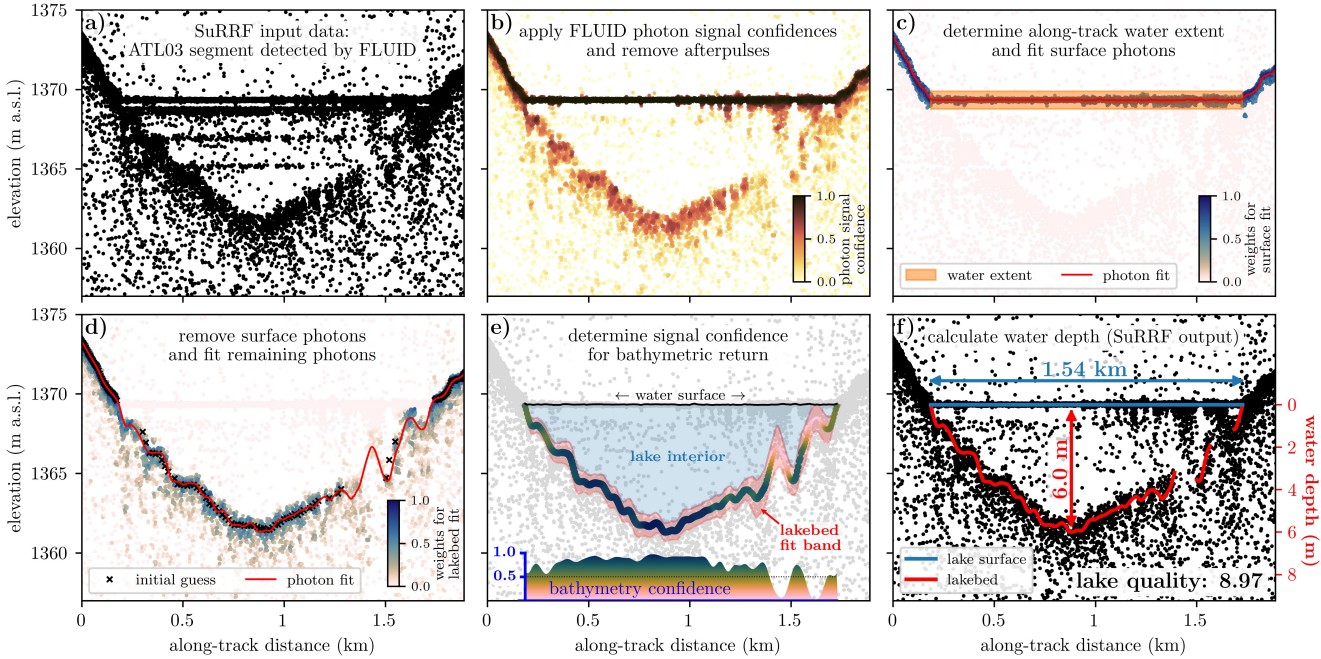

**Figure 8.** SuRRF algorithm for determining supraglacial lake depth from an along-track segment of ATL03 data that was detected by FLUID. (Data from ICESat-2 Track 277 GT2L on 13 July 2020 and centered at 68.1923° N, 48.5134° W; granule: ATL03_20200713115804_02770803_006_01.h5

create lake segments that overlap, the set of buffered lake segments is corrected by separating partially overlapping lake segments at the midpoint of their along-track overlap and removing any lake segments that are fully contained within another lake segment. We apply FLUID steps 2, 3 and 4 (afterpulse removal, signal confidence estimation and bathymetric return check) to all major frames in the buffer (i.e. now included in the along-track range of a lake segment but had not initially passed the flatness check). The resulting final set of lake segments across all ground tracks in the input granule is the output of FLUID.

## 3.3 Supraglacial lake depth determination: the SuRRF algorithm

To estimate the along-track depth for each detected lake segment, we developed the Surface Removal and Robust Fit (SuRRF) algorithm, updated from Fricker et al. (2021). The central idea of SuRRF is to use a robust fitting procedure (Sect. 3.3.1) to first fit a smooth line to all photons returned from the lake surface and the surrounding topography (Sect. 3.3.2), then remove all photons that are part of the water surface and fit another smooth line to the remaining photons to determine the location of the lakebed (Sect. 3.3.3). The along-track water depth estimate is the elevation difference between the fits to the water surface and the lakebed, corrected for the refractive index for the speed of light in water (Sect. 3.3.5). In Fig. 8, we illustrate the main steps of SuRRF using an example of ATL03 along track data, which was determined to be a supraglacial lake segment by FLUID. We summarize the main steps of SuRRF in the following sections.

### 3.3.1 SuRRF Robust Fit

In SuRRF steps 1 and 2 (Sect. 3.3.2 and 3.3.3), we use a tailored robust nonparametric regression method to locate the lake surface and lakebed. The SuRRF Robust Fit is based on Locally Weighted Regression and Smoothing Scatterplots (LOWESS, Cleveland, 1979), which is applied to the data iteratively, while removing outliers during each iteration to converge to an along-track fit that smoothly tracks the elevation of highest photon density. For each evaluation location $x_{\text{fit}}$ in along-track coordinates, we fit a locally weighted $n^{\text{th}}$-degree polynomial regression to the photons that are at a distance of at most $x_{\text{max}}$ in along-track coordinates. The value of $x_{\text{max}}$ is the minimum distance from $x_{\text{fit}}$ within which there are $n_{\text{ph}}$ photons $i$ with a nonzero FLUID-derived signal confidence $p_i$, or a minimum along-track fitting window length of $x_{\text{min}}$, whichever is larger. To achieve smooth local weighting, the along-track weight of a photon $i$ at location $x_i$ is calculated using the tri-cube weight function $w_i^{(\text{x})} = \left(1 - |(x_i - x_{\text{fit}})/x_{\text{max}}|^3\right)^3$. If the method is provided with an initial guess for the first iteration, we calculate the residuals $e_i$ as the difference between each photon's elevation and the linear interpolation of the initial guess elevations to each photon's along-track location. In this case, we consider only photons whose absolute residuals are at most $h_{\text{max}}$, and calculate their residual-based weights as $w_i^{(\text{h})} = \left(1 - (|e_i|/h_{\text{max}})^3\right)^3$. In absence of an initial guess, we set $w_i^{(\text{h})} = 1$ for all photons in the first iteration. The photon weights that are used for the regression are $w_i = p_i w_i^{(\text{x})} w_i^{(\text{h})}$. We evaluate the resulting regression model at each fit location $x_{\text{fit}}$ to obtain an along-track estimate of the fit to the photon heights. For each consecutive iteration, we calculate the residuals $e_i$ as the difference between each photon's elevation and the elevation of the linearly interpolated elevation fit of the previous iteration. Let $\sigma$ be the standard deviation of residuals, weighted by the previous iteration's weights. To achieve a robust fit, we now consider only photons whose absolute residuals are at most $n_{\text{std}}$ standard deviations, and calculate the residual-based weights $w_i^{(\text{h})}$ as defined above, using $h_{\text{max}} = n_{\text{std}}\sigma$. We run this nonparametric weighted regression for a number of $n_{\text{iter}}$ iterations to obtain a smooth fit that tracks the along-track elevation of highest photon density.

### 3.3.2 SuRRF step 1: lake surface fit

To fit an elevation profile to just the surface of a lake segment detected with FLUID, we calculate the along-track extent of photons that belong to the flat lake surface, remove all photons below the lake surface, and then apply the SuRRF Robust Fit to the remaining photons (Fig. 8, panel c). We determine the extent of the open water surface by calculating the photon density within an elevation band of $\pm 0.225\,\text{m}$ around the lake's surface elevation at $1\,\text{m}$ along-track resolution, as well as the corresponding photon densities within the remaining telemetry window and within $2\,\text{m}$ above the surface elevation band. We smooth all photon densities using an along-track gaussian window with a $15\,\text{m}$ standard deviation. We then consider a location along the ground track to contain a water surface if the photon density within the surface elevation band is at least ten times as large as the other photon densities for any continuous along-track section of at least $100\,\text{m}$ in length. Within the resulting estimate of along-track water extent, we remove all photons that are more than $0.4\,\text{m}$ below the lake's surface elevation from the surface fit by setting their signal confidence $p_i = 0$.

To obtain an along-track fit to the lake's surface and its surrounding topography, we apply the SuRRF Robust Fit (Sect. 3.3.1) to all remaining photons with a signal confidence $> 0.5$ at evenly spaced locations $x_{\text{fit}}$ along the ground track, at $5\,\text{m}$ resolution.

In this step, we use a linear regression ($n = 1$) and run it for $n_{\text{iter}} = 10$ iterations, with no initial guess. We here choose $x_{\text{min}} = 20\,\text{m}$ and let $n_{\text{ph}}$ decrease linearly from $n_{\text{ph}}^{(\text{start})} = 300$ in the first iteration to $n_{\text{ph}}^{(\text{end})} = 100$ in the last iteration. Similarly, we choose $n_{\text{std}}^{(\text{start})} = 10$ and $n_{\text{std}}^{(\text{end})} = 4$. We illustrate this surface-fitting procedure by showing an example of a supraglacial lake

segment in ATL03 with photons color-coded by their surface fit weights $p_i$, along with the detected along-track water surface extent and the final smooth photon fit to the lake surface and surrounding topography (Fig. 8, panel c).

### 3.3.3   SuRRF step 2: lakebed fit

To fit an elevation profile to just the lakebed, we remove all photons that belong to the lake's water surface, and then again apply the SuRRF Robust Fit to the remaining photons (Fig. 8, panel d). In this case, we remove all photons that fall within

the along-track water surface extent that was determined in the previous step, and are located at an elevation of $0.35\,\text{m}$ below the lake's surface or higher. Note that this imposes a theoretical minimum depth threshold for detection on lake segments: ATL03 segments need to exhibit a bottom return signal at least $0.35\,\text{m}$ below the lake surface (or $0.26\,\text{m}$ in refraction-corrected water depth) at their deepest along-track point to be considered by SuRRF. However, in practice, such shallow lake segments do not have a discernible bathymetric signal since typical depth retrieval accuracies for ICESat-2 are on the order of $0.5\,\text{m}$

(Dietrich et al., 2024). To provide the SuRRF Robust Fit with an initial guess, we combine the locations of the bathymetric peaks found by FLUID that have a peak prominence value of at least $0.5$ and fall within the lake's along-track water extent with those locations of the smooth surface fit from Sect. 3.3.2 that fall outside of the lake's along-track water extent. We then smooth the values of the initial guess using a running mean with a window of five data points to decrease the influence of any potential outliers. To decrease the influence of near-surface photons from multiple scattering, we further reduce the signal

confidence values $p_i$ of any photons that are between a lower bound of one meter above the initial guess and an upper bound of the lake surface elevation by multiplying them by a factor that linearly decreases from one to zero between the lower and the upper bound. To fit the lakebed, we apply the SuRRF Robust Fit (Sect. 3.3.1) to the same evaluation locations $x_{\text{fit}}$ that were used to fit the lake surface. We here use a third-degree polynomial regression ($n = 3$) and SuRRF Robust Fit parameters $n_{\text{iter}} = 20$, $x_{\text{min}} = 100\,\text{m}$, $n_{\text{std}}^{(\text{start})} = 10$, $n_{\text{std}}^{(\text{end})} = 3$, and $h_{\text{max}} = 10\,\text{m}$ in the first iteration. In this step we choose different values

for $\left( n_{\text{ph}}^{(\text{start})}, n_{\text{ph}}^{(\text{end})} \right)$ depending on ATLAS beam strength: $(200,\ 100)$ for strong beam data and $(100,\ 50)$ for weak beam data. We illustrate this lakebed-fitting procedure by showing photons color-coded by their lakebed fit weights $p_i$, along with the initial guess and the final smooth photon fit to the lakebed (Fig. 8, panel d).

Previous studies have hypothesized that ICESat-2-based depth retrieval algorithms placing the lakebed fit at the along-track elevation of highest subsurface photon density may be biased towards slightly overestimating total water depths due to multiple

scattering within the water column (Fricker et al., 2021; Xiao et al., 2023). To address this, we provide an optional correction, which places the lakebed fit at a higher elevation where the initial SuRRF lakebed fit included photons further below the initial lakebed fit than would be expected from bathymetric signal photons. To achieve this, we remove any photons located at a vertical distance below the initial SuRRF lakebed fit by more than the sum of (1) ICESat-2's single-photon time-of-flight precision ($\sim 12\,\text{cm}$ in ATL03 photon heights or $800\,\text{ps}$; Markus et al., 2017), and (2) the elevation range within ICESat-

2's footprint diameter ($\sim 11\,\text{m}$ Magruder et al., 2021a) obtained by projecting the footprint onto the along-track lakebed

topography estimated by the initial SuRRF lakebed fit. We then reapply the lakebed fit to the remaining photons as described above, while supplying the SuRRF Robust Fit (Sect. 3.3.1) with the uncorrected SuRRF lakebed fit as the initial guess. Since the presence or magnitude of this hypothesized overestimation of water depths cannot be established without any ground truth *in situ* data available along any ICESat-2 lake segments, we provide this scattering correction for reference only, and do not apply it to the water depths presented in this study. If such validation data becomes available in the future, our scattering correction can be tuned to better match observations, and can be readily applied to FLUID-SuRRF output data.

### 3.3.4 SuRRF step 3: bathymetry signal confidence estimation

To estimate the signal confidence of the fit to the lakebed, we calculate the photon density ratio between the lower half of the interior of the lake and the elevation band within $\pm n_{\text{std}}^{(\text{end})}\sigma$ of the last iteration of the lakebed fit for each fit location $x_{\text{fit}} \pm 5\,\text{m}$ along the ground track. Here, we consider the interior of the lake to be the elevation range between the top of the elevation band of the lakebed fit and the surface elevation of the lake. For any along-track points for which there are no lakebed photons or for which the elevation band of the lakebed fit includes the lake surface, we set the ratio to 1. We then set the bathymetry confidence to one minus the density ratio, clip it to the range $[0, 1]$, set it equal to one wherever the lakebed fit is at a higher elevation than the lake surface elevation (i.e. wherever the estimated water depth is zero), and smooth it using an along track gaussian filter with a standard deviation of $10\,\text{m}$. Wherever the elevation range of the interior of the lake is less than the width of the elevation band of the lakebed fit, we further decrease the confidence by multiplying it by the ratio between the two elevation ranges. We illustrate this bathymetry signal confidence estimation procedure by visualizing both the elevation band of the final lakebed fit and the interior of the lake, and showing the resulting along-track confidence estimates for the bathymetric return (Fig. 8, panel e).

### 3.3.5 SuRRF step 4: water depth calculation

To determine the along-track water depth, we take the difference between the lake's surface elevation and the fit to the lakebed, and divide it by the refractive index for the speed of $532\,\text{nm}$ light in $0\,°\text{C}$ freshwater ($\approx 1.336$; Mobley, 1995). For any locations along the lake segment where the final lakebed fit (Sect. 3.3.3) returns a higher elevation than the surface elevation of the lake, we record a water depth of zero meters (i.e. no water is present). We do not correct water depths for the effect of lakebed return geolocation errors caused by refraction, since ICESat-2 is nadir-pointing to its reference ground tracks over land ice, making the water depth correction due to the angle of refraction negligibly small ($\approx 0.003$ of the total water depth for the slightly off-nadir-pointing outer beam pairs, which is about $9\,\text{cm}$ for a water depth of $30\,\text{m}$; Parrish et al., 2019). Final along-track water depths can be selected by applying a threshold to the bathymetry signal confidence (Sect. 3.3.4). Here, we select a confidence threshold of 0.5. For the lake segment example shown, this results in a maximum along-track depth of $6.0\,\text{m}$ and gaps in along-track depth data in locations where no bathymetric return is evident (Fig. 8, panel f).

### 3.3.6 SuRRF step 5: lake segment quality estimate

To provide a relative indication of data quality, we provide an estimate for a summarized quality measure for each lake segment. Let $h_x^{(\text{surf})}$ and $h_x^{(\text{bed})}$ be the surface and bed fits at along-track measurement location $x$. For all $x$ where $\Delta h_x = h_x^{(\text{surf})} - h_x^{(\text{bed})} > 0$, we calculate a histogram of photon counts within a $5\,\text{m}$ along-track window for 300 elevation bins that are evenly spaced between $h_x^{(\text{bed})} - \Delta h_x$ and $h_x^{(\text{surf})} + \Delta h_x$. We then normalize the associated bin elevations $\tilde{h}$ such that $h_x^{(\text{bed})}$ corresponds to $\tilde{h} = 0$ and $h_x^{(\text{surf})}$ corresponds to $\tilde{h} = 1$, and take the per-bin sum across all $x$. We smooth the resulting elevation-normalized histogram using a gaussian filter with a standard deviation of three bins, and calculate the quality ratio $r_q$ as the ratio between the value at $\tilde{h} = 0$ and the mean of the first quartile of the lowest values in $0 > \tilde{h} > 1$.

The "quality ratio" can be considered an along-track average estimate for the photon density ratio between the lakebed and the lowest-photon-density part of the interior of the lake. We classify lake segments with $r_q \leq 2$ as "zero quality" lake segments. Similarly, we classify lake segments with $r_q > 2$ as "high quality" lake segments, for which we report the lake segment quality score as $r_q - 2$. This means that a lake segment is assigned a non-zero quality by SuRRF if the along-track averaged strength of the return signal from the lakebed is at least twice as large as the along-track averaged background noise rate within the interior of the lake. While "zero quality" lake segments might still include a clear bathymetric return along a small part of their associated along-track extent, a quality score of zero is meant to indicate that there may be significant issues with data quality. In the example shown, this results in SuRRF classifying the given lake segment as high quality, with a score of 9.0 (Fig. 8, panel f). We show more examples of FLUID/SuRRF output lake segments and their associated quality scores in a range from 0.2 to 115.5 in Figs. 9 and 10 (Sect. 4.1).

### 3.4 Computational implementation of FLUID/SuRRF

To facilitate large-scale use of FLUID/SuRRF, we implemented the algorithms as a Python routine that can be run on any ATL03 data granule, and developed a framework that allows for estimating all ICESat-2 lake depths for a given region of interest and time span. Given a polygonal region of interest (e.g.; a particular glacier, ice shelf, drainage basin, or other study region) and time span (e.g.; a typical melt season), we use the National Snow and Ice Data Center's Data Access and Service API (NSIDC API) (NSIDC, 2021) to query for a list of all available ATL03 data granules that satisfy the spatio-temporal search parameters. To obtain all desired ICESat-2 lake depths, we can subset these ATL03 granules to the region of interest, run FLUID/SuRRF on each subsetted granule individually, and collect all output melt lake segment and their associated along-track depth data. This allows for parallel processing of data granules.

To apply FLUID/SuRRF to all identified granules in an efficient, cost-effective and reproducible manner, we use the OSG Open Science Pool for distributed High-Throughput Computing (dHTC) (OSG, 2006; Pordes et al., 2007) Since batches of OSG compute jobs run on heterogeneous hardware, we run all jobs in a Singularity container (Kurtzer et al., 2017) that we designed for use with FLUID/SuRRF. We run one OSG compute job per ATL03 granule, where each job receives as input the producer ID of the granule and a shapefile of the corresponding region of interest. Each job makes a request to the NSIDC API to subset the specified granule to the given shapefile, and downloads the subsetted granule. The job then runs FLUID/SuRRF

on the downloaded granule for each of ICESat-2's six ground tracks, and sends back individual HDF5 files of output water depths for each lake segment that was detected by FLUID.

Each output file reports water depth estimates at a $5\,m$ along-track resolution with associated values for longitude, latitude, along-track distance, bathymetry signal confidence and elevations of the lakebed and surface fit to the photon data. We also include lake segment properties such as: surface elevation, SuRRF quality score, and various metadata such as the granule name, beam, time of data acquisition, center longitude/latitude. For reference, we add the underlying ATL03 photon heights and locations with FLUID estimates of photon signal probability, saturation level and afterpulse probability, as well as calculated FLUID parameters at the major frame rate. In addition to each lake segment's data file, we also create an associated "quick look" plot of the photon data with surface and lakebed fits and the ground track shown over the closest available cloud-free Landsat 8/9 or Sentinel-2 imagery (e.g.; Fig. 9, a-j and 10, a-j). The availability of these for all returned lake segments makes it possible to add a final manual quality control step to our method, based on visual inspection of the plots in a custom-made streamlit app. We use this to remove clear false positives from the output data.

## 3.5 Study regions and time span

To evaluate the performance of our method, specifically whether it is able to capture spatial and temporal variability while reliably extracting supraglacial lake depths at scale, we focus on one drainage basin on each of the ice sheets and compare a high-melt with a low-melt season for each. For both the Greenland and Antarctic ice sheets, we define our study regions using the Ice sheet Mass Balance Inter-comparison Exercise (IMBIE) drainage basins (Fig. 2, Mouginot et al., 2017; Mouginot and Rignot, 2019). Since we do not expect significant surface meltwater pooling beyond a certain elevation, we apply elevation thresholds to the drainage basins prior to running FLUID/SuRRF.

In Greenland, we focus on the Central West drainage basin (CW, Fig. 2), and compare ICESat-2 lake depths between the exceptionally warm 2019 melt season (Tedesco and Fettweis, 2020) and the 2020 melt season, which experienced comparatively little surface melt and runoff (Druckenmiller et al., 2021). During these two summers, Central West Greenland experienced a particularly stark contrast in observed surface runoff elevation limits, with surface runoff extending to significantly higher elevations in 2019 than in 2020 (Tedstone and Machguth, 2022). For Central West Greenland, we use an elevation threshold of $2000\,m$ based on Zhang et al. (2023), who reported a mean elevation limit of surface water of $1609\,m$ above sea level in this region during the anomalously warm 2019 melt season. We apply this threshold based on the ArcticDEM digital elevation model (Morin et al., 2016).

In Antarctica, we focus on the Amery Ice Shelf and its surrounding grounded ice catchment (B-C drainage basin, Fig. 2), which on average experiences more meltwater pooling than any other Antarctic ice shelf. We compare the 2018-19 and 2020-21 melt seasons, which exhibit positive and negative anomalies in terms of open-water melt extent, respectively (Tuckett et al., 2022). For the Amery catchment, we use an elevation threshold of $1000\,m$, based on Tuckett et al. (2021), who reported $> 95\,\%$ of lakes at elevations below $500\,m$ in the 2004–20 time period, and only a handful of small lakes above $1000\,m$ even during high-melt summers. We apply this threshold based on the Reference Elevation Model of Antarctica (REMA) digital elevation model (Howat et al., 2019).

Our two study areas cover latitudes from 68.2° N to 72.1° N in Greenland and latitudes from 68.4° S to 74.0° S in Antarctica, meaning that ICESat-2 track spacing is similar over the two regions: in Central West Greenland RGT spacing varies from $\sim 8.8\,\mathrm{km}$ in the north to $\sim 10.8\,\mathrm{km}$ in the south; over the Amery Catchment RGT spacing varies from $\sim 7.9\,\mathrm{km}$ in the south to $\sim 10.7\,\mathrm{km}$ in the north. The total area of the Central West Greenland study region is about $650,000\,\mathrm{km}^2$ with a coverage of 50 distinct ICESat-2 reference ground tracks, and the area of the Amery catchment study region is about 3.5 million $\mathrm{km}^2$ with a coverage of 74 distinct ICESat-2 reference ground tracks.

For Greenland, we consider the annual melt season to be the 5-month period between the first day of May and the last day of September of a given year. Similarly, for Antarctica, we define the melt season to be the 5-month period between the first day of November and the last day of March of the following year. Based on these spatiotemporal parameters, we processed a total of 447 ATL03 granules with a total size of 1.15 TB, amounting to a total along-track distance about 760,000 km and comprising nine billion individual photon locations.

## 4 Results and discussion

Using FLUID, we identified a total of 1249 supraglacial lake segments over our two study areas in the available ATL03 data during the four melt seasons we considered (Table 1). We found that FLUID reliably detects potential supraglacial lake segments, with the number of detected lake segments varying with the strength of the melt season and their locations aligning well with imagery-derived melt extents (Sect. 4.1). Along-track lake depths determined by SuRRF agree well with manually annotated data, with deeper lakes in Central West Greenland than in the Amery catchment (Sect. 4.2). Our method is effective for detection and depth determination of supraglacial lakes over the ice sheets; however, it is not designed for ICESat-2 bathymetry over other targets, for which different methods have been developed (Sect. 4.3). Applying our method at an ice-sheet-wide scale and combining the results with satellite imagery would make it possible to develop data-driven models for accurate estimation of the volume of pooled surface meltwater across the ice sheets at high resolution and spatial coverage (Sect. 4.4).

### 4.1 FLUID lake detection and accuracy

#### 4.1.1 FLUID lake segment detection

Out of the 1249 supraglacial lake segments that we detected in the ICESat-2 data analyzed in this study, 500 were located in Central West Greenland and 749 in the Amery catchment. The number of lake segments that we detected using FLUID varied with strength of the melt season (Figs. 9 and 10). Over Central West Greenland, we identified 325 lake segments during the high-melt 2019 boreal summer versus only 175 during the low-melt 2020 boreal summer. Over the Amery catchment, we identified 721 lake segments during the high-melt 2018-19 austral summer versus only 28 during the (very) low-melt 2020-21 austral summer. To estimate how many unique supraglacial lakes were sampled by these detected ICESat-2 lake segments during each melt season, we calculated the maximum surface meltwater extent for each of the melt seasons independently using Landsat 8 imagery, based on the methods detailed in Tuckett et al. (2021) (blue regions in Fig. 9 and 10). We then matched each

**Table 1.** Summary statistics for the ICESat-2 lake segments extracted by FLUID/SuRRF for our regions and melt seasons of interest.

| | Amery catchment (B-C) | | Central West Greenland (CW) | |
|---|---|---|---|---|
| melt season | 2018-19 | 2020-21 | 2019 | 2020 |
| amount of surface melt | high | very low | high | low |
| area of Landsat 8 maximum melt extent ($km^2$) | 1872 | 100 | 1127 | 431 |
| number of total ICESat-2 lake segments | 721 | 28 | 325 | 175 |
| number of unique lakes sampled | 385 | 25 | 198 | 114 |
| number of high-quality lake segments | 165 | 5 | 196 | 109 |
| fraction of high-quality segments (%) | 23 | 18 | 60 | 62 |
| median lake segment depth (m) | 1.85 | 1.48 | 2.77 | 3.43 |
| maximum lake segment depth (m) | 10.4 | 17.3 | 25.8 | 15.1 |

detected ICESat-2 lake segment to a lake basin in these imagery-based melt extents and counted the number of total basins that were sampled by at least one ICESat-2 lake segment (see supplemental maps; Arndt and Fricker 2024c). Over Central West Greenland, this resulted in 196 unique supraglacial lakes being sampled by our data in 2019, and 109 lakes in 2020. Over the Amery Catchment, FLUID-SuRRF segments sampled 165 unique melt lakes in 2018-19 and 25 lakes in 2020-21.

Across all the data that we analyzed, FLUID and SuRRF determined on average $0.12\%$ of total distance along the ICESat-2 ground tracks to be meltwater surfaces. During the high-melt summers this number was $0.22\%$ and $0.21\%$ for Central West Greenland and the Amery catchment, respectively. The corresponding numbers for low-melt summers were $0.088\%$ and $0.0057\%$. SuRRF assigned a non-zero quality score to 475 of these lake segments, indicating that they likely contain high-quality bathymetric measurements. The fraction of non-zero quality lake segments was significantly higher for Central West Greenland ($61\%$) than for the Amery catchment ($22.7\%$). This is likely because the locations of supraglacial lake basins on Greenland's grounded ice are primarily controlled by bedrock topography (Lampkin and VanderBerg, 2011), which allows for well-defined lake basins to develop in the same locations every year. In contrast, lake basins on Antarctica's floating ice shelves are more difficult to distinguish from their much flatter surrounding topography, on which they usually form sporadically and in different locations each year (Arthur et al., 2022) with existing basins typically being advected to locations significantly further downstream from one melt season to the next (Arthur et al., 2020b). Furthermore, an appreciable portion of meltwater across Antarctic ice shelves is stored as slush (Dell et al., 2024), which often appears in ATL03 data with a surface return that is flat enough to look similar to a supraglacial lake, with large amounts of scattered photons below the surface that can be mistaken for a bathymetric signal.

### 4.1.2 Accuracy of FLUID lake segment detection

To validate the spatial extents of ICESat-2-derived supraglacial lake segments, we also used the Landsat-8-based maximum surface meltwater extents for each of the melt seasons (blue regions in Fig. 9 and 10). We found a high correspondence

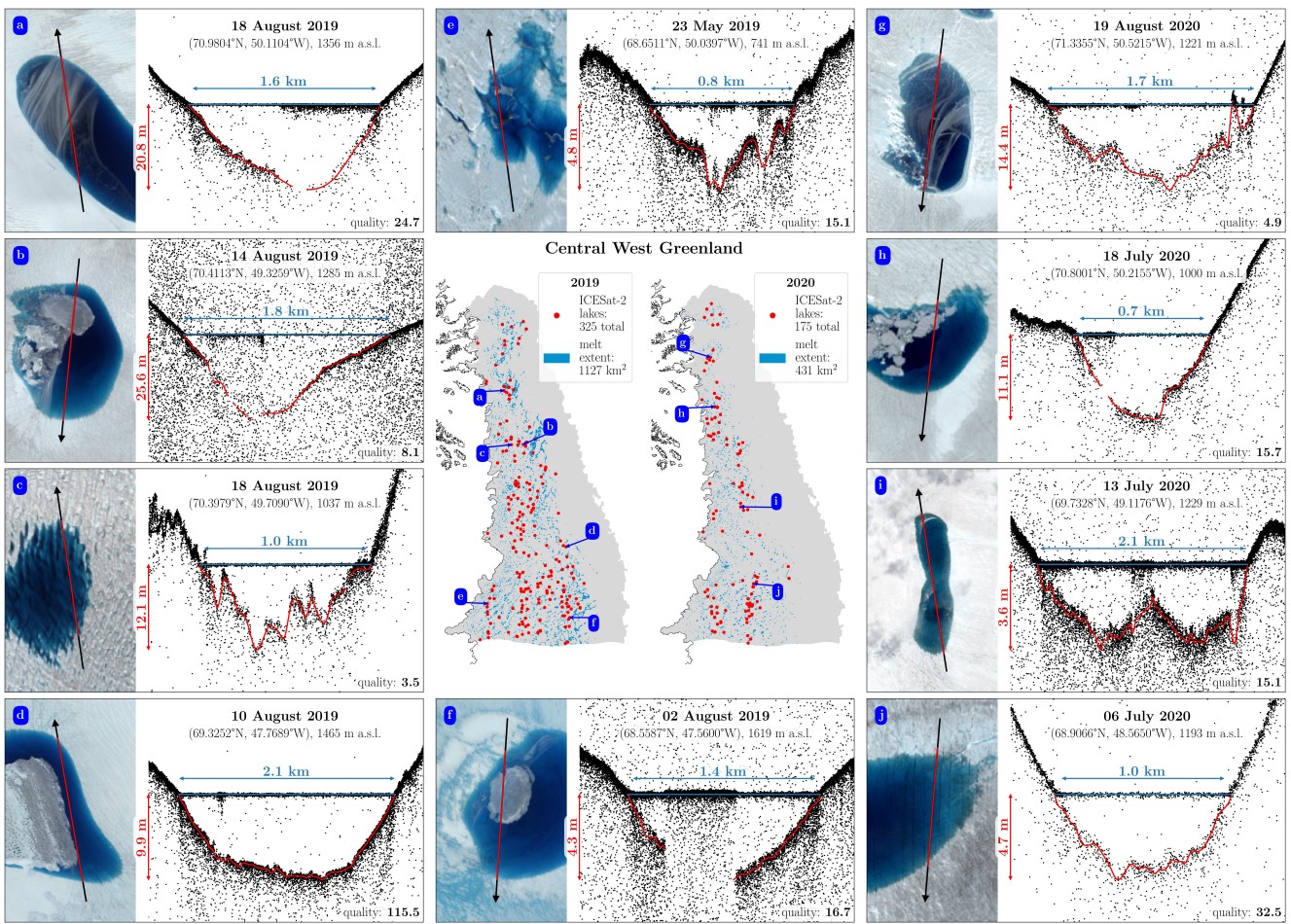

**Figure 9.** Center maps: FLUID/SuRRF algorithm testing in Greenland. Locations of melt lake segments detected in ATL03 data for the Greenland Ice Sheet's Central West drainage basin for melt seasons 2019 and 2020, mapped over the corresponding seasons' maximum meltwater extent from Landsat 8. Panels a-j: Examples showing the underlying ATL03 photon clouds and water depths calculated by SuRRF, for some of the lake segments shown on the maps. Numbers in lower right of panels are SuRRF lake segment quality scores.

between these estimates: For Central West Greenland, the ground tracks of $97.0\%$ of high-quality ICESat-2 lake segments coincided with the Landsat-8-derived maximum seasonal melt extent. The corresponding percentage for the Amery catchment was $95.8\%$.

Since there are no ground truth data of water depths available for any ICESat-2 data over supraglacial lakes, it is necessary to evaluate the performance of our method by manually examining the data in representative granules. We report results for ICESat-2 track 81, GT2L on 2 January 2019 over the Amery catchment, which was also studied in Fricker et al. (2021) and Xiao et al. (2023). To evaluate whether FLUID detects all supraglacial lake segments that have bathymetric data in this ATL03 ground track, we manually inspected the ATL03 data for any evidence of meltwater and determined whether any such along-

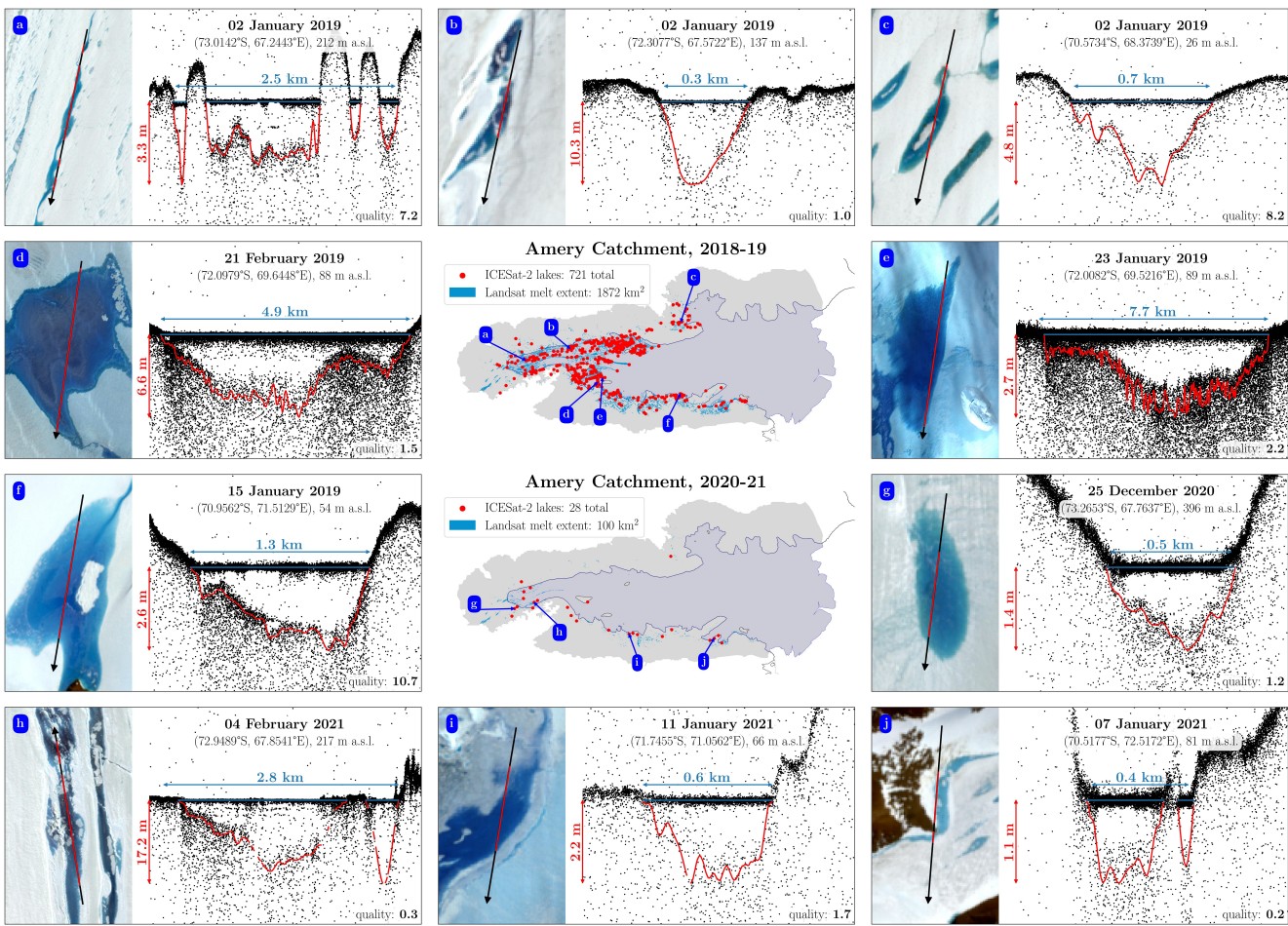

**Figure 10.** Center maps: FLUID/SuRRF algorithm testing in Antarctica. Locations of melt lake segments that FLUID detected in ATL03 data for the Antarctic Ice Sheet's B-C drainage basin for melt seasons 2019-20 and 2020-21, mapped over the corresponding seasons' maximum meltwater extent from Landsat 8. Panels a-j: Examples showing the underlying ATL03 photon clouds and water depths calculated by SuRRF, for some of the lake segments shown on the maps. Numbers in lower right of panels are SuRRF lake segment quality scores.

track contained a return signal from a lakebed. Out of 25 along-track segments with meltwater, we judged that only eight were supraglacial lake segments with a discernible bathymetric signal. We further evaluated whether the ATL03 photon cloud
misses any supraglacial lakes that track 81 GT2L crosses, by mapping it over a mosaic of Sentinel-2 scenes from the same day (same as used in Fricker et al. 2021). Based on visual inspection, the ICESat-2 ground track crossed supraglacial lakes that were clearly distinguishable in the imagery only in the same eight locations that we had also judged to be lake segments in the ATL03 data. Most other ICESat-2 segments that showed evidence of surface water in ATL03 also showed some evidence of meltwater in the imagery, and were associated with ice-filled crevasses, narrow melt channels, likely areas of slush, or melt
lakes with an opaque ice cover (for which depth determination is not possible). FLUID found 16 potential melt lake segments

within the same ground track, which all showed evidence of meltwater, but only the eight that we had also manually picked were classified as high-quality lake segments.

Out of all lake segments that FLUID detected, we deemed 308 to be false positives, many of which showed evidence of surface water but no distinguishable bathymetric return. However, we found that lake segments classified by SuRRF as "high quality" contain few false positives, with only 15 such segments that we manually removed from the data. The majority (11) of these were due to our study region erroneously extending past the calving front of the Amery Ice Shelf in Antarctica or marine-terminating outlet glaciers in Greenland, or due to ice-marginal lakes being included in the study region. Of the remaining four false positives, three were likely due to random noise and one showed a supraglacial lake but the data seemed to be affected by a "Did Not Finish Major Frame" data transfer error in the ATLAS Photon Counting Electronics (Magruder et al., 2024). While testing our algorithm, we found that false positives may also arise when ICESat-2's footprint includes two surfaces at different elevations for a substantial along-track distance, which is possible when the ground track passes over a flat calving front or crevasses with a bottom return at an acute angle. Where the study region extends past an ice shelf calving front, FLUID/SuRRF can also falsely classify the bathymetric return of a submerged "bench" as a supraglacial lake segment (Buck, 2024).

### 4.2 SuRRF depth retrieval and accuracy

#### 4.2.1 SuRRF: meltwater depths

The meltwater depths determined by SuRRF suggest that supraglacial lakes in Central West Greenland are generally deeper than those in the Amery catchment: The median of along-track maximum water depths of lake segments in Greenland was $3.09\,\mathrm{m}$ versus only $1.84\,\mathrm{m}$ in Antarctica. Median lake segment depth in the Amery catchment was greater during the high-melt season ($1.85\,\mathrm{m}$) than it was during the low-melt season ($1.48\,\mathrm{m}$). In contrast, median lake segment depth in Central West Greenland was less during the high-melt season ($2.77\,\mathrm{m}$) than it was during the low melt season ($3.43\,\mathrm{m}$) (Fig. 11, a). We hypothesize that this opposite behavior is due to different relationships between lake elevations and depths in these regions. A linear regression of lake segment depth on surface elevation in our data suggests that lake depth decreases with elevation in Central West Greenland ($p = 0.12$), while it increases with elevation in the Amery catchment ($p = 0.093$) (Fig. 11, b). In Greenland, where supraglacial lake basin locations are controlled by bedrock topography, lakes in a low-melt season are more likely to form at lower elevations, where repeated filling, bottom ablation and draining during most prior melt seasons has resulted in well-defined, deep lake basins (Lampkin and VanderBerg, 2011; Tedesco et al., 2012). In a high-melt season, surface melt extends further upwards above the ablation zone, where lake basins are less well-defined, forming smaller, shallower lakes and slush areas (Glen et al., 2024). On an Antarctic ice shelf, lakes in a low-melt season are more likely to form at lower elevations, where the ice is floating and the surface topography is very flat, resulting in mostly shallow lakes (Banwell et al., 2014). In a high-melt season, melt extends further upward of the grounding line where lakes form in deeper basins that are controlled by the bedrock topography (similar to lakes in Greenland; Bell et al., 2018), thus resulting in deeper lakes.

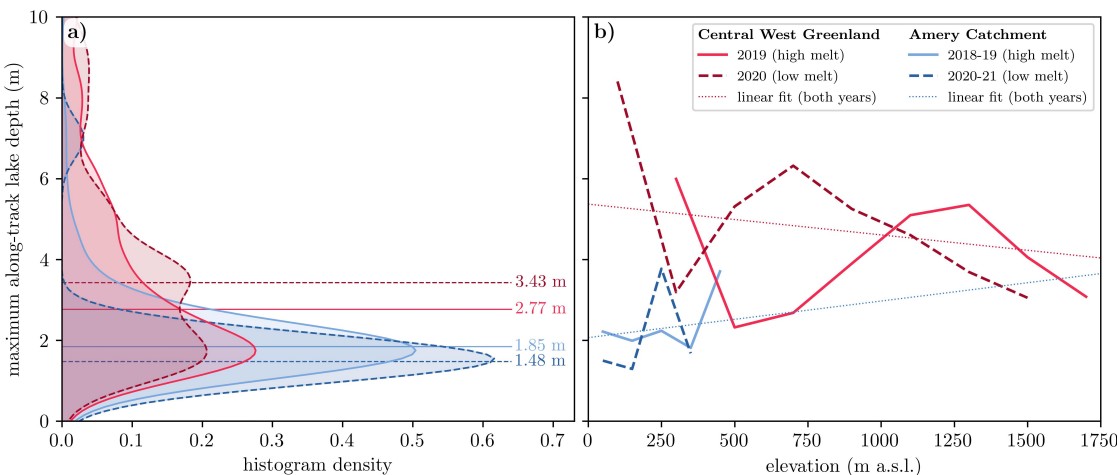

**Figure 11.** SuRRF lake segment depth statistics for two study regions in high and low melt years. a): Density of maximum lake depth distributions. The numbers shown are median lake segment depths during the corresponding melt season and study region. b): Elevation-binned means of maximum lake segment depths for each season and study region. The dotted lines are the linear fit to all data for the corresponding study region.

### 4.2.2 Accuracy of SuRRF depth retrievals and comparison with alternative methods

For the 16 potential melt lake segments that FLUID identified in ICESat-2 track 81, GT2L on Jan 2nd, 2019 over the Amery catchment (Sect. 4.1.2 Fricker et al., 2021), SuRRF assigned a nonzero quality score only to the eight data segments which we had manually determined to be supraglacial lake segments with a discernible bathymetric signal. For four of the lake segments identified in this granule, Fricker et al. (2021) established manually annotated baseline depth estimates, which were used to assess the performance of automated algorithms in the absence of any ground truth data. In addition to these four lake segments, Melling et al. (2024) used a similar method to establish manual estimates for five more segments in Southwest Greenland. We here use the manual annotations from both of these method comparison studies to evaluate SuRRF's depth estimation performance for all nine lake segments, and to briefly compare SuRRF to other methods whose results were included in these two comparative studies (Fig. 12). For a detailed comparison between various ICESat-2 lake depth algorithms (including an earlier version of SuRRF) and RTE-methods, we refer the reader to Fricker et al. (2021). For in-depth discussions comparing manually picked ICESat-2 depths to RTE methods, DEM-based approaches as well as empirical methods using *in situ* data, we refer the reader to Melling et al. (2024) and Lutz et al. (2024).

When run on the corresponding ATL03 granules, FLUID automatically detects all nine lake segments, and the SuRRF depth estimates track the general shape of the manually outlined lakebeds well (Fig. 12). This is also demonstrated by an average Pearson's correlation coefficient of $R = 0.992$ for the nine lake segments examined here, with averages of $R = 0.989$ for the segments on Amery catchment and $R = 0.994$ for the segments in Southwest Greenland. SuRRF depth estimates show an

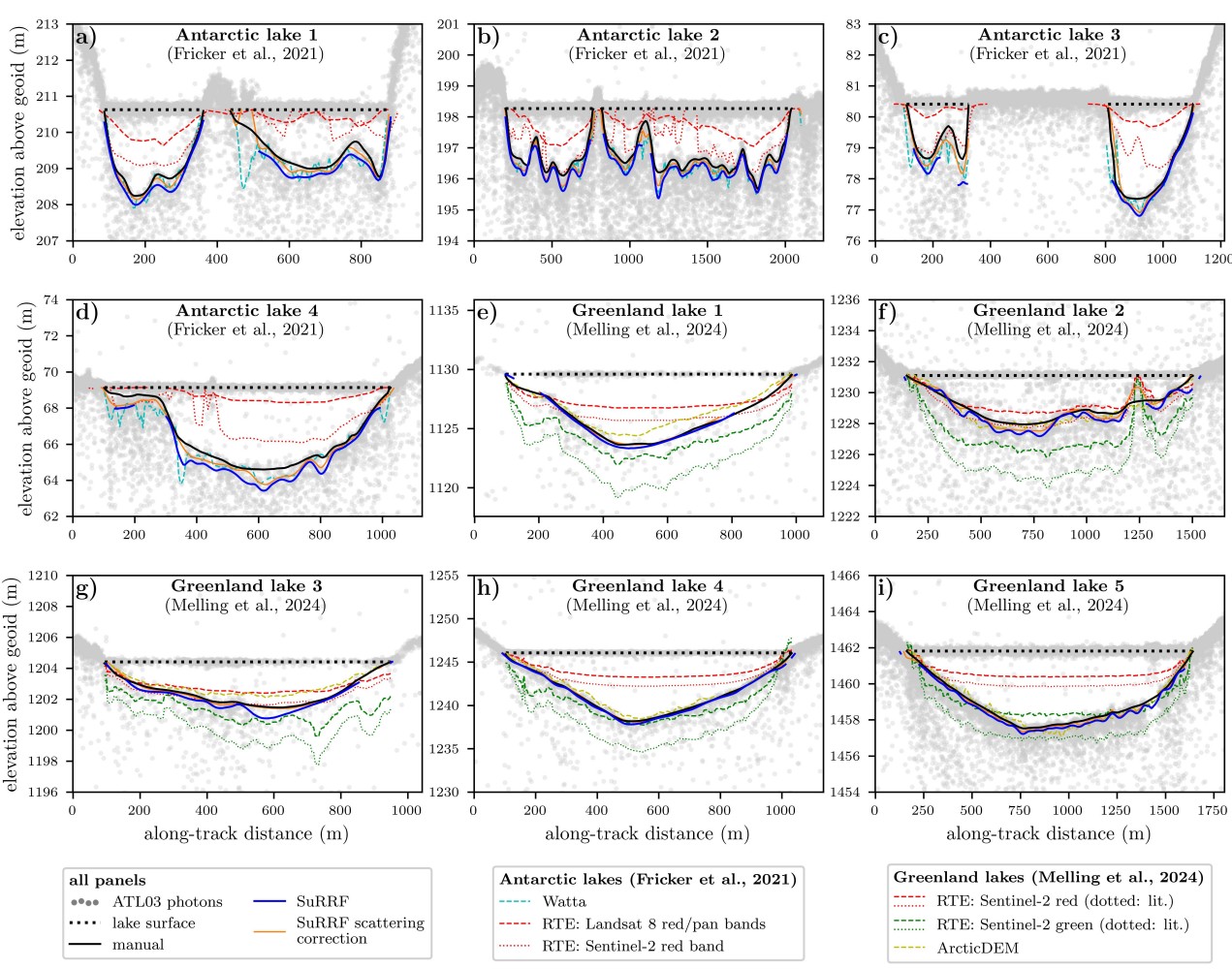

**Figure 12.** Comparison between SuRRF water depth estimates, manually annotated ICESat-2 depths and results from other methods for meltwater depth estimation for lakes in Antarctica and Greenland. a)-d): ICESat-2 melt lake segments on the Amery Ice Shelf, Antarctica, with manual annotations from Fricker et al. (2021). Other meltwater depth estimates that were reported by Fricker et al. are shown for the Watta algorithm (based on ICESat-2; Datta and Wouters, 2021) and for the RTE method applied to the average of Landsat 8's red and panchromatic bands (Spergel et al., 2021) and to Sentinel-2's red band (Moussavi et al., 2020). e)-i): ICESat-2 melt lake segments in Southwest Greenland with manual annotations from Melling et al. (2024). Other meltwater depth estimates that were reported by Melling et al. are shown for the RTE method individually applied to both Sentinel-2's red and green bands, as well as estimates from post-drainage lakebed topography in ArcticDEM (based on Bowling et al. 2019). SuRRF depth estimates are shown only where the estimated bathymetric signal confidence exceeds 0.5. To align along-track depths with the ATL03 photons (gray dots), the elevations of photons below the water surface elevation of each lake (black dotted lines) were corrected by dividing their vertical distance from the lake's surface by the refractive index for the speed of light in water.

average bias of $0.26\,\mathrm{m}$ deeper than the manually picked values, with a mean absolute error (MAE) of $0.27\,\mathrm{m}$. This results in SuRRF reporting a total amount of water that is $10\,\%$ larger than the estimate given by the manual baseline. For the lake segments on Amery Ice Shelf, the average bias is $0.29\,\mathrm{m}$ and the MAE is $0.29\,\mathrm{m}$. For the lake segments in Southwest Greenland, the average bias is $0.23\,\mathrm{m}$ and the MAE is $0.24\,\mathrm{m}$. When applying the correction for multiple scattering (Sect. 3.3.3) to SuRRF depth estimates, the average bias is reduced to $0.07\,\mathrm{m}$ deeper than the manually picked values, with a mean absolute error (MAE) of $0.15\,\mathrm{m}$ and a Pearson's correlation coefficient of $R = 0.993$. This results in the scattering-corrected version of SuRRF reporting a total amount of water that is $3\,\%$ larger than the estimate given by the manual baseline.

Based on the nine lake segments that were used to assess the accuracy of depth estimates, most algorithms (including SuRRF) seem to estimate slightly greater depths than the manually picked values (Xiao et al., 2023). When the bathymetric returns in the ATL03 point cloud are "fuzzy" the difference between the manual baseline and SuRRF water depth estimates tends to become larger. The majority of algorithms for lake depth retrieval from ATL03 operate on the assumption that the elevation with the locally highest photon density corresponds to the elevation of the ice-water interface at the lakebed, yet most altimetry experts placed the secondary return signal higher up in the point cloud, where the photon density first significantly increases below the water surface (Fricker et al., 2021). We agree that the true location of the lakebed is likely at a higher elevation than the along-track elevation of highest photon density, since multiple scattering increases the photon density below the elevation of the lakebed. However, the elevation at which the photon density first significantly increases represents the highest point within ICESat-2's $11\,\mathrm{m}$ footprint and furthermore tracks the upper bound of ATLAS's single-photon time-of-flight uncertainty of $800\,\mathrm{ps}$ (Markus et al., 2017), which corresponds to about $12\,\mathrm{cm}$ in photon height uncertainty. We therefore believe that the true water depth falls somewhere in between our (deeper) SuRRF estimates and the (more shallow) manual baseline estimates from Fricker et al. (2021) and Melling et al. (2024). Our scattering correction to SuRRF depth estimates is an attempt to reconcile this disparity between depth estimates. However, in the absence of ground truth *in situ* validation data for ICESat-2 lake segments, the correct magnitude of this correction remains unknown. This demonstrates an urgent need for *in situ* meltwater depth data that can be used to reliably validate the accuracy of ICESat-2 estimates.

Since many lakes on the Greenland Ice Sheet are transient and drain late in season (Johansson et al., 2013), it could be possible to obtain independent ICESat-2-based meltwater depth estimates by comparing "full vs empty" repeat-track measurements along ICESat-2 lake segments before and after the drainage. However, this approach would suffer from many of the same drawbacks that affect depth estimation from DEMs of a lake's bed topography that were acquired after it drained (Sect. 2.2), for example: effects of lake-bottom ablation, surface elevation change from precipitation and blowing snow deposits, as well as across-track advection of surface topographical features. Furthermore, this approach would not be feasible in Antarctica, where lake drainage is very rare, in particular on grounded ice. In cases where melt lake drainage is observed on the floating ice shelves, obtaining water depth from repeat-track elevation change is not possible due to the advection of surface topography with the ice flow and post-drainage flexural rebound (Warner et al., 2021). Due to these complexities, we do not attempt to validate ICESat-2 lake depth measurements using this method.

In addition to SuRRF and manual depth estimates, panels a)-d) of Fig. 12 show depth estimates for Antarctic lakes that were reported in Fricker et al. (2021), for the Watta algorithm (based on ICESat-2; Datta and Wouters, 2021) and for the RTE method

applied to the average of Landsat 8's red and panchromatic bands (Spergel et al., 2021) and to Sentinel-2's red band (Moussavi et al., 2020). Both SuRRF and Watta track the general shape of lakebed returns in the ATL03 photon clouds well (Pearson's correlation coefficients of $R = 0.99$ and $0.94$, respectively), and and largely agree with manually determined along-track water depths (MAEs of $0.29\,\mathrm{m}$ and $0.30\,\mathrm{m}$, respectively). In contrast to SuRRF, Watta appears to have a tendency to overfit where photon density near the lakebed is high with a large elevation spread, resulting in an unreasonably "wiggly" lakebed fit (e.g., Antarctic lake 1, 500-800 m). SuRRF's smoother fit under these conditions is likely due to the fact that it utilizes an adaptive kernel for its robust fit, whose width increases as the number of photons that narrowly cluster around the previous iteration's fit decreases (Sect. 3.3.1). In contrast to SuRRF, Watta also attempts to fit the lakebed across the entire lake basin, even where the lakebed is not visible or indistinguishable from noise, which can sometimes result in arbitrary, unrealistic depth estimates (e.g., Antarctic lake 1, around 450 m). Under such conditions SuRRF assigns a low confidence score to the lakebed fit and discards associated depth estimates to prevent arbitrary results. However, in some cases this results in SuRRF discarding depth estimates where Watta appears to fit the lakebed reasonably well (e.g., Antarctic lake 3, 250-300 m). The RTE approach based on Landsat 8's red/panchromatic band average consistently underestimates water depths, reporting a total amount of water that is $\sim 73\,\%$ lower than the manual baseline. The RTE approach based on Sentinel-2's red band also underestimates water depths, and reports a total amount of water that is $34\,\%$ lower than the manual baseline.

Panels e)-i) of Fig. 12 also show non-ICESat-2 depth estimates for Greenland lakes that were reported in Melling et al. (2024), for the RTE method individually applied to both Sentinel-2's red and green bands, as well as estimates from post-drainage lakebed topography in ArcticDEM (based on Bowling et al. 2019). The RTE approach based on Sentinel-2's red band generally underestimates depths and reports a total amount of water that is $42\,\%$ lower than the manual baseline (similar to this method's performance over Antarctic lakes). In contrast, the RTE approach based on Sentinel-2's green band generally overestimates depths and reports a total amount of water that is $34\,\%$ larger than the manual baseline. However, Melling et al. also note that when using values of tuneable parameters that have been commonly used in the literature in the past (Sneed and Hamilton, 2007; Georgiou et al., 2009; Pope et al., 2016), the RTE approach for Sentinel-2's green band overestimates lake depths even more, which results in reporting a total amount of water that is $84\,\%$ larger the manual baseline, with individual depths being overestimated by up to $153\,\%$. This implies that RTE-based methods, while being popular for their simplicity, can potentially result in highly inaccurate meltwater volume estimates. The depth estimates derived from DEMs of emptied lake basins match the ICESat-2 manual baseline reasonably well, and when compared with it underestimate the total amount of water by $6\,\%$ with a MAE of $0.34\,\mathrm{m}$. Since this method's performance is comparable to that of ICESat-2-based methods, this implies that DEM-based methods could be used supplement ICESat-2 depth measurements for labeling reflectance in passive optical imagery with supraglacial water depths, at least on the Greenland Ice Sheet where melt lakes on grounded ice drain regularly (Johansson et al., 2013).

## 4.3 ICESat-2 bathymetry over other targets

Beyond estimating the depth of supraglacial lakes, ICESat-2's bathymetric capabilities have been used for various other applications. Many algorithms employed for depth retrieval from ATL03 share significant similarities, enabling method development

for ICESat-2-derived bathymetry to benefit from broader cross-discipline collaboration (Parrish et al., 2022). Methods similar to FLUID-SuRRF have been used for satellite-derived nearshore ocean bathymetry (e.g.; Parrish et al., 2019; Ma et al., 2020; Thomas et al., 2021), estimating water depths of inland waters (e.g.; Li et al., 2019; Xu et al., 2020; Jasinski et al., 2023) and tracking the evolution of melt pond depths on sea ice (e.g.; Farrell et al., 2020; Tilling et al., 2020; Herzfeld et al., 2023; Buckley et al., 2023). However, there are also notable differences between bathymetric applications of ICESat-2 in different environments that have led to the development of specialized approaches, in particular for their large-scale implementation. For nearshore and inland bathymetry applications, the locations of the desired bathymetry estimates are usually known *a priori* and the bottom topography is often considered constant, so bathymetric measurements can be accumulated over time and compared to non-concurrent validation data. In contrast, the ephemeral nature of supraglacial lakes on the ice sheets and melt ponds on sea ice makes it necessary to automatically detect their locations directly from ATL03 and makes it more difficult to reliably validate depth estimates. Our FLUID algorithm addresses this issue for supraglacial lakes on the ice sheets: it automatically detects the locations of lakes by relying on the fact that the water surface reflection over open water presents as a flat line in geoid-corrected ICESat-2 elevation data, while the surrounding topography on the ice sheets is almost always sloped. This makes our algorithm efficient over the ice sheets by discarding most non-lake photon segments in the first "flatness check" step. However, it makes our algorithm less suited for detecting lakes on sea ice, where most segments over the open ocean and thin sea ice would likely pass the flatness threshold.

### 4.4 Future studies

Our ICESat-2-derived water depths are the first comprehensive dataset of supraglacial lake depths directly measured from a satellite. However, these along-track observations alone are too sparsely spaced in space and time to allow for the calculation of lake volumes or the continuous tracking of meltwater throughout the progression of a melt season. The large volume and wide variety of data that our method provides suggest that ICESat-2-based depth measurements obtained from applying FLUID-SuRRF at ice-sheet-wide scale could be used to better constrain parameters in existing methods which estimate meltwater volumes from high-resolution, spatially continuous satellite imagery. Furthermore, our method could be used to extract pan-ice-sheet meltwater depths and combine them with concurrent satellite imagery, thus providing a training data set that would enable the development of data-driven models of the relationship between meltwater depth and satellite imagery reflectances, based on statistical learning methods. Since in the absence of ground truth validation data our depth validation efforts were based on manual annotation of the data, we acknowledge that there may be a small but potentially significant bias towards over-estimating water depths with FLUID-SuRRF. This highlights an urgent need for ground truth *in situ* water depth measurements of supraglacial lakes that coincide with ICESat-2 overpasses, to enable calibration and validation of depth estimates.

### 5 Summary

Supraglacial lakes form seasonally around most of the margins of the Greenland and Antarctic ice sheets. In a warming climate, these lakes have the potential to significantly impact the future stability of both ice sheets through processes that

are not yet understood well enough to be included into models. To confidently project future sea-level rise, better satellite observations of surface meltwater are needed to enable science that produces a better mechanistic understanding of how ice dynamics are impacted by the pooling of surface meltwater in lakes. Until recently, any available methods used to estimate supraglacial lake depth from satellite data have been required to make strong assumptions and to use poorly constrained parameters, making it difficult to accurately assess the distribution of meltwater volumes across ice surfaces. Multiple case studies have successfully demonstrated that supraglacial lake depths can now be directly measured from photon refraction in ICESat-2's laser altimetry data. However, ICESat-2 data had not previously been used at scale for this purpose because its photon-level product comprises hundreds of terabytes of unstructured point cloud data along spatially discrete ground tracks, which makes it difficult to integrate the data with spatially continuous data in existing workflows.

To address this challenge, we have proposed a computational framework that allows users to detect lake segments and determine their water depths across all available ICESat-2 data for any desired ice sheet drainage basins and melt seasons. Using distributed high-throughput computing, this framework applies the fully automated, two-step FLUID/SuRRF algorithm to large numbers of ICESat-2 ATL03 photon data granules in parallel. To test our method, we applied FLUID-SuRRF to all available ICESat-2 data over two drainage basins, one on the Antarctic Ice Sheet and one on the Greenland Ice Sheet, for a high-melt and a low-melt summer. We have demonstrated the following for our method:

1. It reliably detects supraglacial lake segments based on the flatness of their surface and the presence of a lakebed return.

2. There is a potential for false positives, but their impact can be effectively mitigated by filtering for the strength of the bathymetric signal.

3. Water depth estimates are accurate based on manual validation of the data, however, there is an urgent need for *in situ* data for definitive ground truthing.

4. It can be applied at scale by leveraging distributed high-throughput computing.

5. The resulting data effectively capture spatial and temporal variability in meltwater extent and depth.

Our framework can be used to generate a comprehensive data product of supraglacial lake depths for Greenland and Antarctica since the launch of ICESat-2, which would enable the development of data-driven models of meltwater volumes in satellite imagery.

*Code and data availability.* The FLUID/SuRRF source code is freely available at https://doi.org/10.5281/zenodo.10905941 (Arndt and Fricker, 2024a). To execute this code, users need to create a free NASA Earthdata login for ICESat-2 data access. The source code contains a singularity container in which this version of FLUID/SuRRF can be executed. The main Python script `detect_lakes.py` can be run either locally on any individual ATL03 granule, or on many granules in parallel on any computing cluster that supports the specified computing environment or the use of singularity containers. In this study we present our implementation of FLUID/SuRRF on the OSG Open Science Pool because it provided us with free computational infrastructure. Due to funding mandates, free access to the OSG Open Science

Pool is limited to researchers contributing to a US-based project at an academic, government, or non-profit organization, or researchers affiliated with any project or institution that operates its own local access point. This means that to implement FLUID/SuRRF on the OSG Open Science Pool as described here, you need to have at least one collaborator on your team to whom these criteria apply. This collaborator can register your project with OSG on the Open Science Pool. Then, anyone contributing to the project can register for an account on OSG Connect to gain access to the Open Science Pool. For more information, see https://osg-htc.org/services/open_science_pool.html and https://osg-htc.org/about/organization/. More information is also included in the README file. The supraglacial meltwater depth estimates and associated "quicklook" plots for all 1249 lake segments identified in this study are available at https://zenodo.org/doi/10.5281/zenodo.10901737 (Arndt and Fricker, 2024b). All data and code needed to reproduce the figures in this study, as well as supplementary figures, are available at https://doi.org/10.5281/zenodo.10901826 (Arndt and Fricker, 2024c). ICESat-2 ATL03 data is available at NSIDC (Neumann et al., 2023b). Sentinel-2 and Landsat imagery were accessed via Google Earth Engine (Gorelick et al., 2017). Drainage basins for Greenland are available at DRYAD (Mouginot and Rignot, 2019). Drainage basins for Antarctica are available at NSIDC (Mouginot et al., 2017). ArcticDEM mosaics for elevation thresholding are available at https://www.pgc.umn.edu/data/arcticdem/ (Porter et al., 2023) and REMA DEM mosaics are available at https://www.pgc.umn.edu/data/rema/ (Howat et al., 2022). ICESat-2 ground track KML files are available at https://icesat-2.gsfc.nasa.gov/science/specs.

## Appendix A: Calculation of photon density ratios in FLUID flatness check

For a given major frame, let $l_{\mathrm{mframe}} \approx 140\,\mathrm{m}$ be the along-track length of the major frame, and let $P$ be the set of photons with $h_i$ denoting the geoid-corrected height of photon $i \in P$. We calculate the five photon densities used in the flatness check as:

$$d_0 = \frac{\sum_{i \in P} \left[ |h_i - h_{\mathrm{peak}}| \leq w_{\mathrm{peak}} \right]}{2 w_{\mathrm{peak}} l_{\mathrm{mframe}}} \tag{A1}$$

$$d_1 = \frac{\sum_{i \in P} \left[ 0 < h_i - h_{\mathrm{peak}} - w_{\mathrm{peak}} \leq w_{\mathrm{buffer}} \right]}{w_{\mathrm{buffer}} l_{\mathrm{mframe}}} \tag{A2}$$

$$d_2 = \frac{\sum_{i \in P} \left[ -w_{\mathrm{buffer}} \leq h_i - h_{\mathrm{peak}} + w_{\mathrm{peak}} < 0 \right]}{w_{\mathrm{buffer}} l_{\mathrm{mframe}}} \tag{A3}$$

$$d_3 = \frac{\sum_{i \in P} \left[ |h_i - h_{\mathrm{peak}}| > w_{\mathrm{peak}} \right]}{(h_{\mathrm{max}} - h_{\mathrm{min}} - 2 w_{\mathrm{peak}}) l_{\mathrm{mframe}}} \tag{A4}$$

$$d_4 = \frac{\sum_{i \in P} \left[ h_i - h_{\mathrm{peak}} > w_{\mathrm{peak}} \right]}{(h_{\mathrm{max}} - h_{\mathrm{peak}} - w_{\mathrm{peak}}) l_{\mathrm{mframe}}} \tag{A5}$$

where $w_{\mathrm{peak}} = 0.1\,\mathrm{m}$ is the width of the density peak, $w_{\mathrm{buffer}} = 0.35\,\mathrm{m}$ is the width of the buffer around the peak, and $h_{\mathrm{min}}$ and $h_{\mathrm{max}}$ are the bottom and the top of the telemetry window, respectively. Here $[\cdot]$ are Iverson brackets, which evaluate to 1 if the condition inside the brackets is true, and to 0 otherwise.

## Appendix B: Calculation of photon signal confidences in FLUID

Let $P$ be the set of photons within an along-track segment (e.g., a major frame), and for each photon $i \in P$, let $x_i$ and $h_i$ be the along-track distance and elevation, respectively. To give appropriate relative weights to the two spatial dimensions when calculating photon densities, we need to adjust along-track distance by an aspect ratio parameter $r_a$. We found that for typical

supraglacial lake segments, a value of $r_a = 30$ works well. Denote by $\tilde{x} = x/r_a$ the aspect-ratio-adjusted along-track distance. We now want to express a photon's signal confidence as the average of the inverse euclidean distances between the photon and up to $k_{\max}$ nearest neighbors within a search radius of $r_s$, where distances are normalized and clamped to this search radius. Let $K_i$ be the set of photons including $i$ and its $k_{\max}$ nearest neighbors, i.e. $K_i \subseteq P$ s.t. $|K_i| = k_{\max} + 1$ and

$$d_{i,j} \geq \max_{n \in K_i} d_{i,n} \quad \forall j \in P \setminus K_i \tag{B1}$$

where

$$d_{i,j} = \sqrt{(\tilde{x}_j - \tilde{x}_i)^2 + (h_j - h_i)^2} \tag{B2}$$

is the aspect-ratio-adjusted euclidean distance between photons $i$ and $j$. Then, the signal probability of photon $i$ is estimated as

$$p_i = \frac{1}{k_{\max}} \left( k_{\max} - \sum_{j \in K_i} \left( \frac{\min(d_{i,j}, r_s)}{r_s} \right) + 1 \right). \tag{B3}$$

Note that this implies $p_i \in [0, 1]$. In particular, $p_i = 0$ if photon $i$ does not have any neighbors within a radius of $r_s$, and $p_i$ would be equal to $1$ only for a "perfect photon", whose location coincides exactly with that of $k_{\max}$ other photons. To effectively discriminate between signal and noise, we need to choose $k_{\max}$ and $r_s$ in a way such that typical background noise photons are assigned a small target value, which we set to $p_{\text{noise}} = 0.05$.

Assuming that a major frame contains only noise photons and a flat surface with a signal photon spread of $w_{\text{signal}} = 0.3\,\text{m}$, 890 we can estimate the aspect-ratio-adjusted mean area per background photon (i.e. the inverse photon density) as

$$a = \frac{(h_{\max} - h_{\min} - 2w_{\text{signal}})\, l_{\text{mframe}}}{r_a \sum_{i \in P} [|h_i - h_{\text{peak}}| > w_{\text{signal}}]}. \tag{B4}$$

If background noise photons follow a random uniform distribution, this means that to find an expected $k$ neighbors around a typical background photon, one would have to search within a radius of $r(k) = \sqrt{a(k+1)/\pi}$ around that photon. Using the fact that the average inverse distance of a point in a circle with radius $R$ to its origin is $R/3$ (Stone, 1991), we can set

$$r_s = \sqrt{3p_{\text{noise}}} r(k_{\max}) = \sqrt{\frac{3 a p_{\text{noise}}(k_{\max} + 1)}{\pi}} \tag{B5}$$

to ensure that $p_i \leq p_{\text{noise}}$ for typical background noise photons. We found that setting $k_{\max} = 15$ strikes a good balance between being sensitive enough to small-scale signals, and being too sensitive to small amounts of noise photons clustering together by random chance.

## Appendix C: Calculation of quality heuristics in FLUID bathymetry check

To detect any potential bathymetric returns in a major frame, we divide it into $n_{\text{seg}} = 10$ along-track sub-segments of equal length. For each sub-segment, we calculate the median of the FLUID photon-level signal probability $p_i$ for photons $i$ within

0.1 m elevation bins, assigning a value of zero whenever there are no photons within any bin. This results in an empirical function that relates signal probability to elevation $h$, which we here denote by $p(h)$. In addition, we calculate the analogous function of photon density $d(h)$ as a simple histogram for $0.01$ m elevation bins. Let $d'(h) = d(h) \, \forall h : |h - h_{\text{peak}}| > w_{\text{signal}}$ and zero otherwise. Denote by $\cdot|_l$ a smoothing operator using a gaussian window with standard deviation of $l$. Now calculate signal confidence as a function of $h$ as

$$c(h) = p(h)|_l \min \left( \frac{d(h)|_l}{\max_h d'(h)|_l}, 1 \right), \tag{C1}$$

where $p(h)$ is linearly interpolated to match the domain of $d(h)$. Note that this implies $p(h) \in [0, 1] \, \forall h$ in the telemetry window. To determine whether a potential bathymetric signal peak is present, we find peaks in $c(h)$. If there are at least two peaks with prominence $\rho \geq 0.1$, and the elevation of the peak closest to $h_{\text{peak}}$ is found within an elevation difference of $w_{\text{signal}}$, then out of the peaks whose elevation is below $h_{\text{peak}} - w_{\text{signal}}$ the peak with the highest prominence is considered a potential bathymetric peak. Denote by $n_{\text{peaks}}$ the number of potential bathymetry peaks found in the major frame, and let $h_i$ be the elevation and $\rho_i$ the prominence of peak $i \in [1, 2, \ldots, n_{\text{peaks}}]$. We only consider the major frame to be potentially part of a lake segment if $n_{\text{peaks}} \geq 3$. In this case, we define the following heuristics for different components that affect the quality of the bathymetric return within the major frame.

$$q_1 = f^{3/2} \tag{C2}$$

$$q_2 = \min \left( \frac{\sum_i \rho_i}{n_{\text{peaks}}} 2(\min(\{2f, 1\}) - 1) + 1, 1 \right) \tag{C3}$$

$$q_3 = \min \left( \frac{1}{\log_5 \left( \max \left( \Delta h, 1.1 \right) \right)}, 1 \right) \tag{C4}$$

$$q_4 = \frac{1}{1 + \frac{d_h}{\max(\Delta h, 0.5 n_{\text{seg}})}} \tag{C5}$$

where $f = n_{\text{peaks}}/n_{\text{seg}}$ is the fraction of sub-segments for which a potential bathymetric peak was detected, $\Delta h = \max_i h_i - \min_i h_i$ is the elevation spread of the detected peaks, and

$$d_h = - \sum_{i=2}^{n_{\text{seg}}-1} \frac{|h_i - h_{i-1}| + |h_{i+1} - h_i|}{2} \max \left( \text{sgn}(h_i - h_{i-1}) \text{sgn}(h_{i+1} - h_i), 0 \right). \tag{C6}$$

Note that the quality heuristics $q_i \in [0, 1] \forall i$, with higher values implying a "better" bathymetric signal. The expressions used here were derived by trial-and-error and designed such that $q_1$ penalizes major frames with smaller numbers of detected bathymetry peaks, $q_2$ penalizes major frames with less prominent peaks, $q_3$ penalizes major frames with a very large overall spread of peak elevations, and $q_4$ penalizes major frames with peak elevations that do not align along a smooth surface. Based on these quality heuristics for the major frame, we calculate a quality summary $q_s = \prod_{i=1}^4 q_i$ for each major frame.

While the quality heuristics $q_i$ were obtained by trial-and-error, the starting points for this approach were based on the following assumptions and observations:

– The starting point for $q_1$ was the idea that major frames with a smaller number of detected bathymetry peaks are less likely to have a consistent signal from a lakebed. The way the equation for $q_1$ is designed, major frames with a very

small fraction of detected bathymetry peaks (0 to $30\%$ of sub-segments) are disproportionately penalized, since such small numbers of peaks are much more likely to be noise.

- The starting point for $q_2$ was the idea that major frames with less prominent peaks represent either a weak, inconsistent bathymetric signal or noise. Here, $q_2$ is equal to the mean prominence of peaks when the fraction of detected bathymetry peaks $f \leq 0.5$. However, for fractions larger than $0.5$, the assumption is that the bathymetric signal is consistent enough that even smaller mean peak prominence suggests that the bathymetric signal is clearly visible.

- The starting point for $q_3$ was the observation that in most cases supraglacial lake segments with a usable bathymetric signal have fairly small lakebed slopes. Therefore, we do not expect the lakebed elevation of a major frame with a good signal to span a large elevation range. In contrast, we observed that major frames with detected potential bathymetric peaks that span a very large elevation range are often due to noise in the data. Based on this, we designed the equation for $q_3$ such that its value drops off once the total elevation range of detected bathymetry peaks within the major frame (of length $\approx 140\,\mathrm{m}$) exceeds $5\,\mathrm{m}$. There are, however, some lake segments with a clear bathymetric signal that do exhibit a large range of lakebed elevations (often due to a single burst of noise being detected as a potential bathymetric peak). Therefore, we made sure that the value for $q_3$ is large enough for major frames to still pass the bathymetry check if $\Delta h$ is very large but its other quality heuristics $q_i$ are closer to $1$.

- The starting point for $q_4$ was the idea that in most cases bathymetry peak elevations will align along a smooth surface in the along-track distance direction rather than randomly fluctuate (which is usually the case for noise). Here, we penalize every "direction change" (i.e., wherever a peak has two neighbors and its elevation constitutes a local minimum or maximum). We allow for random fluctuations of up to $0.5\,\mathrm{m}$ per peak detection without a large penalty, since we observed that a vertical photon spread of up to about this value is quite possible even for lake segments with a somewhat "fuzzy", yet clearly distinguishable return signal from the lakebed.

A figure that illustrates these quality heuristics is available in the supporting information at https://doi.org/10.5281/zenodo. 10901826 (Arndt and Fricker, 2024c).

*Author contributions.* **Philipp S. Arndt:** Conceptualization, Methodology, Software, Validation, Formal analysis, Investigation, Data Curation, Visualization, Writing – Original Draft, Funding acquisition **Helen A. Fricker:** Supervision, Writing – Review & Editing, Conceptualization, Project administration, Resources, Funding acquisition

*Competing interests.* The authors declare no competing interests.

*Acknowledgements.* We would like to extend our gratitude to the referees, Jennifer Arthur, Ian Brown, and Sammie Buzzard, for their thoughtful and constructive feedback, which greatly contributed to improving the clarity and context of our manuscript. We also thank Bert Wouters for his insightful community comment, which ensured that we gave appropriate credit to past work where it was due, and helped us communicate the motivation and novel aspects of our study more clearly. Additionally, we appreciate the guidance and support of our handling editor, Arjen Stroeven, throughout the review process. We would like to thank the ICESat-2 Project Science Office and Science Team, particularly the Land Ice group for helpful input regarding ICESat-2 data and its peculiarities, the OSG Research Computing Facilitation Team for support with implementation of our method on the Open Science Pool, and the National Snow and Ice Data Center for help with data access. The authors were supported through the following grants: Future Investigators in NASA Earth and Space Science and Technology Award #80NSSC20K1666 and NASA ICESat-2 Science Team Awards #80NSSC20K0977 and #80NSSC23K0934. This research was done using services provided by the OSG Consortium (Pordes et al., 2007; Sfiligoi et al., 2009; OSG, 2006, 2015), which is supported by the National Science Foundation awards #2030508 and #1836650. The figures in this publication were produced using Scientific Color Maps (Crameri et al., 2020) where applicable. The writing in a handful of short sections of this manuscript were guided by the AI tools scite.ai and chatGPT.

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
