# Peer review of "A Framework for Automated Supraglacial Lake Detection and Depth Retrieval in ICESat-2 Photon Data Across the Greenland and Antarctic Ice Sheets"

_EGUsphere, 2024_

## Referee Comment (RC1)

**A Framework for Automated Supraglacial Lake Detection and Depth Retrieval in ICESat-2 Photon Data Across the Greenland and Antarctic Ice Sheets**

Philipp Arndt and Helen Fricker

Jennifer Arthur (Referee) jennifer.arthur@npolar.no

**General comments:**

This manuscript uses ICESat-2's ATL03 altimetry product to develop two new algorithms which together provide a scalable framework for supraglacial lake detection and depth determination from ATL03 data.

Surface melt is an important, yet poorly constrained, component of ice-sheet surface mass balance, leading to surface meltwater accumulating as lakes on ice-shelf surfaces and on grounded ice. In Antarctica, this has been linked to the process of meltwater-driven hydrofracture, which can trigger rapid ice-shelf collapse. Accurately measuring supraglacial lake meltwater depths from satellite data is important due to the challenges in obtaining in situ measurements and is needed for modelling meltwater interactions with ice sheet dynamics. However, few studies have developed automated lake depth estimation methods that are scalable beyond small data subsets, and previous studies rely on methods with poorly constrained parameters and a lack of in situ measurements.

The authors apply their algorithm framework to two regions that experience high surface melt (central west Greenland and the Amery Ice Shelf) and are able to reliably detect lakes where lake bathymetry is visible. The methodology appears robust, and the algorithm performs well even for more complex lakes (especially in Antarctica), including thin, elongated lakes and those with patchy ice cover. The authors found 1249 lakes with their algorithm during four melt seasons and conclude that lake depths agree well with manually-picked lakebeds in ICESat-2 along-track segments.

Overall, it is my view that this study is of broad interest to the cryospheric community as it builds upon previous work focused on supraglacial lake depths on both ice sheets, especially in the context of ice-shelf surface hydrology and dynamics, by paving the way for developing pan-ice sheet supraglacial lake depth and volume products. I think the surface hydrology and ice shelf communities would be very interested to see this algorithm applied at an ice-sheet scale in future.

In general, this is a very well-written manuscript with detailed methods, clear figures and most of my comments are relatively minor. Once the authors address these, I can therefore recommend that this manuscript is suitable for publication in *The Cryosphere*.

I have a few general points:

1. The manuscript could elaborate in a bit more depth about how this algorithm framework approach differs from previous algorithms that have been developed for supraglacial lake depth/bed detection with the same ICESat-2 product (e.g. Datta and Wouters, 2021, who use ICESat-2 ATL03 to derive lake depths and constrain empirically-derived depths from Landsat 8, Sentinel-2, PlanetScope and SkySat imagery). It is clear to me that previous approaches have been tested on small subsets and so are not as scalable as the approach presented here, but I think more detail could be provided for readers.

2. 'Lakes with a bathymetric signal' is referred to often throughout the manuscript, and I suggest perhaps the first time this is mentioned adding for clarity: 'lakes with a bathymetric signal, i.e lakes with a visible or partly-visible lakebed'.

3. Performance of the SuRRF algorithm is compared to manual baseline estimates in Section 4.2.2, but I think this could be interesting to elaborate briefly on how the depth estimates are different from results that derive depths using the Radiative Transfer Equation (given that this is a commonly-applied

method). For example, does SuRRF tend to detect deeper lakes, implying previous methods underestimate lake depths?

4. For those who are less familiar with distributed High-Throughput Computing, how widely useable is this algorithm for others to whom the OSG Open Science Pool is not accessible, aka. non-US-based researchers?

**Specific comments:**

L30: I would cite something more relevant than this EGU abstract here, e.g. Gilbert and Kittel (2024), https://doi.org/10.1029/2020GL091733.

L34: Perhaps add a sentence or two here about what direct observations of supraglacial lake depths do exist, particularly in Antarctica (to highlight the paucity of observations).

L109: add where these in situ measurements were collected (west Greenland).

L484: I would rename the labels 'CW < 200 m' and 'B-C < 1000 m' to 'Surface Elevation < 200 m' and 'Surface Elevation < 1000 m' for clarity, or else clarify this in the figure caption.

L543: could also add here that lakes typically are advected downstream year-on-year (e.g. Arthur et al., 2020).

L579-585: This explanation for differences in lake elevation-depth relationships could be better cited. For example, Banwell and MacAyeal (2015) highlight lake deepening by lake-bottom ablation.

L613: I'm not sure about the overall relevance of Section 4.3 and don't think it adds substantially to the manuscript, because the algorithm application to ocean bathymetry, inland waters or sea ice melt ponds is less relevant in the context of this manuscript. I would suggest removing this section and moving the content from 'The ephemeral nature of supraglacial lakes…' to further up in the introduction as justification for your algorithm.

L630: I don't entirely agree with the part of this sentence that states the calculated water depths prevent the calculation of lake volumes. Surely it would be possible to calculate lake volumes by combining the ICESat-2-derived depths with lake extents derived from optical imagery (Landsat 8) as an initial estimate? I do understand though that with the small datatset you present here it is not enough to track the meltwater through the melt season.

**Technical/minor corrections:**

In some places the Surface Removal and Robust Fit algorithm is referred to as SuRRF and in others as SuRFF, so check throughout for consistency.

L178: delete 'each' (same on L653).

L611: Is a word missing here? '.. in between our (deeper) SuRRF estimates *and* the (more shallow) manual baseline estimates'.

L640: don't hyphenate 'well-enough'.

L710:  origin → its origin

---

## Author Comment (AC1)

**Response to RC1**

A Framework for Automated Supraglacial Lake Detection and Depth Retrieval in ICESat-2 Photon Data Across the Greenland and Antarctic Ice Sheets

Philipp Sebastian Arndt[1] and Helen Amanda Fricker[1]

[1] Scripps Polar Center, University of California San Diego, 8885 Biological Grade, La Jolla, CA 92037, USA

Correspondence:
Philipp Sebastian Arndt (parndt@ucsd.edu)

Discussion: https://doi.org/10.5194/egusphere-2024-1156

Comments from the reviewers are given in black.
Our responses are given in red.
**Quotes from the submitted manuscript are given in bold red.**
Proposed amendments or additions to the revised manuscript are given in blue in the Times New Roman font.

References that were already included in the original manuscript are cited in-text only, in the same format as in the submitted manuscript. New references are added to the end of this document in full.

**Referee Comment 1 (Jennifer Arthur)**

RC1: 'Comment on egusphere-2024-1156', Jennifer Arthur (jennifer.arthur@npolar.no), 11 Jun 2024
Citation: https://doi.org/10.5194/egusphere-2024-1156-RC1

**Summary**

This manuscript uses ICESat-2's ATL03 altimetry product to develop two new algorithms which together provide a scalable framework for supraglacial lake detection and depth determination from ATL03 data. Surface melt is an important, yet poorly constrained, component of ice-sheet surface mass balance, leading to surface meltwater accumulating as lakes on ice-shelf surfaces and on grounded ice. In Antarctica, this has been linked to the process of meltwater-driven hydrofracture, which can trigger rapid ice-shelf collapse. Accurately measuring supraglacial lake meltwater depths from satellite data is important due to the challenges in obtaining in situ measurements and is needed for modelling meltwater interactions with ice sheet dynamics. However, few studies have developed automated lake depth estimation methods that are scalable beyond small data subsets, and previous studies rely on methods with poorly constrained parameters and a lack of in situ measurements.
The authors apply their algorithm framework to two regions that experience high surface melt (central west Greenland and the Amery Ice Shelf) and are able to reliably detect lakes where lake bathymetry is visible. The methodology appears robust, and the algorithm performs well even for more complex lakes (especially in Antarctica), including thin, elongated lakes and those with patchy ice cover. The authors found 1249 lakes with their algorithm during four melt seasons and conclude that lake depths agree well with manually-picked lakebeds in ICESat-2 along-track segments.
Overall, it is my view that this study is of broad interest to the cryospheric community as it builds upon previous work focused on supraglacial lake depths on both ice sheets, especially in the context of ice-shelf surface

hydrology and dynamics, by paving the way for developing pan-ice sheet supraglacial lake depth and volume products. I think the surface hydrology and ice shelf communities would be very interested to see this algorithm applied at an ice-sheet scale in future.

In general, this is a very well-written manuscript with detailed methods, clear figures and most of my comments are relatively minor. Once the authors address these, I can therefore recommend that this manuscript is suitable for publication in The Cryosphere.

Please see the attached for specific comments.

Thank you so much Jenny for taking the time to read and review our manuscript, and for your positive, thoughtful and constructive comments. In particular, we believe that your comments were instrumental in better presenting this manuscript within the context of the broader research community's efforts to characterize surface hydrological processes and quantify meltwater on the ice sheets.

**General points**

1. The manuscript could elaborate in a bit more depth about how this algorithm framework approach differs from previous algorithms that have been developed for supraglacial lake depth/bed detection with the same ICESat-2 product (e.g. Datta and Wouters, 2021, who use ICESat-2 ATL03 to derive lake depths and constrain empirically-derived depths from Landsat 8, Sentinel-2, PlanetScope and SkySat imagery). It is clear to me that previous approaches have been tested on small subsets and so are not as scalable as the approach presented here, but I think more detail could be provided for readers.

   For a detailed discussion of this, as well as various explicit proposed changes, please see our response to CC1. We will make it clear in the introduction that:
   - The main "selling point" of our study is that we present a method allowing users to implement supraglacial lake depth estimation from ICESat-2 at scale, and that we took on the task to demonstrate that this works in practice.
   - The motivation behind large-scale implementation are promising case study results (e.g., Datta and Wouters, 2021; Leeuwen, 2023) suggesting that statistically learning the relationship between reflectance in imagery and water depth using ICESat-2 data has great potential to improve imagery-based approaches for continuously monitoring meltwater depths and volumes. For such statistical models to be able to perform well and generalize, we need as much ICESat-2-labeled training data (from a variety of locations, times and environmental conditions) as possible.
   - The fact that FLUID-SuRRF is automated and scalable is not a unique selling point for our study. Other algorithms that claim automation and scalability have been published (e.g. Datta and Wouters, 2021; Xiao et al., 2023). What makes our study unique is the fact that we actually scale it up across two large domains.

   We will also add a comparison of actual depth retrievals between SuRRF, Watta, RTE and a DEM-based method to Section 4.2.2 by including them in Fig. 12 and giving a short description/explanation in the text. We will, however, not go into much detail about this, and instead refer the reader to the method comparison studies of Fricker et al. (2021) and Melling et al. (2024). Fricker et al. includes an earlier version of SuRRF, and all other depth estimates shown in Fig. 12 are reproduced from the two studies mentioned above. For more details on this, see our response to general point 3.

2. 'Lakes with a bathymetric signal' is referred to often throughout the manuscript, and I suggest perhaps the first time this is mentioned adding for clarity: 'lakes with a bathymetric signal, i.e lakes with a visible

or partly-visible lakebed'.

Thank you for the suggestion. We agree that adding this phrase will make it easier for the reader to understand what is meant. We will make the following clarifying changes: (1) change **"a bathymetric signal is discernible"** to "the lakebed is visible or partially visible" in the abstract, (2) change **"signal"** to "return from the lakebed" in the caption of Fig. 1, and (3) add ", i.e. for lakes with a visible or partly visible lakebed" to the last sentence of section 3.1.

3. Performance of the SuRRF algorithm is compared to manual baseline estimates in Section 4.2.2, but I think this could be interesting to elaborate briefly on how the depth estimates are different from results that derive depths using the Radiative Transfer Equation (given that this is a commonly-applied method). For example, does SuRRF tend to detect deeper lakes, implying previous methods underestimate lake depths?

This is a good point, and we should have made this more clear. We will clarify in Sect. 2.2 (where the RTE approach is introduced) by changing the last sentence from **"As a result, it has been found that the RTE approach can significantly over- or underestimate lake depths (Fricker et al, 2021; Melling et. al, 2024)."** to "As a result, it has been shown that the RTE approach can significantly over- or underestimate lake depths in different environments: in Fricker et al. (2021) the method underestimated depths by 30 to 70 % on Amery Ice Shelf in East Antarctica, while in Melling et al. (2024) it overestimated depths by up to 153 % in Southwest Greenland."
As suggested, we will further add a comparison of actual depth retrievals between SuRRF, Watta, RTE methods and a DEM-based method to Section 4.2.2 by including them in Fig. 12 (see below) and giving a short description/explanation in the text. To reflect this, we will rename Section 4.2.2 to "Accuracy of SuRRF depth retrievals and comparison with alternative methods"

[Figure]

**Figure 12.** Comparison between SuRRF water depth estimates, manually annotated ICESat-2 depths and results from other methods for meltwater depth estimation. a)-d): ICESat-2 melt lake segments on the Amery Ice Shelf with manual annotations from Fricker et al. (2021). Other meltwater depth estimates that were reported by Fricker et al. are shown for the Watta algorithm (based on ICESat-2; Datta and Wouters, 2021), for the RTE method applied to the average of Landsat 8's red and panchromatic bands (Spergel et al., 2021) and to Sentinel-2's red band (Moussavi et al., 2020). e)-i): ICESat-2 melt lake segments in Southwest Greenland with manual annotations from Melling et al. (2024). Other meltwater depth estimates that were reported by Melling et al. are shown for the RTE method individually applied to both Sentinel-2's red and green bands, as well as estimates from post-drainage lakebed topography in ArcticDEM (based on Bowling et al., 2019).

We will, however, not go into much detail in the text beyond briefly describing what is shown on the figure, and instead refer the reader to the method comparison studies of Fricker et al. (2021), Melling et al. (2024) and Lutz et al. (2024). Fricker et al. includes an earlier version of SuRRF, and all other depth estimates shown in Fig. 12 are reproduced from either Fricker et al. or Melling et al.

On line 592, we will specify that we are showing results from other methods, but that for in-depth comparisons, we refer the reader to the relevant studies:
"We here use the manual annotations from both of these method comparison studies to evaluate SuRRF's depth estimation performance for all nine lake segments, and to briefly compare SuRRF to other methods whose results were included in previous comparisons (Fig. 12). For a detailed comparison between various ICESat-2 lake depth algorithms (including an earlier version of SuRRF) and RTE-methods, we refer the reader to Fricker et al. (2021). For in-depth discussions comparing manually picked ICESat-2 depths to RTE methods, DEM-based approaches as well as empirical methods using in-situ data, we refer the reader to Melling et al. (2024) and Lutz et al. (2024)."

We will add the following comparison between SuRRF and alternative methods to the end of section 4.2.2 (line 612):

In addition to SuRRF and manual depth estimates, panels a)-d) of Fig. 12 show depth estimates for Antarctic lakes that were reported in Fricker et al. (2021), for the Watta algorithm (based on ICESat-2; Datta and Wouters, 2021) and for the RTE method applied to the average of Landsat 8's red and panchromatic bands (Spergel et al., 2021) and to Sentinel-2's red band (Moussavi et al., 2020). Both SuRRF and Watta track the general shape of lakebed returns in the ATL03 photon clouds well (Pearson's correlation coefficients of R = 0.99 and 0.94, respectively), and and largely agree with manually determined along-track water depths (MAEs of 0.29 m and 0.30 m, respectively). In contrast to SuRRF, Watta appears to have a tendency to overfit where photon density near the lakebed is high with a large elevation spread, resulting in an unreasonably "wiggly" lakebed fit (e.g., Antarctic lake 1, 500-800 m). SuRRF's smoother fit under these conditions is likely due to the fact that it utilizes an adaptive kernel for its robust fit, whose width increases as the number of photons that narrowly cluster around the previous iteration's fit decreases (Sect. 3.3.1). In contrast to SuRRF, Watta also attempts to fit the lakebed across the entire lake basin, even where the lakebed is not visible or indistinguishable from noise, which can sometimes result in arbitrary, unrealistic depth estimates (e.g., Antarctic lake 1, around 450 m). Under such conditions SuRRF assigns a low confidence score to the lakebed fit and discards associated depth estimates to prevent arbitrary results. However, in some cases this results in SuRRF discarding depth estimates where Watta appears to fit the lakebed reasonably well (e.g., Antarctic lake 3, 250-300 m). The RTE approach based on Landsat 8's red/panchromatic band average consistently underestimates water depths, reporting a total amount of water that is ~ 73 % lower than the manual baseline. The RTE approach based on Sentinel-2's red band also underestimates water depths, and reports a total amount of water that is 34 % lower than the manual baseline.

Panels e)-i) of Fig. 12 also show non-ICESat-2 depth estimates for Greenland lakes that were reported in Melling et al. (2024), for the RTE method individually applied to both Sentinel-2's red and green bands, as well as estimates from post-drainage lakebed topography in ArcticDEM (based on Bowling et al. 2019). The RTE approach based on Sentinel-2's red band generally underestimates depths and reports a total amount of water that is 42 % lower than the manual baseline (similar to this method's performance over Antarctic lakes). In contrast, the RTE approach based on Sentinel-2's green band generally overestimates depths and reports a total amount of water that is 34 % larger than the manual baseline. However, Melling et al. also note that when using values of tuneable parameters that have been commonly used in the literature (Sneed and Hamilton, 2007; Georgiou et al., 2009; Pope et al., 2016), the RTE approach for Sentinel-2's green band overestimates lake depths even more, which results in reporting a total amount of water that is 84 % larger the manual baseline, with individual depths being overestimated by up to 153 %. This implies that RTE-based methods, while being popular for their simplicity, can potentially result in highly inaccurate meltwater volume estimates. The depth estimates derived from DEMs of emptied lake basins match the ICESat-2 manual baseline reasonably well, and when compared with it underestimate the total amount of water by 6 % with a MAE of 0.34 m. Since this method's performance is comparable to that of ICESat-2-based methods, this implies that DEM-based methods could be used supplement ICESat-2 depth measurements for labeling reflectance in passive optical imagery with supraglacial water depths, at least on the Greenland Ice Sheet where melt lakes on grounded ice drain regularly (Johansson et al., 2013).

4. For those who are less familiar with distributed High-Throughput Computing, how widely useable is this algorithm for others to whom the OSG Open Science Pool is not accessible, aka. non-US-based researchers?

Our Python code works without the need for using any OSG services: it can be run on a local computer or on any computing cluster (locally or in the cloud) in the provided singularity container. We used the OSG Open Science Pool because it provided us with free computational infrastructure. We do not have the resources to scale this method up to ice-sheet-wide implementation on commercial cloud

computing platforms at a feasible price point, and during initial development PA had no access to NASA-provided computational resources due to the COVID19 pandemic preventing in-person fingerprinting of non-US citizens. We believe that the OSG Open Science Pool is the most accessible option for large-scale implementation, but there are unfortunately some limitations to who can use it. As a non US-based researcher, one could gain access to the OSG Open Science Pool if collaborating with a US-based researcher or an institution that operates its own access point. Since we want our methods and data to be accessible and encourage others to use it in their own research, we will explain this in more detail in the Code and data availability section:

"The FLUID/SuRRF source code is freely available at https://doi.org/10.5281/zenodo.10905941 (Arndt and Fricker, 2024a). To execute this code, users need to create a free NASA Earthdata login for ICESat-2 data access. The source code contains a singularity container in which this version of FLUID/SuRRF can be executed. The main Python script detect_lakes.py can be run either locally on any individual ATL03 granule, or on many granules in parallel on any computing cluster that supports the specified computing environment or the use of singularity containers. In this study we present our implementation of FLUID/SuRRF on the OSG Open Science Pool because it provided us with free computational infrastructure. Due to funding mandates, free access to the OSG Open Science Pool is limited to researchers contributing to a US-based project at an academic, government, or non-profit organization, or researchers affiliated with any project or institution that operates its own local access point. This means that to implement FLUID/SuRRF on the OSG Open Science Pool as described here, you need to have at least one collaborator on your team to whom the above criteria apply. This collaborator can register your project with OSG on the Open Science Pool. Then, anyone contributing to the project can register for an account on OSG Connect to gain access to the Open Science Pool. For more information, see https://osg-htc.org/services/open_science_pool.html and https://osg-htc.org/about/organization. More information is also included in the README file."

**Specific comments**

- L30: I would cite something more relevant than this EGU abstract here, e.g. Gilbert and Kittel (2024), https://doi.org/10.1029/2020GL091733.
  We will replace the citation with the suggested one.

- L34: Perhaps add a sentence or two here about what direct observations of supraglacial lake depths do exist, particularly in Antarctica (to highlight the paucity of observations).
  This is a good suggestion, and we will briefly mention this here:
  "However, there are few direct in-situ observations of supraglacial lake depths (none in Antarctica, and ten lakes up to 11.5 m deep in Greenland), which leads to errors in total water volume estimates."
  We will elaborate on this further in Sect. 2.2, where we introduce empirical models derived from regression of in-situ depth measurements with optical imagery (L109, see comment below).

- L109: add where these in situ measurements were collected (west Greenland).
  We include a description of where the in-situ measurements were collected, as suggested. This does not only include West Greenland, but also South-East and North-East Greenland. Based on the previous comment, we also reiterate that no observations are available for Antarctica. See the updated paragraph below:
  "Another approach to estimating lake depths is using **empirical models** derived from regression of in-situ depth measurements with optical imagery (Tedesco and Steiner, 2011; Legleiter et al., 2014; Pope et al., 2016; Lutz et al., 2024). However, in-situ measurements of supraglacial lake depths are very sparse, with – to the best of our knowledge and at the time of writing – no such data is available for Antarctica, and data available for only ten lakes up to 11.5 m deep on the Greenland Ice Sheet between 2005 and 2024: Box and Ski (2007) sampled two

lakes on Jakobshavn Isbræ and Sermeq Avannarleq in 2005, Sneed and Hamilton (2007) sampled one lake on Helheim Glacier in 2008, Tedesco and Steiner (2011) sampled one lake in Central West Greenland in 2010, Legleiter et al. (2014) sampled three supraglacial water bodies on Isunnguata Sermia and Russell Glacier in 2012 and Lutz et al. (2024) sampled three lakes on Zachariæ Isstrøm in 2022. This makes the observations provided by Lutz et al. the only in-situ depth data of supraglacial lakes that overlap with the Landsat 8 and Sentinel-2 missions. Further, it has been shown that the relationship between water depth and reflectance values in optical imagery can vary significantly by geographical region (Lutz et al., 2024). Thus, the regression coefficients of these empirical models are limited to the spatial area of the original in-situ measurements, making them impractical for application on a larger, ice-sheet-wide scale."

For reference, for a full list of in-situ measurements of supraglacial lake depth available in the literature, see the table below:

| source | lon | lat | date | IMBIE_ basin | IMBIE_name | depth_min | depth_max | n_obs | accuracy _m | instrument |
|---|---|---|---|---|---|---|---|---|---|---|
| Box and Ski (2007) | -49.1254 | 69.5345 | 2005-08-14 | CW | JAKOBSHAVN_ISBRAE | 1.05 | 11.5 | 335 | 0.1 | Garmin Fish Finder 100 |
| Box and Ski (2007) | -49.5345 | 69.4891 | 2005-08-15 | CW | SERMEQ_AVANNARLEQ | 1.25 | 10 | 338 | 0.1 | Garmin Fish Finder 100 |
| Sneed and Hamilton (2011) | -38.45 | 66.46145 | 2008-07-11 | SE | HELHEIMGLETSCHER | 0.7 | 3 | 21 | 0.03 | Satlantic Profiler II |
| Tedesco and Steiner (2011) | -49.4944 | 69.60972 | 2010-07-02 to 05 | CW | CW_NONAME1 | n/a | 4.5 | 6000 | 0.25 | HDS-5 Lowrance |
| Legleiter et al. (2014) | -48.322 | 67.11825 | 2012-07-20 | SW | ISUNNGUATA-RUSSELL | n/a | 3.16 | 3264 | 0.025 | Ohmex SonarMite 235KHz |
| Legleiter et al. (2014) | -48.1008 | 66.9025 | 2012-07-21 | SW | SAQQAP-MAJORQAQ-S OUTHTERRUSSEL_SOU THQUARUSSEL | n/a | 10.45 | 4383 | 0.025 | Ohmex SonarMite 235KHz |
| Legleiter et al. (2014) | -48.7649 | 67.18042 | 2012-07-23 | SW | ISUNNGUATA-RUSSELL | n/a | 1.66 | 1164 | 0.025 | Ohmex SonarMite 235KHz |
| Lutz et al. (2024) | -21.8182 | 78.92271 | 2022-07-09 | NE | ZACHARIAE_ISSTROM | 0.41 | 8.21 | 2129 | 0.2 | Lawrence Elite 7 FS |
| Lutz et al. (2024) | -21.9217 | 78.90026 | 2022-07-03 | NE | ZACHARIAE_ISSTROM | 0.46 | 6.78 | 981 | 0.2 | Lawrence Elite 7 FS |
| Lutz et al. (2024) | -21.9373 | 78.87014 | 2022-07-09 | NE | ZACHARIAE_ISSTROM | 0.39 | 7.32 | 2991 | 0.2 | Lawrence Elite 7 FS |

(some of the information for Lutz et al. is from private communication, but we were told that their dataset should be up on PANGAEA soon)

- L484: I would rename the labels 'CW < 200 m' and 'B-C < 1000 m' to 'Surface Elevation < 200 m' and 'Surface Elevation < 1000 m' for clarity, or else clarify this in the figure caption.
  We will follow this suggestion. See the updated figure below.

[Figure]

- L543: could also add here that lakes typically are advected downstream year-on-year (e.g. Arthur et al., 2020).
  We will follow this suggestion and add the following at the end of the sentence on L543: ", with existing basins typically being advected to locations significantly further downstream from one melt season to the next (Arthur et al., 2020)."

- L579-585: This explanation for differences in lake elevation-depth relationships could be better cited. For example, Banwell and MacAyeal (2015) highlight lake deepening by lake-bottom ablation.
  We agree that this section should have been better supported by existing literature. We will reference Lampkin and VanderBerg (2011) and Tedesco et al., (2012) to support the claim that fixed lakebed locations controlled by bedrock topography and lake bottom ablation cause deep lakes in Greenland. We will reference Bell et al. (2018) as a reference for arguing that lakes on grounded ice in Antarctica are more prevalent under warmer conditions and behave similarly to lakes on grounded ice in Greenland. We will also add two new references:
  - Glen et al. (2024) as a reference supporting that there are more small, shallow and slushy lakes in Greenland in warm summers than in cooler summers.
  - Banwell et al. (2014) as a reference supporting the claim that lakes which form on the fairly flat topography of Antarctic ice shelves tend to be more shallow than those on grounded ice on the Greenland Ice Sheet.

- L613: I'm not sure about the overall relevance of Section 4.3 and don't think it adds substantially to the manuscript, because the algorithm application to ocean bathymetry, inland waters or sea ice melt ponds is less relevant in the context of this manuscript. I would suggest removing this section and moving the content from 'The ephemeral nature of supraglacial lakes…' to further up in the introduction as justification for your algorithm.
  ICESat-2 is an interdisciplinary instrument where scientists from different fields have worked together to improve algorithms that apply to 'cross-cutting' themes, such as bathymetry. In that spirit, we consider the section to be highly relevant in the context of this largely methods-focused manuscript, because

method development for ICESat-2 water depth estimation benefits from cross-discipline collaboration between scientists who work on using the satellite's bathymetric capabilities in different environments (Parrish et al., 2022). As such, Section 4.3 is meant to (1) give credit to some of the work that is not directly concerned with supraglacial lakes but has inspired our own work and (2) encourage readers with an interest in ICESat-2-specific methods for supraglacial lake depth estimation to draw inspiration from the broader related literature themselves.

However, this comment has made us realize that we did not justify this well in Section 4.3, and therefore suggest the following change:
"Beyond estimating the depth of supraglacial lakes, ICESat-2's bathymetric capabilities have been used for various other applications. Many algorithms employed for depth retrieval from ATL03 share significant similarities, enabling method development for ICESat-2-derived bathymetry to benefit from broader cross-discipline collaboration (Parrish et al., 2022). Methods similar to FLUID-SuRRF have been used for [...]. However, there are also notable differences between bathymetric applications of ICESat-2 in different environments that have led to the development of specialized approaches, in particular for their large-scale implementation. For nearshore and inland bathymetry applications [...]"

- L630: I don't entirely agree with the part of this sentence that states the calculated water depths prevent the calculation of lake volumes. Surely it would be possible to calculate lake volumes by combining the ICESat-2 derived depths with lake extents derived from optical imagery (Landsat 8) as an initial estimate? I do understand though that with the small dataset you present here it is not enough to track the meltwater through the melt season.

Thank you for catching this! We meant to say that the ICESat-2 observations by themselves are not enough to calculate lake volumes and therefore need to be combined with other types of data such as imagery to achieve this goal. We will change the sentence to "Our ICESat-2-derived water depths are the first comprehensive dataset of supraglacial lake depths that were directly measured from a satellite; however, these along-track observations alone are too sparsely spaced…". We will also add a reference to ICESat-2 in the next sentence to make this more clear: "However, the large volume and wide variety of data that our method provides implies that the ICESat-2 based depth measurements presented here could be used to…"

**Technical/minor corrections**

- In some places the Surface Removal and Robust Fit algorithm is referred to as SuRRF and in others as SuRFF, so check throughout for consistency.
  Thank you for catching that! We will replace 14 instances where we accidentally used the abbreviation **"SuRFF"** with "SuRRF".

- L178: delete 'each' (same on L653).
  We will delete **"each"** in both instances, as suggested.

- L611: Is a word missing here? '.. in between our (deeper) SuRRF estimates and the (more shallow) manual baseline estimates'.
  Yes, thanks for catching that. We will add "and", as suggested.

- L640: don't hyphenate 'well-enough'.
  We will remove the hyphen, as suggested.

- L710: it's origin → its origin
  We will remove the apostrophe.

**References:**

- Bowling, J. S., Livingstone, S. J., Sole, A. J., & Chu, W. (2019). Distribution and dynamics of Greenland subglacial lakes. Nature communications, 10(1), 2810.

- Banwell, A. F., Caballero, M., Arnold, N. S., Glasser, N. F., Mac Cathles, L., & MacAYEAL, D. R. (2014). Supraglacial lakes on the Larsen B ice shelf, Antarctica, and at Paakitsoq, West Greenland: a comparative study. Annals of Glaciology, 55(66), 1-8.

- Georgiou, S., Shepherd, A., McMillan, M., & Nienow, P. (2009). Seasonal evolution of supraglacial lake volume from ASTER imagery. Annals of Glaciology, 50(52), 95-100.

- Glen, E., Leeson, A. A., Banwell, A. F., Maddalena, J., Corr, D., Noël, B., & McMillan, M. (2024). A comparison of supraglacial meltwater features throughout contrasting melt seasons: Southwest Greenland. EGUsphere, 2024, 1-31.

- Johansson, A. M., Jansson, P., & Brown, I. A. (2013). Spatial and temporal variations in lakes on the Greenland Ice Sheet. Journal of hydrology, 476, 314-320.

- Lutz, K., Bever, L., Sommer, C., Humbert, A., Scheinert, M., & Braun, M. (2024). Assessing supraglacial lake depth using ICESat-2, Sentinel-2, TanDEM-X, and in situ sonar measurements over Northeast Greenland. EGUsphere, 2024, 1-25.

- Parrish, C. E., Magruder, L., Herzfeld, U., Thomas, N., Markel, J., Jasinski, M., ... & Caballero, I. (2022, October). ICESat-2 bathymetry: Advances in methods and science. In OCEANS 2022, Hampton Roads (pp. 1-6). IEEE.

---

## Author Comment (AC2)

**Response to RC2**

A Framework for Automated Supraglacial Lake Detection and Depth Retrieval in ICESat-2 Photon Data Across the Greenland and Antarctic Ice Sheets

Philipp Sebastian Arndt[1] and Helen Amanda Fricker[1]

[1] Scripps Polar Center, University of California San Diego, 8885 Biological Grade, La Jolla, CA 92037, USA

Correspondence:
Philipp Sebastian Arndt (parndt@ucsd.edu)

Discussion: https://doi.org/10.5194/egusphere-2024-1156

Comments from the reviewers are given in black.
Our responses are given in red.
**Quotes from the submitted manuscript are given in bold red.**
Proposed amendments or additions to the revised manuscript are given in blue in the Times New Roman font.

References that were already included in the original manuscript are cited in-text only, in the same format as in the submitted manuscript. New references are added to the end of this document in full.

**Referee Comment 2 (Ian Brown)**

RC2: 'Comment on egusphere-2024-1156', Ian Brown, 17 Jun 2024
Citation: https://doi.org/10.5194/egusphere-2024-1156-RC2

**Summary**

This is an important contribution to our ability to monitor the supra-glacial hydrology of the ice sheets. It is a very well executed investigation and a well written manuscript. I have some small concerns regarding validation and some suggestions for minor edits.

Thank you so much Ian for taking the time to read and review our manuscript, and for your positive, thoughtful and constructive comments. In particular, we believe that your suggestions regarding a clearer distinction between ICESat-2 lake segments and unique "real" supraglacial lakes have helped to make the manuscript more clear, especially to readers who are not already very familiar with ICESat-2 data.

**Specific comments**

- The first sentence sets the tone so it is a little odd to address ice shelf collapse when that is not the focus of the article. Consider refocusing the opening sentence.
  We propose adding an opening that specifically addresses the difficulty of monitoring the water depths of supraglacial lakes on the ice sheets, then continue with arguing why knowing water depths is important: "Water depths of supraglacial lakes on the ice sheets are difficult to monitor continuously due the lakes' ephemeral nature and inaccessible locations. Supraglacial lakes have been linked to ice shelf collapse in Antarctica and accelerated flow of grounded ice in Greenland. However, the impact of supraglacial lakes on ice

dynamics has not been quantified accurately enough to predict their contribution to future mass loss and sea level rise. This is largely because ice-sheet-wide assessments of meltwater volumes rely on models that are poorly constrained due to a lack of accurate depth measurements."

- Line 38. "Recently" is relative. ICESAT-2 has been in orbit for 5 years and readers may access the article in a decade meaning "recently" is not appropriate. Delete the word.
  Good point, and in fact it is now nearly 6 years. We will change this to: "Launched in 2018, NASA's Ice, Cloud and land Elevation Satellite (ICESat-2)..."

- Line 205-206. Can you describe in more detail the empirical observations: how many were used to establish the thresholds?
  We admit that the phrasing of this was confusing in the submitted manuscript. What we meant to convey is that we established these thresholds based on trial-and-error testing on a number of hand-picked granules that we judged to be likely representative of various possible environments. For clarification, we propose to change this part to the following:
  "Based on these assumptions, and using a trial-and-error approach, we defined the following thresholds on the density ratios that need to hold for a major frame to pass the flatness check: $d_0/d_1 \geq 2$, $d_0/d_2 \geq 5$, $d_0/d_3 \geq 10$ and $d_0/d_4 \geq 100$. As part of this trial-and-error approach, we manually assessed the effects of tweaking the above thresholds on a number of hand-picked granules, which we judged to be likely representative of various possible environments, to ensure adequate performance (i.e. granules without surface melt vs. pervasive surface melt, granules with smooth vs. rough background topography, granules containing ice-covered and partially ice-covered lakes, granules containing slush areas, granules containing exposed bedrock, partially cloudy granules, weak vs. strong beam data, night- vs. daytime acquisitions, etc.)."

- Line 227. Does FLUID work as well in the presence of wind roughened water surfaces: I assume it does though perhaps fewer photons penetrate the surface. Please comment (as you specifically address the impact of wind on optical estimates of water depth on line 105).
  We propose to re-write this part to include a discussion of the effects of wind, as suggested:
  "Afterpulses only become noticeable when the sensor is nearly or fully saturated, which means they often appear in ATL03 data over supraglacial lakes because smooth open water surfaces (i.e. the surface of stationary water bodies that are not affected by wind) can result in specular reflection. This suggests that the presence of wind ripples increases the likelihood of detecting a lake with a clear bathymetric signal in ATL03 data by preventing sensor saturation and afterpulsing (Lu et al., 2019; Tilling et al., 2020) and also explains why we observe afterpulsing more frequently near the (more wind-shielded) margins of melt lakes than over their (more wind-exposed) interior."

- Line 251. c is presumably the speed of light in a vacuum or freshwater (line 434)? Please clarify.
  Thanks for catching this! c is the speed of light in a vacuum, which is the value that is used to calculate photon elevations from each photon's time of flight. We will clarify this by adding the following to the end of this sentence: ", where c is the speed of light in a vacuum."

- Line 321. Define "a few".
  We agree that we should be more specific here. The phrase "along-track extent" was poorly chosen. The 10 in Eq. (3) is what describes this: the maximum separation between two clusters to be merged is 10 major frames, so clusters that are separated by more than 1.4-1.6 km are not merged. To clarify this we propose to replace (ii) on line 321 with:
  "(ii) a ground track rarely crosses the same lake in two distinct locations that are separated by more than about 1.5 km."
  We will further explain equations (2) and (3) in plain language after they are defined:

"Equation (2) states that neighboring clusters are only merged if their respective lake surface elevations are within 0.1 m of each other, and Eq. (3) further states that neighboring clusters are only merged if they are separated by ten major frames that did not pass the bathymetry check, or less (about 1.5 km). This means that if FLUID encounters the unlikely but possible scenario in which a ground track crosses two arms of the same lake, which are separated in along-track distance by more than ten major frames, then these two crossings are considered to be separate lake segments and returned as two separate files in the output data rather than being merged together into one lake segment. If these two conditions do not result in…"

- Line 343-345. It would be useful to estimate the real number of lakes mapped if possible. If it is not possible please discuss this in the appropriate section of the manuscript.
  This is a great suggestion. We will add a row to Table 1 that shows the estimated number of unique, contiguous melt lakes that are sampled by our ICESat-2 lake segments:

Table 1. Summary statistics for the ICESat-2  lake segments extracted by FLUID/SuRRF for our regions and melt seasons of interest.

|  | Amery catchment (B-C) | | Central West Greenland (CW) | |
|---|---|---|---|---|
|  melt season | 2018-19 | 2020-21 | 2019 | 2020 |
|  amount of surface melt | high | very low | high | low |
|  area of Landsat 8 maximum melt extent | $1872\,\mathrm{km}^2$ | $100\,\mathrm{km}^2$ | $1127\,\mathrm{km}^2$ | $431\,\mathrm{km}^2$ |
| number of total ICESat-2 lake segments | 721 | 28 | 325 | 175 |
|  number of unique lakes sampled | 385 | 25 | 198 | 114 |
| number of high-quality lake segments | 165 | 5 | 196 | 109 |
|  percentage of high-quality segments | 23% | 18% | 60% | 62% |
| median lake segment depth | 1.85 m | 1.48 m | 2.77 m | 3.43 m |
| maximum lake segment depth | 10.4 m | 17.3 m | 25.8 m | 15.1 m |

We describe this in the text at the end of the first paragraph of section 4.1.1 (Line 612-):
"To estimate how many unique supraglacial lakes were sampled by these detected ICESat-2 lake segments during each melt season, we calculated the maximum surface meltwater extent for each of the melt seasons independently using Landsat 8 imagery, based on the methods detailed in Tuckett et al. (2021) (blue regions in Fig. 9 and 10). We then matched each detected ICESat-2 lake segment to a lake basin in these imagery-based melt extents and counted the number of total basins that were sampled by at least one ICESat-2 lake segment (see supplemental maps; Arndt and Fricker 2024c). Over Central West Greenland, this resulted in 196 unique supraglacial lakes being sampled by our data in 2019, and 109 lakes in 2020. Over the Amery Catchment, FLUID-SuRRF segments sampled 165 unique melt lakes in 2018-19 and 25 lakes in 2020-21."

We further explain the difference between ICESat-2 lake segments and the individual "real" lakes that they sample in Section 3.2.5 (FLUID step 5: along-track aggregation of lake segments), see our comment about this section further below

- Figure 8., Line 483. Consider moving figure 8 to the front of the manuscript, for example, at the end of the Introduction where you cite the study areas.
  We will move the figure to the end of the introduction, as suggested. We will change the sentence in the introduction that refers to the study regions to include a reference to the figure, and to the detailed "Study regions and time span" section: "Here, we present this algorithm, demonstrate its performance over two drainage basins in Greenland and Antarctica (Sect. 3.5, Fig. 2), and provide a framework for its large-scale implementation using distributed high-throughput computing."

- Section 4.1.1 (line 529-). The number of lakes detected over West Greenland is very small compared with other studies. Especially considering you measure over a season. Also, it is odd that so few lakes

are detected in 2020-21 over the Amery ice shelf. The number of false negatives is presumably very high (i.e. your detection rate is low). Presumably this is a function of the ground track spacing of ICESAT-2. I think it is important to discuss this and the impact it would have on operational implementation of your algorithm (or the limits to that).

It seems like there is some confusion about ICESat-2 data coverage here, likely because we have not made it clear enough in our writing that ICESat-2 can by no means sample all of the supraglacial lakes – or even close to all of them. There are other (image-based) methods out there that are much better-suited at finding the locations of lakes and determining their horizontal extents. What makes ICESat-2 unique is its capability to make direct and accurate measurements of water depth from space. So we cannot find all supraglacial lakes with ICESat-2, but for the (comparatively few, but many more than we could measure in-sit) that we do find with ICESat-2 we now have depth measurements.

The reason that there were so few lakes detected in 2020-21 over the Amery Ice Shelf is that there was barely any surface melt in this region during that melt season (as shown in Fig. 10). We also included the total area of Landsat 8-based melt entents to Table 1 (see above) to make this more clear, and will change the description in the row titled "amount of surface melt" to "very low" for this melt season.

- We propose to rewrite part of the introduction, to make it clear that a data set based on ICESat-2 alone cannot find all the supraglacial lakes, and that we don't intend to claim that it can: "While ICESat-2 has the unique capability to make direct and accurate measurements of water depth from space, the coverage of the mission's photon-level data product (ATL03) is limited to discrete, one-dimensional ground tracks that are coarsely spaced on the Earth's surface (~9.9 km between neighboring reference tracks and ~3.3 km between all neighboring beam pair tracks at 70°N/S) and have a relatively long repeat period of three months. This means that no supraglacial lake depth data product derived from ICESat-2 alone is able to provide samples of all (or even nearly all) supraglacial lakes on the ice sheets: ICESat-2's track spacing means that the majority of lakes form in locations that ICESat-2 ground tracks never sample, and the three-month return period means that for a significant number of tracks ICESat-2 never passes over at the time at which melt lakes are visible. ICESat-2 is also unable to penetrate optically thick clouds, thus further limiting the amount of data available for water depth measurements. While ICESat-2 data alone cannot be used to continuously monitor melt lake volumes,..."

- We will replace **"lake"** with "lake segment" or similar, in all instances where we simply refer to a single-ground-track segment of ATL03 data with visible supraglacial lake bathymetry. In particular, we further clarify this in Sect. 3.2.5 (FLUID step 5: along-track aggregation of lake segments): "The resulting final clustering is now considered the set of ICESat-2 supraglacial lake segments that have been found on each ground track. Note that for simplicity we here use the term "ICESat-2 lake segment" (or simply "lake segment") to refer to any single-ground-track segment of ATL03 data with visible bathymetry from one supraglacial lake. If multiple ICESat-2 ground tracks contain data from the same supraglacial lake, the distinct ground track segments are still considered different "ICESat-2 lake segments" for the purpose of this algorithm. For example, the two ATL03 profiles acquired by the two neighboring ground tracks of the center beam pair shown in Fig. 1 would be considered two distinct "lake segments" despite ICESat-2 having acquired their underlying data during the same overpass and from the same supraglacial lake. Since multiple ICESat-2 lake segments can be associated with the same supraglacial lake, this means that the total number of unique supraglacial lakes sampled by ICESat-2 is smaller than the total number of supraglacial lake segments reported by FLUID-SuRRF (Sect. 4.1, Table 1)."

- In line 529, we will also replace **"Out of the 1249 supraglacial lakes that we detected in total,"** with "Out of the 1249 supraglacial lake segments that we detected in the ICESat-2 data analyzed in this study,..." to emphasize that our results are valid only for lakes that are visible to ICESat-2.

- - We will further emphasize track spacing in Sect. 3.5 (Study regions and time span) by adding the following: "Our two study areas cover latitudes from 68.2°N to 72.1°N in Greenland and latitudes from 68.4°S to 74.0°S in Antarctica, meaning that ICESat-2 track spacing is similar over the two regions: in Central West Greenland RGT spacing varies from ~8.8 km in the north to ~10.8 km in the south; over the Amery Catchment RGT spacing varies from ~7.9 km in the south to ~10.7 km in the north."

- Line 550-551. Did you consider identifying lakes that have emptied. Johansson et al., (2013; J. Hydrol. 476) show that many lakes are transient and will empty late in season. This would allow you to evaluate the accuracy of the estimate from filled-emptied conditions.
  We did consider this, but came to the conclusion that due to the many complexities involved in comparing repeat tracks before and after lake drainage this is outside the scope of this study. We propose adding a short paragraph explaining this after L630 to the end of Sect. 4.2.2 ("Accuracy of SuRRF depth retrievals"):

  "Since many lakes on the Greenland Ice Sheet are transient and drain late in season (Johansson et al., 2013), it could be possible to obtain independent ICESat-2-based meltwater depth estimates by comparing "full vs empty" repeat-track measurements along ICESat-2 lake segments before and after the drainage. However, this approach would suffer from many of the same drawbacks that affect depth estimation from DEMs of a lake's bed topography that were acquired after it drained (Sect. 2.2), for example: effects of lake-bottom ablation, surface elevation change from precipitation and blowing snow deposits, as well as across-track advection of surface topographical features. Furthermore, this approach would not be feasible in Antarctica, where lake drainage is very rare, in particular on grounded ice. In cases where melt lake drainage is observed on the floating ice shelves, obtaining water depth from repeat-track elevation change is not possible due to the advection of surface topography with the ice flow and post-drainage flexural rebound (Warner et al., 2021). Due to these complexities, we do not attempt to validate ICESat-2 lake depth measurements using this method."

- Line 657. I think you need to mention there is a bias towards over-estimation.
  This is a good call. We propose adding the following caveat to the end of this section, combined with noting that ground truth observations are desperately needed to validate depth estimates and quantify potential biases in the data:
  "Since in the absence of ground truth validation data our depth validation efforts were based on manual annotation of the data, we acknowledge that there may be a small but potentially significant bias towards overestimating water depths with FLUID-SuRRF. This highlights an urgent need for ground truth *in situ* water depth measurements of supraglacial lakes that coincide with ICESat-2 overpasses, to enable calibration and validation of depth estimates."
  In addition, we add a description of a correction that can be applied to SuRRF depth estimates to attempt correcting for the effects of multiple scattering in the water column (Sect. 3.3.3 "SuRRF step 2: lakebed fit", line 418):
  "Previous studies have hypothesized that ICESat-2-based depth retrieval algorithms placing the lakebed fit at the along-track elevation of highest subsurface photon density may be biased towards slightly overestimating total water depths due to multiple scattering within the water column (Fricker et al., 2021; Xiao et al., 2023). To address this, we provide an optional correction, which places the lakebed fit at a higher elevation where the initial SuRRF lakebed fit included photons further below the initial lakebed fit than would be expected from bathymetric signal photons. To achieve this, we remove any photons located at a vertical distance below the initial SuRRF lakebed fit by more than the sum of (1) ICESat-2's single-photon time-of-flight precision (~ 12 cm in ATL03 photon heights or 800 ps; Markus et al., 2017), and (2) the elevation range within ICESat-2's footprint diameter (~ 11 m Magruder et al., 2021a) obtained by projecting the footprint onto the along-track lakebed topography estimated by the initial SuRRF lakebed fit. We then reapply the lakebed fit to the remaining photons as described

above, while supplying the SuRRF Robust Fit (Sect. 3.3.1) with the uncorrected SuRRF lakebed fit as the initial guess. Since the presence or magnitude of this hypothesized overestimation of water depths cannot be established without any ground truth in situ data available along any ICESat-2 lake segments, we provide this scattering correction for reference only, and do not apply it to the water depths presented in this study. If such validation data becomes available in the future, our scattering correction can be tuned to better match observations, and can be readily applied to FLUID-SuRRF output data."

We then further discuss how applying this correction would change water depth estimates, and how they compare to the manually annotated baseline estimates for validation (4.2.2 Accuracy of SuRRF depth retrievals and comparison with alternative methods, line 598):

"When applying the correction for multiple scattering (Sect. 3.3.3) to SuRRF depth estimates, the average bias is reduced to 0.07 m deeper than the manually picked values, with a mean absolute error (MAE) of 0.15 m and a Pearson's correlation coefficient of R = 0.993. This results in the scattering-corrected version of720 SuRRF reporting a total amount of water that is 3 % larger than the estimate given by the manual baseline."

We then refer back to the scattering correction later in this section when discussion the disparity between manual altimetry expert picks and algorithmic fits to the ICESat-2 data (line 611):

**"We therefore believe that the true water depth falls somewhere in between our (deeper) SuRRF estimates and the (more shallow) manual baseline estimates from Fricker et al. (2021) and Melling et al. (2024).** "Our scattering correction to SuRRF depth estimates is an attempt to reconcile this disparity between depth estimates. However, in the absence of ground truth *in situ* validation data for ICESat-2 lake segments, the correct magnitude of this correction remains unknown." **This demonstrates an urgent need for in situ meltwater depth data that can be used to reliably validate the accuracy of ICESat-2 estimates."**

- Line 660. I do not think this is demonstrated given the very low numbers of detections and the fact that you can't identify whether multiple measurement lines are from the same lake.
  Given that past empirical models (e.g., Pope et al., 2016) were based on in-situ measurements of a small portion of a single supraglacial lake, and that Datta et al. (2021) were quite successful at empirically estimating lake depths from imagery using a much smaller amount of ICESat-2 data, we believe that the general sentiment of this statement is justified. However, we acknowledge that our claim was phrased quite strongly. In particular, we propose to replace **"imply"** with "suggest" to acknowledge that we have not directly demonstrated this. In addition, we propose changing **"...that the results presented here could be used to…"** with "...that ICESat-2-based depth measurements obtained from applying FLUID-SuRRF at ice-sheet-wide scale could be used to…".

**References**

- Johansson, A. M., Jansson, P., & Brown, I. A. (2013). Spatial and temporal variations in lakes on the Greenland Ice Sheet. Journal of hydrology, 476, 314-320.

- Lu, X., Hu, Y., & Yang, Y. (2019, December). Ocean subsurface study from ICESat-2 mission. In 2019 photonics & electromagnetics research symposium-fall (PIERS-fall) (pp. 910-918). IEEE.

**Maps of ICESat-2 lake segments and unique supraglacial lake extents**

Reviewer comments about distinguishing "unique supraglacial lakes" from distinct "ICESat-2 melt lake segments" prompted us to estimate which distinct melt lakes are sampled by FLUID lake segments more than once. We report the number of unique supraglacial lakes in the text of the manuscript and Table 1, and explain how we estimated the given numbers. In a new version of the supplement at https://doi.org/10.5281/zenodo.10901826 (Arndt and Fricker, 2024c), we will additionally provide high-resolution maps of the ground tracks of FLUID lake segments over Landsat 8 maximum melt extents, with unique lakes identified by their borders. These maps can be used to better understand the spatial distribution of FLUID melt lake segments and how they relate to the overall distribution of pooled meltwater over our study regions. Due to their large scale, these maps cannot be reasonably included in the main manuscript. Lower-resolution reproductions of these maps are shown on the following pages.

**Central West Greenland: 2019**

[Figure]

325 ICESat-2 ground track segments over lakes
198 unique melt lakes sampled

ICESat-2 lake segments overlapping
with Landsat 8 melt extent

ICESat-2 lake segments not overlapping
with Landsat 8 melt extent

Landsat 8 melt extent
polygons with ICESat-2 data

Landsat 8 melt extent
polygons without ICESat-2 data

outlines
of unique
supraglacial
lakes

**Central West Greenland: 2020**

175 ICESat-2 ground track segments over lakes
114 unique melt lakes sampled

[Figure]

Legend:
- ICESat-2 lake segments overlapping with Landsat 8 melt extent
- ICESat-2 lake segments not overlapping with Landsat 8 melt extent
- Landsat 8 melt extent polygons with ICESat-2 data
- Landsat 8 melt extent polygons without ICESat-2 data
- outlines of unique supraglacial lakes

**Amery Catchment: 2018-19**

[Figure]

720 ICESat-2 ground track segments over lakes
385 unique melt lakes sampled

ICESat-2 lake segments overlapping
with Landsat 8 melt extent

ICESat-2 lake segments not overlapping
with Landsat 8 melt extent

Landsat 8 melt extent
polygons with ICESat-2 data

Landsat 8 melt extent
polygons without ICESat-2 data

outlines
of unique
supraglacial
lakes

**Amery Catchment: 2020-21**

[Figure]

28 ICESat-2 ground track segments over lakes
25 unique melt lakes sampled

ICESat-2 lake segments overlapping
with Landsat 8 melt extent

ICESat-2 lake segments not overlapping
with Landsat 8 melt extent

Landsat 8 melt extent
polygons with ICESat-2 data

Landsat 8 melt extent
polygons without ICESat-2 data

outlines
of unique
supraglacial
lakes

---

## Author Comment (AC3)

**Response to RC3**

A Framework for Automated Supraglacial Lake Detection and Depth Retrieval in ICESat-2 Photon Data Across the Greenland and Antarctic Ice Sheets

Philipp Sebastian Arndt[1] and Helen Amanda Fricker[1]

[1] Scripps Polar Center, University of California San Diego, 8885 Biological Grade, La Jolla, CA 92037, USA

Correspondence:
Philipp Sebastian Arndt (parndt@ucsd.edu)

Discussion: https://doi.org/10.5194/egusphere-2024-1156

Comments from the reviewers are given in black.
Our responses are given in red.
**Quotes from the submitted manuscript are given in bold red.**
Proposed amendments or additions to the revised manuscript are given in blue in the Times New Roman font.

References that were already included in the original manuscript are cited in-text only, in the same format as in the submitted manuscript. New references are added to the end of this document in full.

**Referee Comment 3 (Sammie Buzzard)**

RC3: 'Comment on egusphere-2024-1156', Sammie Buzzard, 08 Jul 2024
Citation: https://doi.org/10.5194/egusphere-2024-1156-RC3

**Summary**

This paper presents two algorithms that together allow the retrieval of supraglacial lake depths using ICESat-2. This method is scalable beyond the case studies presented and therefore a useful contribution to the community that I would strongly recommend for publication.

Understanding surface melt, and lake depths is key for predicting ice shelf stability, and datasets of lake depths are limited. In situ data is scarce and remote sensing methods are necessary for providing validation and calibration for models as well as understanding the development of these lakes so this work will clearly be of use to the community.

While there are methods existing to determine lake depths remotely e.g. using Landsat-8, this to my knowledge is one of only a small number using ICESat-2, the advantage of this particular method appears to be scalability (although see my comments below on this).

The methodology and results presented are in my opinion sufficient for publication, and my recommendations for changes to the manuscript are mostly minor.

Thank you so much Sammie for taking the time to read and review our manuscript, and for your positive, thoughtful and constructive comments. In particular, we believe that your comments suggesting more detailed comparisons to similar studies as well as better explanations of the limitations of our method have helped to significantly improve the manuscript.

**General Comments**

1. It would be good to clarify the limitations of the algorithm e.g. max/min lake widths/ lengths/ depths detected clearly early on in the paper (and state how this compares to other methods).
Based on this and some of the other referee comments we realize that we have not made it clear enough early on in our writing that ICESat-2 data coverage is sparse. The most important limitation of the algorithm that readers need to understand is that due to its coarse track spacing and 3-month return period ICESat-2 cannot sample all supraglacial lakes on the ice sheets – or even close to all of them. ICESat-2's unique selling point is its ability to accurately and directly measure water depths from space, but ICESat-2 does not provide a comprehensive map of the locations and spatial extents of melt lakes (plus, there are image-based methods out there that are much better-suited).

   - We propose to rewrite part of the introduction, to make it clear that the fundamental limitation of this method is that a data set based on ICESat-2 alone cannot find all the supraglacial lakes, and that we do not intend to claim that it can: "While ICESat-2 has the unique capability to make direct and accurate measurements of water depth from space, its fundamental limitation is spatial coverage. ICESat-2 data are limited to discrete, one-dimensional ground tracks that are coarsely spaced on the Earth's surface (~9.9 km between neighboring reference tracks and ~3.3 km between all neighboring beam pair tracks at 70°N/S) with a relatively long revisit time of three months. This means that no supraglacial lake depth data product derived from ICESat-2 alone is able to provide samples of all (or even nearly all) supraglacial lakes on the ice sheets: ICESat-2's track spacing means that the majority of lakes form in locations that ICESat-2 ground tracks never sample, and the three-month return period means that for a significant number of ground tracks ICESat-2 never passes over at the time at which melt lakes are visible. ICESat-2 is also unable to penetrate optically thick clouds, thus further limiting the amount of data available for water depth measurements. While ICESat-2 data alone cannot be used to continuously monitor melt lake volumes,..."
   - We will further emphasize track spacing in Sect. 3.5 (Study regions and time span) by adding the following: "Our two study areas cover latitudes from 68.2°N to 72.1°N in Greenland and latitudes from 68.4°S to 74.0°S in Antarctica, meaning that ICESat-2 track spacing is similar over the two regions: in Central West Greenland RGT spacing varies from ~8.8 km in the north to ~10.8 km in the south; over the Amery Catchment RGT spacing varies from ~7.9 km in the south to ~10.7 km in the north."

   The other limitations mentioned here ("max/min lake widths/ lengths/ depths"):
   - widths/lengths are the same in ICESat-2, since the ground track is one-dimensional
   - max depths: This is the limitation that everyone seems to be interested in. It is controlled by how far ICESat-2's green light can penetrate the water column under ideal conditions (very clear water, high bottom reflectivity). To clarify this early on, we will add the following to the introduction (line 41):
     "This allows ICESat-2 to measure water depths up to 41 m under ideal conditions (very clear water and high bottom reflectivity), with typical accuracies of about 0.5 m (Dietrich et al., 2024)."
   - Min depths: This is indirectly answered by the above. Since typical accuracies are around 0.5 m, it is difficult to detect lakes shallower than that because the lakebed return is hardly distinguishable from the lake surface return. As you noted, FLUID can not detect any ATL03 lake segments that are less than 0.35 m below the lake surface at their deepest along-track location (which is about 26 cm in refraction-corrected water depth), since all photons from the bottom return would be removed with the surface. However, in practice, we have not encountered any examples where a lakebed <0.35 m would be discernible in the photon data. Furthermore, such lake segments are often short and tend to be in locations with relatively flat surrounding surface topography, which makes it also harder to detect them in the first place. We

will make a short note on this in the section where this is mentioned (3.3.3 SuRRF step 2: fitting the lakebed, line 405):

"Note that this imposes a theoretical minimum depth threshold for detection on lake segments: ATL03 segments need to exhibit a bottom return signal at least 0.35 m below the lake surface (or 0.26 m in refraction-corrected water depth) at their deepest along-track point to be considered by SuRRF. However, in practice, such shallow lake segments do not have a discernible bathymetric signal since typical depth retrieval accuracies for ICESat-2 are on the order of 0.5 m (Dietrich et al., 2024)."

- Min lengths: Since FLUID assesses the flatness of major frames (~ 140 m in length), lakes with open-water surfaces of a shorter extent in along-track distance are not guaranteed to be detected. However, in practice, significantly shorter lake segments are regularly detected by FLUID. This is likely because the return signal from flat water surfaces tends to be much stronger than the return signal from surrounding ice surfaces, which makes even short, flat water surfaces dominate the overall distribution of photon elevations within a major frame. We will make a note about this in the relevant section (3.2.1 FLUID step 1: identification of flat water surfaces, line 210):

"Since FLUID assesses the flatness of the surface of full major frames that cover an along-track distance of ~ 140 m, lake segments with shorter open-water surfaces are not guaranteed to be detected by FLUID. However, lake segments that are significantly shorter than 140 m are regularly detected by FLUID in practice. This is because the return signal from flat water surfaces is typically much stronger than the return signal from the surrounding ice surfaces, which makes even very short, flat water surfaces dominate the overall distribution of photon elevations within a major frame."

- Max lengths: There is no limitation to this. (While we have not detected any supraglacial lake segments with a water extent larger than 10 km, this is not a limitation of our algorithms. Supraglacial lakes just typically aren't that large. An exception is a large melt lake that sometimes forms on Amery Ice Shelf and can get up to 70 km long, but ICESat-2 ground tracks do not cross this lake at the right orientation for lake segments to become this long.)

- Comparison with other methods: Since we explain that in practice these limitations arise from the ICESat-2 data themselves, there is no reason to compare these specific limitations with regard to other ICESat-2-based methods.

2. This method appears to only be scalable if you live in the US based on the information in the paper. Some comments on this would be useful e.g. can your methodology transfer to a reader's local (i.e. institutional) supercomputer or does it need to be a National level facility? How possible would it be to do this?

Our Python code works without the need for using any OSG services: it can be run on a local computer or on any computing cluster (locally or in the cloud) in the provided singularity container. We used the OSG Open Science Pool because it provided us with free computational infrastructure. We do not have the resources to scale this method up to ice-sheet-wide implementation on commercial cloud computing platforms at a feasible price point, and during initial development PA had no access to NASA-provided computational resources due to the COVID19 pandemic preventing in-person fingerprinting of non-US citizens. We believe that the OSG Open Science Pool is the most accessible option for large-scale implementation, but there are unfortunately some limitations to who can use it. As a non US-based researcher, one could gain access to the OSG Open Science Pool if collaborating with a US-based researcher or an institution that operates its own access point. Since we want our methods and data to be accessible and encourage others to use it in their own research, we will explain this in more detail in the Code and data availability section:

"The FLUID/SuRRF source code is freely available at https://doi.org/10.5281/zenodo.10905941 (Arndt and Fricker, 2024a). To execute this code, users need to create a free NASA Earthdata login for ICESat-2 data access.

The source code contains a singularity container in which this version of FLUID/SuRRF can be executed. The main Python script detect_lakes.py can be run either locally on any individual ATL03 granule, or on many granules in parallel on any computing cluster that supports the specified computing environment or the use of singularity containers. In this study we present our implementation of FLUID/SuRRF on the OSG Open Science Pool because it provided us with free computational infrastructure. Due to funding mandates, free access to the OSG Open Science Pool is limited to researchers contributing to a US-based project at an academic, government, or non-profit organization, or researchers affiliated with any project or institution that operates its own local access point. This means that to implement FLUID/SuRRF on the OSG Open Science Pool as described here, you need to have at least one collaborator on your team to whom the above criteria apply. This collaborator can register your project with OSG on the Open Science Pool. Then, anyone contributing to the project can register for an account on OSG Connect to gain access to the Open Science Pool. For more information, see https://osg-htc.org/services/open_science_pool.html and https://osg-htc.org/about/organization. More information is also included in the README file."

**Detailed Comments**

- Line 44: It would be good to have a more detailed comparison with the Datta and Wouters and Leeuwen methodologies to explain what the differences are here. Given the method presented here is promoted as scalable would it be possible to do a direct comparison to their results?
Datta and Wouters (2021) and Leeuwen (2023) demonstrate that ICESat-2 data can be used to improve depth extraction from imagery, which we use to motivate our study. To clarify this, and based on other referee's comments, we are adding the following to the introduction:
"While ICESat-2 data alone cannot be used to continuously monitor melt lake volumes, several case studies have shown that ICESat-2 depth measurements can be used to constrain parameters in models that estimate lake volumes from satellite imagery (Datta and Wouters, 2021; Leeuwen, 2023; Lutz et al., 2024). For instance, Datta and Wouters demonstrated that it is possible to accurately extrapolate depths along ICESat-2's ground track segments to the full lake basins that these segments intersect. To be able to use ICESat-2 to improve depth estimates of supraglacial lakes in locations where (and at times when) ICESat-2 measurements are not directly available, it will be necessary to rely on statistical methods that can generalize the relationship between water depth and reflectance for a particular passive optical sensor under a wide variety of conditions and independently of the availability training data that is close-by in space and time (Hastie et al., 2009). For this to work effectively, the data that is used to train statistical learning models capable of multiple non-linear regression for representing a complex depth-reflectance relationship need to adequately cover the parameter space defined by the combination of predictors that are included (Markham and Rakes, 1998; Wang et al., 2022). Since ICESat-2 observations of melt lakes are relatively sparse, it is therefore crucial to to obtain as many ICESat-2 depth estimates as possible from different locations and times (and thus under a wide variety of environmental conditions) to be able to effectively use ICESat-2 to improve monitoring of meltwater volumes across the ice sheets. This suggests that large-scale extraction of accurate supraglacial lake depths from a wide range of ICESat-2 photon-level data in combination with concurrent optical satellite imagery can provide a labeled training data set enabling the application of machine learning methods (e.g., Leeuwen, 2023) capable of generating a well-constrained data-driven model for ice-sheet-wide lake volume estimation (Melling et al., 2023)."

However, neither study published their depth estimates (Datta and Wouters published Matlab code, Leeuwen did not publish any data or code with their Master's thesis), so there is no easy way for us to compare results. Fricker et al. (2021) is a great resource, which compares different algorithms for supraglacial lake depth estimates from ICESat-2 including four estimates from Watta and an earlier version of SuRRF. We will point the reader to this paper for depth estimate comparisons, and also include the Watta results for the lakes shown in Fricker et al. (2021) in Fig. 12 (see below).

We will add the following comparison between SuRRF and alternative methods to the end of section 4.2.2 (line 612):

[revised manuscript text omitted]

- Line 57: Most of the Greenland ablation zone rather than most of Greenland?
  We will include this, as suggested: "Across most of the Greenland Ice Sheet's ablation zone, ..."

- Line 60: There are enough notable examples of melt away from the grounding zone (over the ice shelves of Larsen B, George VI) or on grounded ice (e.g. Corr et al. found more than a quarter or meltwater features on grounded ice https://essd.copernicus.org/articles/14/209/2022/) that maybe 'mostly' could be changed?
  Thanks for pointing out ice shelves that have meltwater features away from the grounding zone (such as Larsen B and George VI). We will change our statement to include these examples. Besides this, we believe that our statement is quite accurate. While Corr et al. found more than a quarter of meltwater features on grounded ice, the vast majority of these were indeed located at low elevations near the grounding zones of ice shelves. Corr et al. actually is a great reference for this, so we will include a citation to their study. We will also change the terminology from **"grounding lines"** to "grounding zones", as suggested. Given the above, we propose to change our statement to: "...is mostly observed on the floating ice shelves and at low elevations near their grounding zones (Stokes et al. 2019; Corr et al., 2022)."

- Line 73: Acceleration isn't always the case. Some of these references are fairly old, Davison et al. have a nice review (https://doi.org/10.3389/feart.2019.00010), although there may be updated literature.
  To reflect this, we will change the phrasing to: "..., which has the potential to lubricate the bedrock and cause

acceleration of ice flow due to enhanced basal sliding." Davison et al. (2019) is a great reference for this and they explain very nicely how rapid lake drainage *usually* results in transient ice speed-up and net acceleration, so we will include a citation to this review.

- Line 75: I'm not sure this is strictly true it's been observed, more than it's been suggested as a possible mechanism.
  To reflect this, we will change the phrasing to: "... but recent observations suggest that this mechanism is also driving ice flow speed-ups on the Antarctic Ice Sheet."

- Line 97: If we're providing estimates we can also model this (e.g. https://tc.copernicus.org/articles/12/3565/2018/tc-12-3565-2018.html for individual lakes) but remote sensing is quicker/ more scalable but it's not technically true we have to rely on it.
  We will change **"estimates"** to "observations" to fix this, and simultaneously remind the reader that this section is about *observations* of supraglacial lakes.

- Line 152: Is it a good assumption that lakes are non-turbid? Certainly for sea ice ponds we can assume they are turbulent (of course they are shallower) but I wonder if e.g. it's a very windy day when the measurements are taken how much this impacts the flatness.
  The statement that we make is simply saying that the lakebed can be visible to ICESat-2 when water is non-turbid. If the water is too turbid, then ICESat-2 will not see a bathymetric signal. We are not making the assumption or claiming that all or most lakes are non-turbid. We would like to argue that windy days, lake flatness and turbulence are outside of the scope of an introductory sentence, which tells the reader that "if water is clear, ICESat-2 can look through it". (and we admit that we also don't quite understand ourselves how all these factors are related within the scope of line 152)
  To attempt to broadly answer the questions brought up here: it seems like most supraglacial lakes are sufficiently non-turbid for ICESat-2 to obtain a bathymetric signal, which makes sense because there are less sediments on the ice sheets than there are around inland water bodies, and algal growth in meltwater lakes seems to be fairly limited. The fact that ICESat-2 does not obtain a lakebed signal whenever turbidity is too high is actually helpful, since in this case the optical methods based on imagery that are commonly used will usually result in wrong depth estimates (e.g., Lutz et al., 2024).
  To discuss the effects of wind (also requested by RC2), we propose to re-write the part starting on line 227:
  "Afterpulses only become noticeable when the sensor is nearly or fully saturated, which means they often appear in ATL03 data over supraglacial lakes because smooth open water surfaces (i.e. the surface of stationary water bodies that are not affected by wind) can result in specular reflection. This suggests that the presence of wind ripples increases the likelihood of detecting a lake with a clear bathymetric signal in ATL03 data by preventing sensor saturation and afterpulsing (Lu et al., 2019; Tilling et al., 2020) and also explains why we observe afterpulsing more frequently near the (more wind-shielded) margins of melt lakes than over their (more wind-exposed) interior."

- Line 207: How were these numbers determined (empirical observation is a little broad e.g. how many lakes were examined?)
  We agree that "empirical observations" is a bit too unspecific. We used a trial-and-error approach based on a number of different hand-picked granules that featured a broad range of surface conditions, including some that contained many lakes and some that did not contain any lakes. For clarification, we propose to change this part to the following:
  "Based on these assumptions, and using a trial-and-error approach, we defined the following thresholds on the density ratios that need to hold for a major frame to pass the flatness check: $d_0/d_1 \geq 2$, $d_0/d_2 \geq 5$, $d_0/d_3 \geq 10$ and $d_0/d_4 \geq 100$. As part of this trial-and-error approach, we manually assessed the effects of

tweaking the above thresholds on a number of hand-picked granules, which we judged to be likely representative of various possible environments, to ensure adequate performance (i.e. granules without surface melt vs. pervasive surface melt, granules with smooth vs. rough background topography, granules containing ice-covered and partially ice-covered lakes, granules containing slush areas, granules containing exposed bedrock, partially cloudy granules, weak vs. strong beam data, night- vs. daytime acquisitions, etc.)."

- Line 259 onwards: What if the lake is e.g. 0.92m deep, would the bathymetric signal still get picked up? I think this is what you are saying in line 273 but you could clarify how you might discern the two, or if it is likely to be possible.
  We agree that we did not describe this clearly enough. Since we remove afterpulses in saturated pulses only, bathymetric signals will be preserved in unsaturated pulses – unless if all the pulses are saturated, in which case an actual flat bathymetric signal is practically indistinguishable from an afterpulse in the point cloud data. To clarify this, we propose to add the following to the end of the paragraph that starts on line 259:
  "Since this procedure removes photons in saturated pulses only, true bathymetric signals that overlap with the elevation of a known afterpulse are still retained as long as they appear in any unsaturated pulses. However, if all pulses within an along-track section of the data are saturated, any true bathymetric signals from a flat lakebed at the elevation of a known afterpulse will be removed from the data because they are practically indistinguishable from the afterpulses that we expect to see in the point cloud under such highly saturated conditions."

- Line 314: Can you determine why there is no signal from the lake bed here? Is it in the ice covered lake areas?
  It appears we did not include this in the text, but these two major frames overlap exactly with the partially ice-covered area. Thanks for catching that. We will add the following to the end of the paragraph:
  "These two major frames visibly overlap with the location of a thin partial ice cover near the lake's northern shore (Fig. 7, panel a), which explains why some of the lakebed is occluded in the photon cloud. While such areas, where part of the lakebed is occluded, may not pass the bathymetry check, they are later included in the data that make up a full ICESat-2 lake segment, as explained in the next section."

- Line 319: Not sure if the word 'each' here is a typo?
  We will delete **"each"**.

- Line 321: See my general comment about lake sizes, what does a 'few' mean here, please be more specific. Is this related to the 10 in equation 3? (Or if not where does that come from?)
  We agree that we should be more specific here. The phrase "along-track extent" was poorly chosen. The 10 in equation 3 is indeed what describes this: the maximum separation between two clusters to be merged is 10 major frames, so clusters that are separated by more than 1.4-1.6 km are not merged. To clarify this we propose to replace (ii) on line 321 with:
  "(ii) a ground track rarely crosses the same lake in two distinct locations that are separated by more than about 1.5 km."
  We will further explain equations (2) and (3) in plain language after they are defined:
  "Equation (2) states that neighboring clusters are only merged if their respective lake surface elevations are within 0.1 m of each other, and Eq. (3) further states that neighboring clusters are only merged if they are separated by ten major frames that did not pass the bathymetry check, or less (about 1.5 km). This means that if FLUID encounters the unlikely but possible scenario in which a ground track crosses two arms of the same lake, which are separated in along-track distance by more than ten major frames, then these two crossings are considered to be separate lake segments and returned as two separate files in the output data rather than being merged together into

one lake segment. If these two conditions do not result in…"

- Line 350: I'm not sure what you mean here by 'removing any lakes that fully overlap with another lake'. Does that not just make them the same lake (and is some information about that lake then lost)?
We agree that **"fully overlap"** sounds confusing. We propose to change this to:
"Since this expansion of the along-track ranges of lake segments can create lake segments that overlap, the set of buffered lake segments is corrected by separating partially overlapping lake segments at the midpoint of their along-track overlap and removing any lake segments that are fully contained within another lake segment."
[Explanation: Note that a lake segment being fully contained within another one means that if we remove it no information is lost, we just remove information that was duplicated. This does not happen often in practice, but imagine a large lake that has a physically thin but optically thick ice cover that stretches for more than 1.5 km. If ICESat-2 obtains a bathymetric signal on both sides of the ice cover, then FLUID will consider those two stretches of signal to be two separate lake segments (Eq. 2 above), but since the ice cover is physically thin (and therefore flat and very close in elevation to the water surface), FLUID will also expand the along-track ranges of both segments across the entire lake, including its ice cover. In this situation, there will be two lake segments that both have data for the entire lake, so one of them is a duplicate and needs to be removed. → We consider this to be such a fringe case that we do not think it needs to be explained in full in the manuscript.]

- Line 405: So is this a limitation on minimum lake depths that can be detected?
Yes, 0.35 m is a theoretical minimum, but in practice lakes < 0.5 m deep are not very reliably detected due to noise in the data, and the fact that such lake segments are often short and tend to be in locations with relatively flat surrounding surface topography. See our answer to the first general comment.

- Line 435: What is the situation where this could happen? Does this suggest a lack of confidence in surface retrievals or is the algorithm picking up something else (an ice lens?).
This happens anywhere where the fit to the photons that remain after removing the ones reflected from the water surface is above the water level of the lake. The most frequent example for this is the surface of the ice that surrounds a lake (e.g., Fig 7, panel d: in about the first and last 200 m in along-track distance, the red line is clearly above the elevation of the water surface), but it can also happen when there is a thick ice cover on a lake. SuRRF is usually quite confident in the fit in these situations, because surface returns above water tend to be much stronger than bathymetric returns. Either way, SuRRF is fitting something that is at an elevation higher than the surface of the water, so we can be generally confident that water depth is zero there.

- Section 4.1.2: Could you compare with the Moussavi et al dataset? That goes up to 2020 according to the data description: https://www.usap-dc.org/view/dataset/601401
The Moussavi et al. (2020) dataset includes only one of the four melt seasons considered in this manuscript. Furthermore, it is >70GB in size and has to be downloaded as a single file by contacting the data center and making an individualized manual request. As such, it is not very accessible and not very practical to use. While this dataset includes estimates of lake depths, in Section 4.1.2 we assess FLUID lake _detection_ only (FLUID itself does not provide lake depths). This means we are concerned with whether the locations of detected lakes overlap with the locations detected in imagery. For this purpose, the Moussavi dataset would be overkill, and only give us information about one out of four melt seasons. Mapping the spatial extent of melt lakes based on imagery is quite simple, and Tuckett et al. (2021) made it even easier by sharing their methods. Due to all the reasons above, we believe that it makes much more sense in the context of this section to simply map the extent of supraglacial lakes using Google Earth Engine and compare their extents to the locations of FLUID lake segments.

To make this more intuitive to the invested reader, we will include high-resolution maps of the Landsat 8 lake extents and ICESat-2 lake segments for all the melt seasons included in the paper in the updated release of the supplementary material at https://zenodo.org/doi/10.5281/zenodo.10901826 (see also attached to the end of this document).

To compare our SuRRF _depth_ estimates to RTE-based depth estimates from imagery that are similar to the Moussavi et al. (2020) dataset, we will include the RTE depth estimated reported in the papers that we use to assess depth estimation accuracy in section 4.2.2 (Fricker et al., 2021 and Melling et al, 2024). The Sentinel-2 based depth estimates in Fricker et al. (2021) were contributed by M. Moussavi based on the dataset that you mentioned. We also include Antarctic lake depth estimates based on Landsat 8 (contributed to Fricker et al. 2021 by J. Spergel and J. Kingslake and based on Spergel et al. 2021) and Greenland lake depths based on Sentinel-2 from Melling et al. 2024. (see comment above about comparison to alternative methods)

- Section 4.2.2: How many lakes are there actually in track 81? Is this something that could be determined manually to get an idea of false positives/ negatives from SuRFF?
  There are eight lakes in track 81 GT2L. We described this in section 4.1.2:
  **"To evaluate whether FLUID detects all supraglacial lakes that have bathymetric data in this ATL03 ground track, we manually inspected the data for any evidence of meltwater and determined whether any such along-track contained a clearly discernible return from a lakebed. Out of 25 along-track segments with meltwater, we judged that only eight were supraglacial lakes with a clear bathymetric signal."**
  We did further examine same-day Sentinel-2 imagery to visually verify that ICESat-2's ground track does not overlap with any melt lakes that are not captured in the ATL03 data in track 81 GT2L. We did not mention this in the submitted manuscript, so we will add this to the revised manuscript:
  "We further evaluated whether the ATL03 photon cloud misses any supraglacial lakes that track 81 GT2L crosses, by mapping it over a mosaic of Sentinel-2 scenes from the same day (same as used in Fricker et al. 2021). Based on visual inspection, the ICESat-2 ground track crossed supraglacial lakes that were clearly distinguishable in the imagery only in the same eight locations that we had also judged to be lake segments in the ATL03 data. Most other ICESat-2 segments that showed evidence of surface water in ATL03 also showed some evidence of meltwater in the imagery, and were associated with ice-filled crevasses, narrow melt channels, likely areas of slush, or melt lakes with an opaque ice cover (for which depth determination is not possible)."
  In Section 4.2.2, we refer back to section 4.1.2 where it was explained.
  **_"For the 16 potential melt lakes that FLUID identified in ICESat-2 track 81, GT2L on Jan 2nd, 2019 over the Amery catchment (Sect. 4.1.2 Fricker et al., 2021), SuRRF assigned a nonzero lake quality score only to the eight data segments which we had manually determined to be supraglacial lakes with a clear bathymetric signal."_**
  To clarify that the eight lake segments mentioned in these sections are the only ones present in track 81, we will replace **"clear"** with "discernible" (because "clear" leaves it open to interpretation whether there could still be lakes segments with bathymetric signals in the track, whose bathymetric signals just aren't super obvious).

- Line 630: 'this' satellite rather than 'a' satellite. Datasets exist for other satellites.
  ICESat-2 is the only satellite (to date, and likely for a while into the future) that can directly measure lake depths from space, so **"a satellite"** is correct. However, we never stated this explicitly in the manuscript, and should absolutely do so. To make this evident to the reader, we will state this early on in the introduction (line 37): "Launched in 2018, NASA's Ice, Cloud and land Elevation Satellite (ICESat-2) laser altimeter became the first (and thus far only) satellite capable of making direct, accurate water depth measurements from space, …"

- Line 740: What were these equations based on? I understand you used trial and error but they are complex equations that must have had a starting point.
  We propose to add the following to the end of Appendix C to illustrate our thought process:
  "While the quality heuristics $q_i$ were obtained by trial-and-error, the starting points for this approach were based on the following assumptions and observations:
    - The starting point for $q_1$ was the idea that major frames with a smaller number of detected bathymetry peaks are less likely to have a consistent signal from a lakebed. The way the equation for $q_1$ is designed, major frames with a very small fraction of detected bathymetry peaks (0-30% of sub-segments) are disproportionately penalized, since such small numbers of peaks are much more likely to be noise.
    - The starting point for $q_2$ was the idea that major frames with less prominent peaks represent either a weak, inconsistent bathymetric signal or noise. Here, $q_2$ is equal to the mean prominence of peaks when the fraction of detected bathymetry peaks $f < 0.5$. However, for fractions larger than 0.5, the assumption is that the bathymetric signal is consistent enough that even smaller mean peak prominence suggests that the bathymetric signal is clearly visible.
    - The starting point for $q_3$ was the observation that in most cases supraglacial lake segments with a usable bathymetric signal have fairly small lakebed slopes. Therefore, we do not expect the lakebed elevation of a major frame with a good signal to span a large elevation range. In contrast, we observed that major frames with detected potential bathymetric peaks that span a very large elevation range are often due to noise in the data. Based on this, we designed the equation for $q_3$ such that its value drops off once the total elevation range of detected bathymetry peaks within the major frame (of length ~140 m) exceeds 5 m. There are, however, some lake segments with a clear bathymetric signal that do exhibit a large range of lakebed elevations (often due to a single burst of noise being detected as a potential bathymetric peak). Therefore, we made sure that the value for $q_3$ is large enough for major frames to still pass the bathymetry check if $\Delta h$ is very large but its other quality heuristics $q_i$ are closer to 1.
    - The starting point for $q_4$ was the idea that in most cases bathymetry peak elevations will align along a smooth surface in the along-track distance direction rather than randomly fluctuate (which is usually the case for noise). Here, we penalize every "direction change" (i.e., wherever a peak has two neighbors and its elevation constitutes a local minimum or maximum). We allow for random fluctuations of up to 0.5 m per peak detection without a large penalty, since we observed that a vertical photon spread of up to about this value is quite possible even for lake segments with a somewhat "fuzzy", yet clearly distinguishable return signal from the lakebed.

A figure that illustrates these quality heuristics is available in the supporting information at https://doi.org/10.5281/zenodo.10901826 (Arndt and Fricker, 2024c). (and shown below for convenience)

[Figure]

[Figure]

325 ICESat-2 ground track segments over lakes
198 unique melt lakes sampled

ICESat-2 lake segments overlapping
with Landsat 8 melt extent

ICESat-2 lake segments not overlapping
with Landsat 8 melt extent

Landsat 8 melt extent
polygons with ICESat-2 data

Landsat 8 melt extent
polygons without ICESat-2 data

outlines
of unique
supraglacial
lakes

**Central West Greenland: 2020**

[Figure]

175 ICESat-2 ground track segments over lakes
114 unique melt lakes sampled

ICESat-2 lake segments overlapping
with Landsat 8 melt extent

ICESat-2 lake segments not overlapping
with Landsat 8 melt extent

Landsat 8 melt extent
polygons with ICESat-2 data

Landsat 8 melt extent
polygons without ICESat-2 data

outlines
of unique
supraglacial
lakes

**Amery Catchment: 2018-19**

[Figure]

720 ICESat-2 ground track segments over lakes
385 unique melt lakes sampled

ICESat-2 lake segments overlapping
with Landsat 8 melt extent

ICESat-2 lake segments not overlapping
with Landsat 8 melt extent

Landsat 8 melt extent
polygons with ICESat-2 data

Landsat 8 melt extent
polygons without ICESat-2 data

outlines
of unique
supraglacial
lakes

**Amery Catchment: 2020-21**

[Figure]

28 ICESat-2 ground track segments over lakes
25 unique melt lakes sampled

ICESat-2 lake segments overlapping
with Landsat 8 melt extent

ICESat-2 lake segments not overlapping
with Landsat 8 melt extent

Landsat 8 melt extent
polygons with ICESat-2 data

Landsat 8 melt extent
polygons without ICESat-2 data

outlines
of unique
supraglacial
lakes

---

## Author Comment (AC4)

**Response to CC1**

A Framework for Automated Supraglacial Lake Detection and Depth Retrieval in ICESat-2 Photon Data Across the Greenland and Antarctic Ice Sheets

Philipp Sebastian Arndt[1] and Helen Amanda Fricker[1]

[1] Scripps Polar Center, University of California San Diego, 8885 Biological Grade, La Jolla, CA 92037, USA

Correspondence:
Philipp Sebastian Arndt (parndt@ucsd.edu)

Discussion: https://doi.org/10.5194/egusphere-2024-1156

Comments from the reviewers are given in black.
Our responses are given in red.
**Quotes from the submitted manuscript are given in bold red.**
Proposed amendments or additions to the revised manuscript are given in blue in the Times New Roman font.

References that were already included in the original manuscript are cited in-text only, in the same format as in the submitted manuscript. New references are added to the end of this document in full.

**Community Comment 1 (Bert Wouters)**

CC1: 'Comment on egusphere-2024-1156', Bert Wouters, 26 Jun 2024
Citation: https://doi.org/10.5194/egusphere-2024-1156-CC1

*Notes:*
- *B. Wouters, H.A. Fricker and P. Arndt are all co-authors of Fricker et al. (2021);*
- *P. Arndt was a referee for Datta and Wouters (2021).*

I agree with the two other reviewers that this is a well-written and important contribution, presenting an elegant method to derive supraglacial lake bathymetry. Nevertheless, I would like to comment on the two statements below, in the Introduction and Summary sections:

- L47-53: *Previous ICESat-2 studies have been limited to applying depth estimation methods to a handful of manually picked lakes or data granules, with no clear pathway to large-scale computational implementation across the ATL03 data catalog, which comprises hundreds of terabytes of unstructured point cloud data (Neumann et al., 2023b). To address this challenge, we have created a fully automated and scalable algorithm for lake detection and depth determination from ICESat-2 data.*
- L646-651: *ICESat-2 data had not previously been used at scale for this purpose because its photon-level product comprises hundreds of terabytes of unstructured point cloud data along spatially discrete ground tracks, which makes it difficult to integrate the data with spatially continuous data in existing workflows. To address this challenge, we have presented the fully automated, two-step FLUID/SuRRF algorithm for the detection and depth determination of supraglacial lakes on the ice*

*sheets in ICESat-2 photon data, and proposed a computational framework that allows for its large-scale implementation across any desired ice sheet drainage basins and melt seasons.*

Whereas it is true that other methods have not been used at such a large scale as in the manuscript, the Watta algorithm (Datta and Wouters, 2021) is fully automated (i.e. it detects potential lake locations based on a flatness criterion and then estimates the bathymetry, similar to the framework presented in this manuscript) and it is designed to be run in parallel, allowing large-scale application at any location or time period. The reason Watta hasn't been applied at large scale is a lack of computational infrastructure.

It would be nice to acknowledge that the automated and scalable nature of the method, while advantageous, is not a unique selling point. This doesn't take away that there is plenty of novelty in the manuscript to merit publication. Emphasizing these specific innovations would strengthen the manuscript, in my view.

Thanks very much Bert for your positive and constructive comment, and for bringing this particular issue to our attention. First off, we would like to express our appreciation for the scientific contributions that you made in Datta and Wouters (2021) and acknowledge that the results you published had a large impact on motivating our own study and informing our opinion that it is crucial to retrieve and publicly share as many ICESat-2 water depth estimates as possible, to improve our ability to continuously monitor supraglacial meltwater volumes across the ice sheets. We explain this in the later paragraphs.

Thank you for pointing out that automation and scalability are selling points that are not unique to our algorithm. We agree that this should be pointed out in the manuscript. We will explain that other automated and scalable algorithms exist for this purpose, and that the new contribution of our study is the fact that in addition to developing another such algorithm, we also propose a computational framework for applying it at ice-sheet-wide scale and then demonstrate that it works in practice.

For completeness, we would like to note here that in our own experience, the task of computationally scaling up FLUID-SuRRF was hugely more time- and work-intensive than developing the initial algorithm. At the time, we already considered it to be automated because it ran smoothly on a few dozen granules that we had used to develop empirical thresholds and tune parameters. It is amazing that multiple automated and scalable algorithms have been proposed for this purpose, and in Fricker et al. (2021) we pointed out that ensemble estimates from various different algorithms outperformed any single algorithm on the small number of lakes presented in that case study. However, just because such automated and theoretically scalable algorithms exist, does not mean that it is easy at all to use them to generate large, comprehensive data products. This is why we spent three years since the initial algorithm was presented in Fricker et al. (2021) working on actually implementing FLUID-SuRRF at scale.

We would like to make it clear we are in no way trying to diminish the scientific significance of Watta. In Datta and Wouters (2021) you are showing many other significant and very impressive scientific results that go well beyond what we are presenting in our manuscript. In fact, our own study is largely motivated by the fact that you successfully demonstrated that depths and volumes of supraglacial lakes can be accurately estimated by combining along-track depth measurements from Watta/ICESat-2 with concurrent imagery from various passive optical sensors. As we explain in the manuscript, this is absolutely necessary for using ICESat-2 to improve estimates for continuously monitoring melt lake volumes through space and time. In Datta and Wouters (2021), you use imagery to extrapolate depths along ICESat-2 segments to the full lake basins that these segments intersect. You convincingly showed that this indeed works very well, and these results inspired us to attempt to go one step further: To be able to use ICESat-2 to empirically estimate depths/volumes of supraglacial lakes in locations and at times where/when ICESat-2 measurements *are not directly available*, it is necessary to rely on statistical methods that can generalize the depth-reflectance relationship for a particular

passive optical sensor independently of the availability training data that is closeby in space and time. For this to work, the data that are used to train statistical learning models capable of multiple non-linear regression for representing this depth-reflectance relationship need to adequately cover the parameter space defined by the combination of predictors that are included.

If instead we attempted to train a model to learn the relationship between Sentinel-2 reflectance and water depth from the lake segments provided in Datta and Wouters (2021), the training data would be limited to estimates based on five ICESat-2 tracks from five different dates, all of which cluster closely together near Sermeq Kujalleq and Sarqardliup Sermia. This means that we would sample associated Sentinel-2 data from only a handful of satellite overpasses, which would likely be acquired under similar conditions (e.g., sun angles). If we now show the model unseen Sentinel-2 data that originated under different conditions and ask it to estimate water depth, it will likely not be able to extrapolate well. Since ICESat-2 observations are fundamentally quite sparse, we believe that it is necessary to obtain as many ICESat-2 depth estimates from different locations and times to be able to effectively use ICESat-2 to improve monitoring of meltwater volumes across the ice sheets: in this case, we argue that "more data is always better". We hope that this explains why we considered it to be so important to take on the task to scale up ICESat-2 lake detection on depth determination algorithms, and to share the resulting data.

For reference, see Domingos (2012) for an explanation of how more data usually improves statistical / machine learning models. This article also mentions the need for models to generalize, as well as the benefits of using ensemble methods. While somewhat outdated for such a fast-moving field, the article does a great job at explaining the fundamental principles.

We propose the following changes:

- It seems that we have not detailed our motivation for large-scale extraction of ICESat-2 melt lake depths (based on promising recent results from combining ICESat-2 depths with imagery / fundamental principles of statistical learning) clearly enough. We therefore propose to rewrite the corresponding section in the introduction in its own paragraph:
  "While ICESat-2 data alone cannot be used to continuously monitor melt lake volumes, several case studies have shown that ICESat-2 depth measurements can be used to constrain parameters in models that estimate lake volumes from satellite imagery (Datta and Wouters, 2021; Leeuwen, 2023; Lutz et al., 2024). For instance, Datta and Wouters demonstrated that it is possible to accurately extrapolate depths along ICESat-2's ground track segments to the full lake basins that these segments intersect. To be able to use ICESat-2 to improve depth estimates of supraglacial lakes in locations where (and at times when) ICESat-2 measurements are not directly available, it will be necessary to rely on statistical methods that can generalize the relationship between water depth and reflectance for a particular passive optical sensor under a wide variety of conditions and independently of the availability training data that is close-by in space and time (Hastie et al., 2009). For this to work effectively, the data that are used to train statistical learning models capable of multiple non-linear regression for representing a complex depth-reflectance relationship need to adequately cover the parameter space defined by the combination of predictors that are included (Markham and Rakes, 1998; Wang et al., 2022). Since ICESat-2 observations of melt lakes are relatively sparse, it is therefore crucial to to obtain as many ICESat-2 depth estimates as possible from different locations and times (and thus under a wide variety of environmental conditions) to be able to effectively use ICESat-2 to improve monitoring of meltwater volumes across the ice sheets. This suggests that large-scale extraction of accurate supraglacial lake depths from a wide range of ICESat-2 photon-level data in combination with concurrent optical satellite imagery can provide a labeled training data set enabling the application of machine learning methods (e.g., Leeuwen, 2023) capable of generating a well-constrained data-driven model for ice-sheet-wide lake volume estimation (Melling et al., 2023)."

- L47-53: We had to restructure the entire paragraph to make sure we are not misleading the reader by making it sound like automation and scalability are unique selling points of our algorithm. Here are our proposed changes:

  "While automated and scalable algorithms for lake detection and depth retrieval in ATL03 photon data have been proposed (e.g., Datta and Wouters, 2021; Xiao et al., 2023), in practice no previous ICESat-2 studies have applied supraglacial lake depth estimation methods to more than a handful of manually picked lake segments or data granules, or presented a straightforward pathway to large-scale computational implementation across the ATL03 data catalog, which comprises hundreds of terabytes of unstructured point cloud data (Neumann et al., 2023b). To address this challenge, we present a framework for ice-sheet-wide implementation of our own fully automated and scalable algorithm for along-track lake segment detection and depth determination from ICESat-2 data. Here, we present this algorithm, apply it to two entire drainage basins in Greenland and Antarctica (Sect. 3.5, Fig. 2) using distributed high-throughput computing, and demonstrate its performance for two full melt seasons."

- L646-651: We believe that by now it should be sufficiently clear to the reader that other automated and scalable algorithms do exist, but have not been applied at scale. However, we propose to switch the sentence structure to make it more clear that we are "addressing this challenge" primarily by taking on the actual implementation rather than by proposing an additional automated algorithm:

  "ICESat-2 data had not previously been used at scale for this purpose because its photon-level product comprises hundreds of terabytes of unstructured point cloud data along spatially discrete ground tracks, which makes it difficult to integrate the data with spatially continuous data in existing workflows. To address this challenge, we have proposed a computational framework that allows users to detect lake segments and determine their water depths across all available ICESat-2 data for any desired ice sheet drainage basins and melt seasons. Using distributed high-throughput computing, this framework applies the fully automated, two-step FLUID/SuRRF algorithm to large numbers of ICESat-2 ATL03 photon data granules in parallel. To test our method, we applied FLUID-SuRRF to all available ICESat-2 data over two drainage basins, one on the Antarctic Ice Sheet and one on the Greenland Ice Sheet, for a high-melt and a low-melt summer."

**References:**

- Domingos, P. (2012). A few useful things to know about machine learning. Communications of the ACM, 55(10), 78-87.

- Hastie, T., Tibshirani, R., & Friedman, J. H. (2009). The elements of statistical learning: data mining, inference, and prediction (Vol. 2, pp. 1-758). New York: springer.

- Lutz, K., Bever, L., Sommer, C., Humbert, A., Scheinert, M., & Braun, M. (2024). Assessing supraglacial lake depth using ICESat-2, Sentinel-2, TanDEM-X, and in situ sonar measurements over Northeast Greenland. EGUsphere, 2024, 1-25.

- Markham, I. S., & Rakes, T. R. (1998). The effect of sample size and variability of data on the comparative performance of artificial neural networks and regression. Computers & operations research, 25(4), 251-263.

- Wang, J., Lan, C., Liu, C., Ouyang, Y., Qin, T., Lu, W., ... & Philip, S. Y. (2022). Generalizing to unseen domains: A survey on domain generalization. IEEE transactions on knowledge and data engineering, 35(8), 8052-8072.